# FRET-FISH probes chromatin compaction at individual genomic loci in single cells

Ana Mota [1,2], Szymon Berezicki [1,2], Erik Wernersson [1,2], Luuk Harbers [1,2], Xiaoze Li-Wang[1,2], Katarina Gradin[1,2], Christiane Peuckert[3], Nicola Crosetto [1,2,4] & Magda Bienko [1,2,4] ✉

Chromatin compaction is a key biophysical property that influences multiple DNA transactions. Lack of chromatin accessibility is frequently used as proxy for chromatin compaction. However, we currently lack tools for directly probing chromatin compaction at individual genomic loci. To fill this gap, here we present FRET-FISH, a method combining fluorescence resonance energy transfer (FRET) with DNA fluorescence in situ hybridization (FISH) to probe chromatin compaction at select loci in single cells. We first validate FRET-FISH by comparing it with ATAC-seq, demonstrating that local compaction and accessibility are strongly correlated. FRET-FISH also detects expected differences in compaction upon treatment with drugs perturbing global chromatin condensation. We then leverage FRET-FISH to study local chromatin compaction on the active and inactive X chromosome, along the nuclear radius, in different cell cycle phases, and during increasing passage number. FRET-FISH is a robust tool for probing local chromatin compaction in single cells.

Three major types of chromatin have been described: active, repressed and inactive[1–3]. Active chromatin corresponds to the euchromatin described in early electron microscopy studies and is characterized by active gene expression, histone depletion around transcriptional start sites, and histone marks promoting the recruitment and function of transcriptional complexes, such as histone H3 acetylated on lysine 27 (H3K27ac) or tri-methylated on lysine 4 (H3K4me3)[4]. Repressed chromatin marks genomic regions that are not being actively transcribed in a particular cell type and is enriched in histone H3 tri-methylated on lysine 27 (H3K27me3)[5]. On the other hand, inactive chromatin encompasses genomic regions that are typically less transcribed compared to active chromatin and is decorated by heterochromatic proteins like heterochromatin protein 1 (HP1) or histone H3 methylated on lysine 9 (H3K9me)[6]. It is generally accepted that different chromatin types correspond to different levels of compaction defined as chromatin mass per volume. Traditionally, gross differences in chromatin compaction have been revealed by fluorescence microscopy using DNA intercalating fluorescent dyes, such as 4′,6-diamidino-2-phenylindole (DAPI), and more recently, by genetically

encoded photo-switchable DNA labels such as H-NS-based indicator for nucleic acid stainings (Hi-NESS)[7]. However, the use of these labels alone does not allow direct measurements of chromatin compaction at defined loci. Moreover, DNA intercalating dyes can preferentially bind to certain genomic sequences, such as in the case of DAPI having a higher affinity for AT-rich DNA stretches[8]. To avoid potential biases, several alternative approaches have been deployed. For example, a photo-switchable protein fused to histone H2B (H2B-PATagRFP) was used in combination with super-resolution microscopy, revealing that inactive chromatin has lower compaction and mobility compared to active chromatin[9]. Fusion of histone H2B to a fluorescent protein also revealed substantial heterogeneity in chromatin compaction across the cell nucleus, by combining fluorescence lifetime imaging (FLIM) and fluorescence energy transfer (FRET)[10,11]. One limitation of these approaches is that they are blind to the underlying DNA sequence and thus are not informative about local compaction at specific loci. Instead, DNA fluorescence in situ hybridization (FISH) allows visualizing selected DNA loci. However, DNA FISH has not been used to systematically probe for chromatin compaction.

[1]Department of Microbiology, Tumor and Cell Biology, Karolinska Institutet, Stockholm SE-17165, Sweden. [2]Science for Life Laboratory, Tomtebodavägen 23A, Solna SE-17165, Sweden. [3]Stockholm University, The Department of Molecular Biosciences, The Wenner-Gren Institute, Stockholm, Sweden. [4]Human Technopole, Viale Rita Levi-Montalcini 1, 20157 Milan, Italy. ✉e-mail: magda.bienko@ki.se

Following the advent of massively parallel sequencing technologies, several methods have been developed to probe the accessibility of chromatin genome wide, including DNase-seq[12], MNase-seq[13], and ATAC-seq[14]. These methods do not measure chromatin compaction directly, but rather probe the accessibility of the linear genome to DNA nucleases (DNase-seq), restriction enzymes (MNase-seq) or transposases (ATAC-seq). Although it is assumed that chromatin accessibility and compaction are inversely correlated, the exact relationship between these two biophysical properties remains unknown. For instance, a locus that is densely coated by various proteins might be inaccessible to nucleases or transposases, yet its chromatin might occupy a relatively large volume and hence have relatively low compaction. Thus, new tools that can probe chromatin compaction directly at defined loci are needed to gain deeper insights into the biophysical structure of chromatin in the nucleus.

Towards this goal, here we present a microscopy-based method integrating DNA FISH with FRET to measure chromatin compaction at individual gene loci in single cells (FRET-FISH). We show that chromatin compaction at a given locus can be detected by FRET-FISH by targeting the region with carefully designed and empirically tested probes consisting of oligonucleotides (oligos) carrying alternating FRET donor and acceptor dyes. We demonstrate that compaction measured by FRET-FISH strongly correlates with accessibility measured by ATAC-seq. We then leverage FRET-FISH to study chromatin compaction at select loci at different radial positions in the nucleus as well as changes in local compaction in cells treated with drugs that induce global changes in chromatin condensation in cultured cells with increasing passage number as well as in different cell-cycle stages. We conclude that FRET-FISH is a sensitive assay for studying chromatin compaction at selected gene loci and assessing different compaction states within a cell population.

## Results

### FRET-FISH implementation

FRET-FISH is based on the hybridization—in fixed cells—of DNA FISH probes composed of oligos coupled to two fluorescent dyes with overlapping spectra targeting two proximal DNA sequences, so that if the two sequences are closer than ~100 Å (depending on the pair of FRET dyes used) FRET can be detected (Fig. 1a). We reasoned that a probe consisting of multiple oligos targeting a given genomic locus and carrying alternating FRET donor (D) and acceptor (A) dyes should enable probing chromatin compaction at that locus by measuring the resulting FRET signal (Fig. 1a). Since the efficiency of FRET is influenced by the dipole orientation and the molecular distance between D and A dyes, we initially conceived three different FRET-FISH probe designs (Fig. 1b). To test which probe design yields the highest FRET efficiency, we targeted a region of ~20 kilobases (kb) encompassing the human *MYC* gene locus (Supplementary Data 1). The first design is essentially identical to the iFISH probe design that we previously described[15] and consists of primary oligos with a target (T) sequence complementary to the genomic DNA target (60 nucleotides (nt) long instead of 40 nt as in iFISH) flanked by a left (L) and right (R) adapter sequence (Fig. 1b, Design 1). The L and R sequences are needed for PCR during the production of the probes and serve as docking sites for fluorophore-conjugated detection oligos (L* and R*, respectively). In the second design, the L and R sequences of each primary oligo are extended with a left and right stabilizing sequence (LSS and RSS, respectively), where the 3′ 6 nt of the RSS in one oligo are complementary to the 5′ 6 nt of the LSS in the next primary oligo along the linear genomic target (Fig. 1b, Design 2). We reasoned that this design should stabilize the proximity between D and A dyes, thus enhancing FRET efficiency. In the third design, the stabilizing sequence is added to the L* and R* detection oligos, so that the 3′ 6 nt of an L* oligo can anneal to the 5′ 6 nt of the R* oligo bound to the next primary oligo along the linear genomic target (Fig. 1b, Design 3). In all three cases, we designed each

probe to contain primary D-A oligo pairs with a minimum distance of 5 nt between the 3′ of the T sequence of a primary D oligo and the 5′ of the T sequence of the next primary A oligo along the linear genomic target (Methods: 'FRET-FISH probe design'). In these initial proof-of-principle experiments, we designed the probes based on our extensive experience with iFISH probes[15], which are typically composed of 96 oligos and target a locus of ~8 kb. We found that this number of oligos and probe size or span (i.e., the genomic distance from the first to the last oligo in a probe) is a good compromise between resolution (i.e., the minimum size of a locus that can be detected) and sensitivity. However, we reasoned that, for FRET-FISH, we might need to increase the number of D and A oligos per probe, since enough D and A oligos need to be simultaneously bound to their target region for FRET to occur. Therefore, we designed these FRET-FISH probes to contain ~130 oligos for each fluorophore type (Supplementary Table 1). We used Cy3 and Cy5 as D and A dyes, respectively (Förster radius, $R_0 = 52$ Å), since they have been widely used in FRET experiments due to their relatively high brightness and lower price compared to other fluorophores.

To test each probe design, we hybridized HAP1 human chronic myeloid leukemia cells inside custom-designed 9-well silicone-coated coverslips to minimize technical variability between samples and compare all three probe designs within the same experiment (Supplementary Fig. 1a and Methods: 'FRET-FISH sample preparation' and 'FRET-FISH probe production and hybridization'). As a proxy for chromatin compaction, we calculated a FRET-FISH score (expressed as percentage) by dividing the signal intensity detected in the FRET channel (Cy3 excitation and Cy5 emission) by the sum of the signal intensity in the D (Cy3 excitation and Cy3 emission) and FRET channel (Methods: 'Image processing and identification of FRET-FISH dots' and 'FRET-FISH score calculation'). A higher score indicates higher compaction, whereas a lower score indicates a more relaxed chromatin. In two independent experiments, all three probe designs produced readily detectable FRET signals, with Design 3 yielding the highest FRET-FISH score (39.5% ± 6.8%, mean ± s.d.) (Fig. 1c, d and Supplementary Fig. 1b). This is most likely attributable to the presence of complementary annealing sequences in the D oligos in Design 3, which stabilize the primary D-A oligo pairs that are in physical proximity (Fig. 1b). Surprisingly, in both replicate experiments, Design 2 yielded the lowest FRET-FISH score (24.8% ± 5.2%, mean ± s.d.), possibly because the stabilization sequence in the primary oligos hinders the energy transfer between the D and A dyes or because it acts as a quencher (Fig. 1c and Supplementary Fig. 1b). Independently of the probe design, the score was consistently lower in control samples in which only D or A primary oligos were hybridized together with the detection oligos, demonstrating the specificity of our approach (Fig. 1c and Supplementary Fig. 1b). Importantly, the FRET-FISH scores were similar between controls, indicating that the influence of cross-excitation was homogenous across all the designs tested. We obtained similar results by performing three experiments using a different setup in which we compared two designs side-by-side in 6-well-chambered coverslips (Supplementary Fig. 1c–f). Altogether, these results demonstrate that proximity between in situ hybridized oligos carrying alternating FRET acceptor and donor dyes can be detected by measuring the resulting FRET signal in fixed cells.

### Optimization of FRET-FISH probe design to measure local chromatin compaction

We then sought to further optimize the FRET-FISH probe design to measure local chromatin compaction at select genomic loci. We reasoned that the three probe designs described above might not allow detecting changes in chromatin compaction, since the D and A oligos bind very closely along the linear genome and already yield a high FRET signal. We therefore designed probes consisting of D and A oligos separated by larger linear genomic distances, aiming at increasing

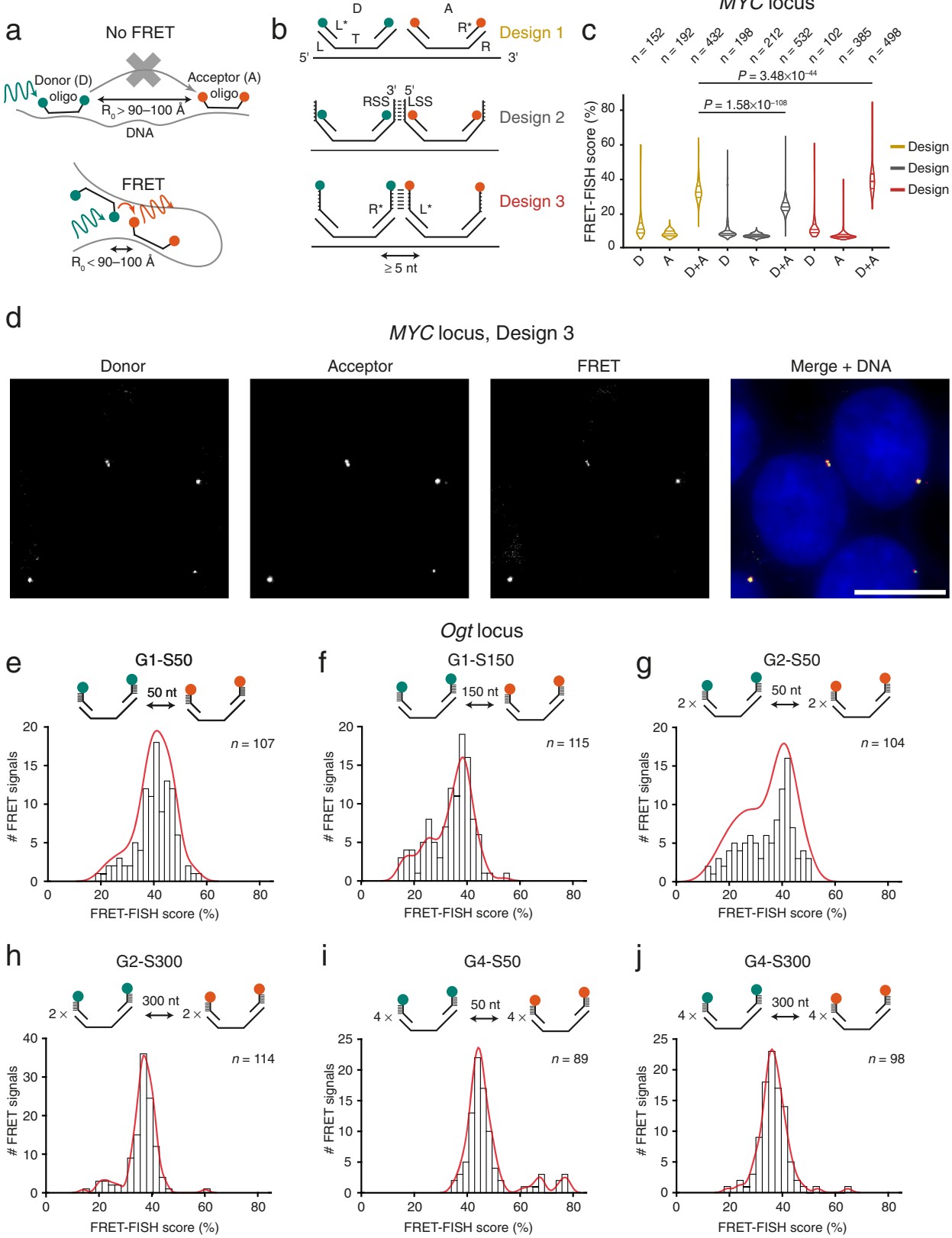

the dynamic range of compaction detectable by FRET-FISH. We tested three different spacing (S) distances (50, 150, and 300 nt) between consecutive oligos, since in a previous study FRET dyes positioned at similar distances on purified nucleosomes yielded detectable FRET signals[16]. We also modified the probe design so that each probe would consist of alternating groups (G) of 1, 2 or 4 D oligos followed by

groups of 1, 2 or 4 A oligos (Fig. 1e–j). Since the oligo hybridization efficiency in DNA FISH most likely never reaches 100%, we reasoned that having consecutive groups of multiple D and A oligos in the same probe would maximize the chances of multiple D-A pairs to be in sufficient spatial proximity to yield a FRET signal. We named the different designs G1-S50, G1-S150, G2-S50, G2-S300, G4-S50, and

**Fig. 1 | FRET-FISH implementation and optimization. a** Scheme of FRET-FISH. A FRET-FISH probe consisting of alternating oligos labeled with donor (D) or acceptor (A) fluorescent dyes is hybridized to its DNA target in fixed cells. If most D and A oligos are farther than the Förster distance, $R_0$, no FRET is detected. If a substantial number of D and A oligos are closer than $R_0$, FRET is detected. **b** FRET-FISH probe designs tested. Each probe contains ~200 pairs of D and A primary oligos containing a sequence complementary to the genomic target (T) flanked by two orthogonal adapter sequences (L and R) used as docking sites for PCR primers during probe production and for complementary detection oligos (L* and R*) conjugated to FRET donor (green) and acceptor (orange) dyes. In design 2, L and R adapter sequences are extended with a 6 nt stabilizing sequence (LSS and RSS), allowing annealing of the RSS of one D oligo with the LSS of the next A oligo. In design 3, the LSS and RSS sequences are added to the 5′ and 3′ end of the L* and R* detection oligos, respectively. **c** Distributions of FRET-FISH scores obtained with a probe targeting the *MYC* gene in human HAP1 cells, for each probe design shown in (**b**). D, Probes containing only donor oligos. A, probes containing only acceptor

oligos. D+A, probes containing both donor and acceptor oligos. *n*, number of FRET signals analyzed. *P*, Wilcoxon test, two-tailed. Violins extend from the minimum to the maximum value and horizontal lines represent (from top to bottom) the 75th percentile, the median, and the 25th percentile of the distributions. **d** Maximum z-projections exemplifying donor (Cy3 excitation, Cy3 emission), acceptor (Cy5 excitation, Cy5 emission) and FRET (Cy3 excitation, Cy5 emission) signals in two nuclei of HAP1 cells hybridized with a FRET-FISH probe targeting *MYC* designed based on Design 3 in (**b**). Blue, DNA stained with Hoechst 33342. Scale bar, 10 μm. **e–j** FRET-FISH score distributions for six different types of FRET-FISH probes targeting the *Ogt* locus in female mouse embryonic fibroblasts (MEFs). Probe design is represented above each histogram. G, group of D or A oligos. S, spacing between consecutive oligos. Cy3 and Cy5 were used as FRET donor and acceptor dyes, respectively. Red lines, kernel density estimation function. *n*, number of FRET signals analyzed. Source data for all the plots shown in the figure are provided as a separate Source Data file.

G4-S300 (Fig. 1e–j). As a proof-of-concept, we designed probes targeting the mouse *Ogt* gene locus located on chromosome (chr) X and hybridized female mouse embryonic fibroblasts (MEFs), reasoning that we might detect differences in chromatin compaction since this locus is known to escape female X chromosome inactivation, albeit at low (~6%) frequency[17] (Supplementary Fig. 2a, Supplementary Data 1, Supplementary Table 1, and Methods: 'FRET-FISH probe design', 'FRET-FISH sample preparation', and 'FRET-FISH probe production and hybridization'). All the six probe designs yielded detectable FRET signals, however the FRET-FISH score distributions differed depending on the probe design (Fig. 1e–j). For design G1-S50, G1-S150 and G2-S50 the score distributions were relatively broad, indicating their ability to detect a range of compaction states. Moreover, the distributions of the G1-S150 and G2-S50 designs were clearly bimodal, featuring a higher mode on the right—presumably corresponding to *Ogt* loci with more compacted chromatin—and a lower mode on the left likely corresponding to a less compacted state (Fig. 1f, g). Of note, when we examined the distributions of the FRET acceptor intensities—in the absence of donor excitation—we could only detect unimodal distributions (Supplementary Fig. 2b). This indicates that the ability of FRET-FISH to detect compaction differences depends on the generation of FRET and not simply on the intensity of the signals generated by FRET-FISH probes alone. Importantly, we obtained similar FRET-FISH score distributions using a different cell line (NIH3T3 mouse fibroblasts), with only some small differences detected, indicating the generalizability of our approach (Supplementary Fig. 2c).

Next, we aimed at testing whether the binding of FRET-FISH probes to their targets might perturb the chromatin structure of the targeted loci. To this end, we designed two classical iFISH probes (~8 kb in size) targeting the regions flanking the locus to which the six *Ogt* FRET-FISH probes bind (Supplementary Data 1). To assess whether the *Ogt* locus is perturbed upon binding of the FRET-FISH probes, we hybridized the two iFISH probes with or without each FRET-FISH probe separately (omitting the D and A detection oligos) and then measured the physical distance (in 3D) between the two latter probes, in multiple single cells (Methods: 'Testing whether FRET-FISH probes disrupt the target locus conformation'). The distributions of 3D distances largely overlapped between different probe designs, indicating that, upon hybridization, FRET-FISH probes do not disrupt the conformation of the targeted loci (Supplementary Fig. 2d, e). Based on these results and following our reasoning that a larger spacing between D and A oligos might allow for a broader dynamic range of compaction detection, we selected G1-S150 as the default FRET-FISH probe design for all subsequent experiments.

Lastly, we tested a different pair of FRET dyes—Alexa Fluor 488 (AF488) and Alexa Fluor 594 (AF594) ($R_0 = 60$ Å)—which are characterized by a higher quantum yield and lower fluorescence signal degradation over time compared to Cy3 and Cy5. Using the same *Ogt*

G1-S150 probe and AF488 and AF594 dyes also yielded clearly detectable FRET signals (Supplementary Fig. 3a). The resulting FRET-FISH scores were lower than those obtained with Cy3 and Cy5, however this difference can be explained by the higher level of crosstalk and bleed-through observed with the latter dyes (Supplementary Fig. 3a–c). Of note, when we labeled the *Ogt* S1-G150 probe with AF488 and AF594 dyes, the resulting FRET-FISH score distribution displayed two separate modes even more clearly than in the case of probes labeled with Cy3 and Cy5 dyes, indicative of a higher sensitivity for AF488 and AF594 in detecting different chromatin compaction states (Supplementary Fig. 3d). For these reasons, we adopted AF488 (FRET donor) and AF594 (FRET acceptor) as default FRET-FISH dyes for all our subsequent experiments.

## FRET-FISH validation

Next, we sought to assess the reproducibility of FRET-FISH and validate our method. To this end, we first designed FRET-FISH probes against six genes on mouse chrX (*Atp2b3*, *Ddx3x*, *Kdm5c*, *Magix*, *Pbdc1*, and *Tent5d*) including three genes (*Atp2b3*, *Magix*, and *Tent5d*) that are constitutively inactivated on one chrX copy and three genes (*Ddx3x*, *Kdm5c*, and *Pbdc1*) that frequently escape inactivation (so-called 'escapees')[17] (Supplementary Data 1). In three independent experiments performed on female MEFs, the six probes yielded the expected FISH dot counts and reproducible FRET-FISH score bimodal distributions in G1-phase cells, suggesting that these loci can be found in two distinct chromatin compaction states (Fig. 2a and Supplementary Fig. 4a–c). Of note, lowering the number of oligos per probe led to a loss of the left mode in the typical FRET-FISH score distributions—which likely corresponds to the lower compaction state—indicating that, with the current design of FRET-FISH probes, ~200 oligos per channel are needed to detect both the higher and the lower compaction state (Supplementary Fig. 4d, e). The difference in the FRET-FISH score of the two homologues in each cell was rather broad, even though there was a significant correlation between the two homologues, for all the six loci examined (Supplementary Fig. 5a, b). Importantly, the FRET-FISH score distributions were highly correlated between experimental replicates, highlighting the reproducibility of FRET-FISH (Supplementary Fig. 5c).

Since lack of chromatin accessibility is used as a proxy for chromatin compaction, we next compared chromatin compaction measurements by FRET-FISH with chromatin accessibility previously measured by ATAC-seq in the same MEF cell line[17]. For all the six genes monitored by FRET-FISH on chrX, the mean FRET-FISH score was inversely correlated with the corresponding ATAC-seq score, indicating that local chromatin compaction and accessibility are related (Fig. 2b and Methods: 'Comparison between FRET-FISH and ATAC-seq'). Of note, the mean intensity measured in the A channel was not significantly correlated with ATAC-seq, indicating that the intensity of

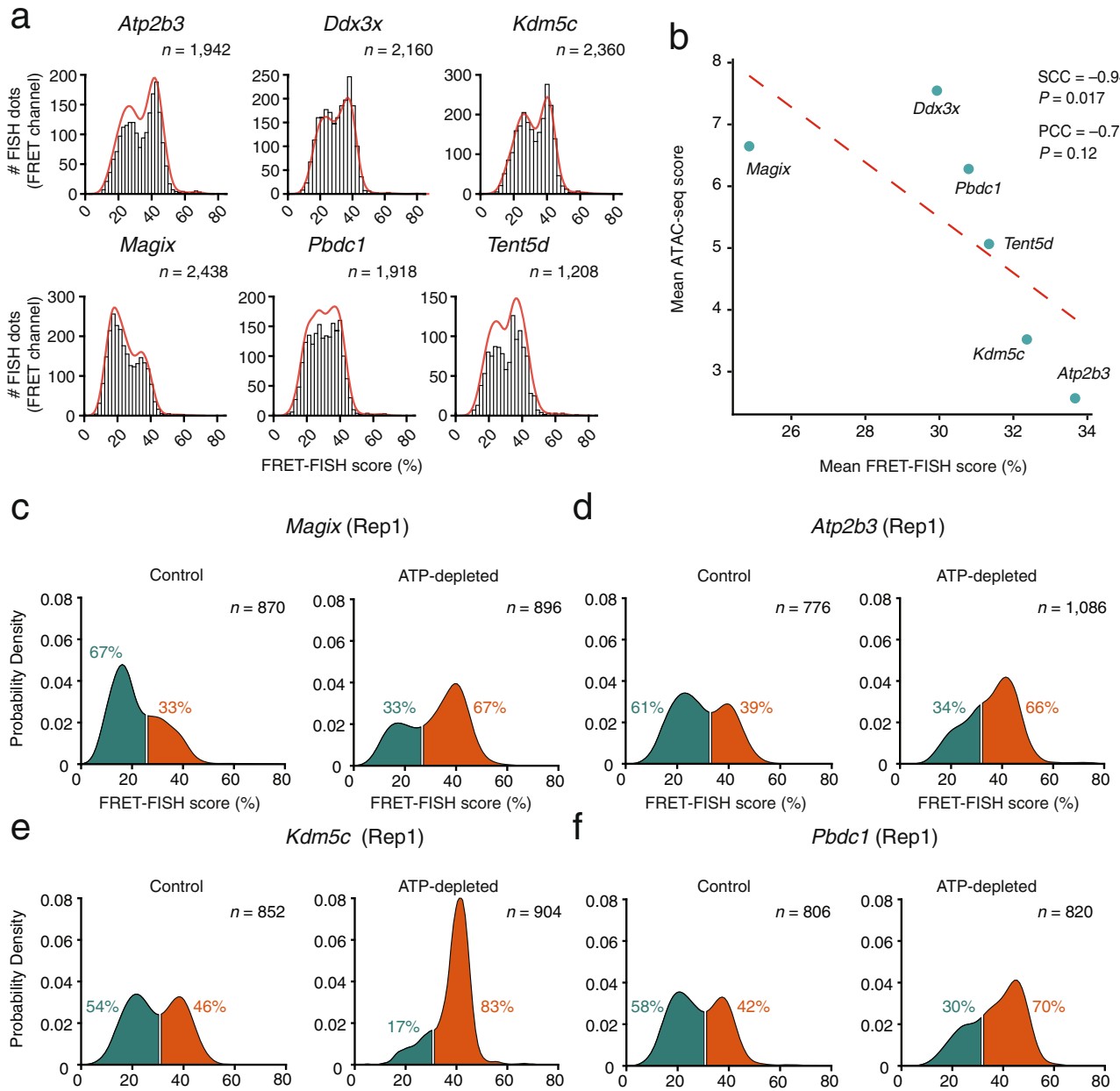

**Fig. 2 | FRET-FISH validation. a** FRET-FISH score distributions for six loci targeted on chrX in female mouse embryonic fibroblasts (MEFs). Measurements from three replicate experiments pooled together are shown. Red lines, kernel density estimation function. *n*, number of FRET signals analyzed from three independent experiments. **b** Correlation between the mean FRET-FISH score of the six gene loci shown in (**a**) and the mean ATAC-seq score of the corresponding genomic regions. SCC Spearman's correlation coefficient, PCC Pearson's correlation coefficient, *P* Wilcoxon test, two-tailed. Dashed red line, linear regression fit. **c–f** FRET-FISH score distributions for four of the loci shown in (**a**) in MEFs treated (ATP-depleted) or not (Control) with sodium azide and 2-deoxy-d-glucose, in one of two replicate (Rep) experiments. The inflection point in each bimodal distribution was used to separate between the lower (green) and higher (orange) FRET-FISH score mode corresponding, respectively, to a less and more compact chromatin state. The percentages indicate the proportion of FRET signals in each group. *n* number of FRET signals analyzed. Source data for all the plots shown in the figure are provided as a separate Source Data file.

DNA FISH signals per se is not a good proxy of local chromatin compaction (Supplementary Fig. 6a).

We then sought to validate our FRET-FISH measurements using fluorescence lifetime imaging microscopy (FLIM), which is not affected by crosstalk or bleed-through and therefore should, in principle, provide more accurate FRET efficiency measurements[18]. To this end, we used the FRET-FISH probe targeting the *Magix* locus and performed FLIM in the same female MEF cell line described above (Methods: 'FLIM-FRET'). In line with the results obtained with intensity-based FRET measurements, the distribution of the FRET efficiency measured by FLIM was also bimodal and both approaches yielded very similar mean FRET-FISH scores (26.9 ± 4.5% and 24.9 ± 9.1%, respectively, for FLIM and for intensity-based FRET measurements, mean ± s.d.) even though the calculation differs between the two approaches (Supplementary Fig. 6b and Methods: 'FLIM-FRET data analysis'). These results demonstrate that FRET-FISH coupled with FLIM can be used to assess local chromatin compaction, further validating our method. However, we note that FLIM requires a dedicated setup that is available only in few specialized laboratories. Furthermore, the throughput of FLIM is considerably lower compared to classical widefield epifluorescence

microscopy (in total, we managed to analyze only 30 FRET-FISH signals). Therefore, for future applications of FRET-FISH, we recommend relying on intensity-based FRET measurements.

Lastly, to further validate our method, we tested whether it could detect local changes in chromatin compaction associated with global changes in chromatin condensation. To this end, we treated the same MEF cell line described above with a combination of sodium azide and 2-deoxy-d-glucose, which causes intracellular ATP depletion and, in turn, increases the intracellular pool of polyamines and divalent cations that neutralize the negative charges on DNA, leading to chromatin condensation[19] (Methods: 'Induction of chromatin condensation by ATP depletion'). The same treatment also results in drastic reduction of gene expression[20]. To confirm these effects in our experimental setup, we assessed global chromatin compaction by quantifying the fluorescence intensity of nuclei stained with the DNA intercalator dye, Hoechst 33342 (Methods: 'Induction of chromatin condensation by ATP depletion'). In two replicate experiments, the nuclear intensity significantly increased upon ATP depletion, whereas the area of nuclei 2D projections decreased (Supplementary Fig. 7a–d). In parallel, we assessed global transcription by labeling nascent transcripts with the fluorescent uridine analog 5-ethynyluridine (EU) (Methods: 'Induction of chromatin condensation by ATP depletion'). As expected, the total nuclear fluorescence corresponding to nascent transcripts drastically decreased upon treatment with sodium azide and 2-deoxy-d-glucose, confirming the validity of our experimental setup (Supplementary Fig. 7e, f). We then tested whether these global changes in chromatin condensation are reflected at the level of individual genes. To this end, we performed FRET-FISH using four of the six probes targeting different genes on mouse chrX described above (Atp2b3, Kdm5c, Magix, and Pbdc1). In two independent experiments, the FRET-FISH score was significantly higher in ATP-depleted cells compared to controls, for all four gene loci examined (Fig. 2c–f and Supplementary Fig. 7g–j). Notably, the higher FRET-FISH score mode in the bimodal distributions consistently increased upon the treatment, further indicating that this mode corresponds to a higher compaction chromatin state. Altogether, these results highlight the reproducibility of FRET-FISH and demonstrate that our method can detect local changes in chromatin compaction that mirror global changes in chromatin condensation.

## Comparison of chromatin compaction between loci on the active and inactive chrX

Having successfully implemented and validated FRET-FISH, we then sought to investigate whether FRET-FISH would detect differences in compaction for loci located on chrX in female cells where one of the two chrX copies is inactive and forms a more compacted chromosome territory[21]. The latter mainly contains Polycomb-repressed chromatin[22,23], which has been associated with high chromatin packaging density[2]. To this end, we combined FRET-FISH using probes against the Magix and Kdm5c gene loci with single-molecule RNA FISH (smFISH)[24] with a probe targeting the X-inactive specific transcript (Xist) that marks the inactive chrX copy, in the same female MEF cell line described above (Supplementary Fig. 8a, Supplementary Data 1, and Methods: 'FRET-FISH combined with single-molecule FISH for Xist'). As expected, the inactive chrX territory occupied a lower volume, however the combination of FRET-FISH with smFISH led to a deterioration of the DNA FISH signal quality to the point that we could not detect any more a clear bimodality in the FRET-FISH score distributions even when pooling measurements from both active and inactive loci (Supplementary Fig. 8b–d). This could be caused by the procedure used to preserve RNA for smFISH or by the fluorescent dye used to detect Xist interfering with the FRET-FISH dyes. To test whether deconvolution would improve the quality of the FRET-FISH signals, we applied our newly developed open-source deconvolution software Deconwolf[25] (Methods: 'FRET-FISH score calculation for loci on the active and inactive chrX'). Indeed, after deconvolution the

resulting FRET-FISH score distributions appeared bimodal, although not as clearly as in our previous experiments in which we did not combine FRET-FISH with smFISH (Supplementary Fig. 8e–g). Especially in the case of Magix, which is a non-escapee gene, the FRET-FISH score distribution of the inactive homologue was shifted towards higher score values indicating that this locus is associated with a more compacted chromatin state on the inactive chrX copy, although the difference between the two distributions did not reach statistical significance (Supplementary Fig. 8g–j). The fact that the FRET-FISH score distribution of the active homologue appears bimodal for both Magix and Kdm5c suggests that the observed bimodality for the loci tested on chrX cannot be fully explained by the presence of one inactive copy of chrX in female cells. Of note, previous studies showed that the reduction in chromosome territory volume during chrX inactivation likely results from high-order restructuring of the chromosome instead of increased compaction of individual gene loci[21,26].

## FRET-FISH detects chromatin compaction differences along the nuclear radius

We then wondered whether FRET-FISH would also capture different compaction states at gene loci located on autosomes. To this end, we designed six FRET-FISH probes targeting six different loci on chr18 and assessed the resulting FRET-FISH score distributions (Supplementary Data 1). In two replicate experiments, the FRET-FISH score distributions were reproducibly bimodal and, as expected, the FRET-FISH score was anti-correlated with the ATAC-seq score for the same loci (Fig. 3a, b and Supplementary Fig. 9a–c). Of note, three of the targeted genes (Minar2, Grxcr2, and Rik) are comprised within constitutive lamina associated domains (cLADs), whereas the other three genes (AtpSa1, Hspa9, and Nars) fall within inter-LAD regions (iLADs)[27] (Supplementary Data 1). We therefore wondered whether the FRET-FISH score is higher for genes in cLADs compared to genes in iLADs, since cLADs are typically found within highly compacted and repressed chromatin at the nuclear periphery and around nucleoli[27]. Indeed, the FRET-FISH scores of cLADs genes on chr18 were significantly higher compared to genes in iLADs on the same chromosome (Fig. 3c and Supplementary Fig. 10a). Furthermore, the distance to the nuclear lamina of the cLAD genes examined was significantly shorter than for iLAD genes, in line with the typical localization of cLADs near the nuclear lamina (Supplementary Fig. 10b). This observation further prompted us to inquire whether the radial position of a given locus in the nucleus might influence its associated FRET-FISH score distribution. To this end, we divided 3D segmented nuclei into four concentric nuclear layers of equal volume and calculated the FRET-FISH score distribution for each layer, for each of the six loci targeted on chr18 as well as the six loci on chrX (Methods: 'Analysis of chromatin compaction along the nuclear radius'). In all cases, the right mode in the bimodal FRET-FISH score distributions progressively decreased moving from the peripheral layer inwards (Fig. 3d and Supplementary Fig. 10c). In line with these observations, a single-cell analysis revealed that the more peripheral copy of each gene displayed a significantly higher FRET-FISH score compared to the more central copy (Supplementary Fig. 10d). Altogether these results demonstrate that the radial position of a locus in the nucleus influences its chromatin compaction state and suggest that the observed bimodality in the FRET-FISH score distributions might be a general property of genes modulated by radial position.

## Local chromatin compaction varies during the cell cycle

We then wondered whether the bimodality in the FRET-FISH score distributions might also be affected by the cell cycle, during which chromatin undergoes dramatic condensation changes reaching the highest compaction during mitosis[28]. To this end, we calculated the FRET-FISH score for three of the loci assessed on chrX (Atp2b3, Kdm5c, and Magix) in different phases of the cell cycle as determined based on

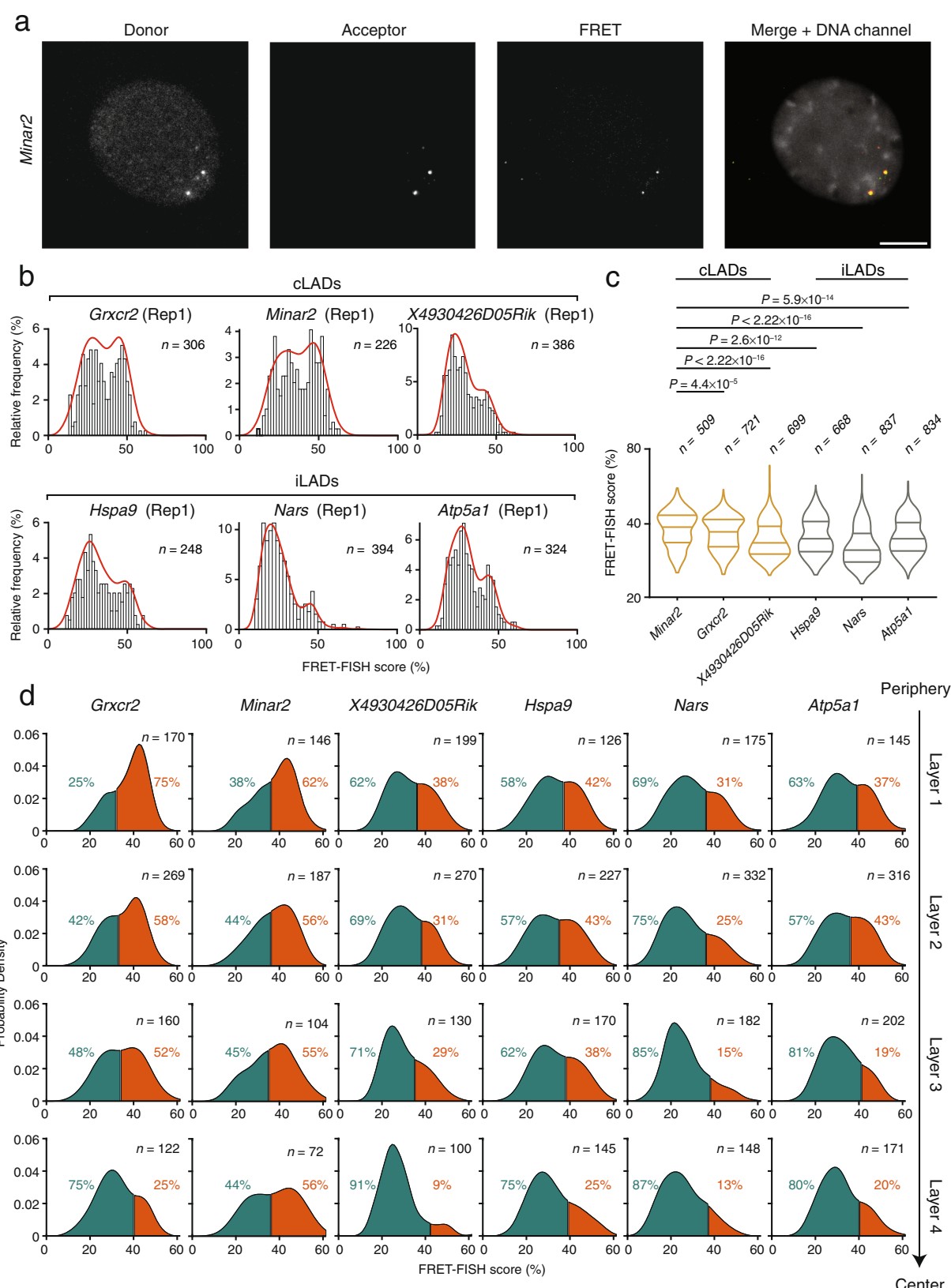

Hoechst 33342 staining (Supplementary Fig. 11a and Methods: 'Assignment of cells to different cell cycle phases'). In the case of *Magix*, we also managed to identify mitotic cells with detectable FRET signals. As expected, the FRET-FISH score was significantly higher ($P = 0.008$, Wilcoxon test, two-tailed) in mitotic cells compared to cells in G1 (Supplementary Fig. 11b). Notably, in two independent

experiments, the FRET-FISH score was substantially lower in non-G1 compared to G1 cells for all three genes tested, which was reflected in a decrease in the right mode of the bimodal FRET-FISH score distributions in non-G1 cells (Fig. 4a–f and Supplementary Fig. 11c–e). These results are consistent with prior observations based on the ATAC-see method[29] where G1 cells were found to display lower DNA accessibility

**Fig. 3 | Local chromatin compaction is influenced by radial position in the nucleus. a** Maximum z-projections exemplifying donor (Cy3 excitation, Cy3 emission), acceptor (Cy5 excitation, Cy5 emission) and FRET (Cy3 excitation, Cy5 emission) signals in one nucleus of female mouse embryonic fibroblasts (MEFs) hybridized with a FRET-FISH probe targeting the *Minar2* locus on chr18. Gray, DNA stained with Hoechst 33342. Scale bar, 10 μm. **b** FRET-FISH score distributions for six loci targeted on chr18 in female MEFs, in one of two replicate (Rep) experiments. Three genes are in constitutive lamina-associated domains (cLADs) and three inside inter-LAD regions (iLADs). Red lines, kernel density estimation function. *n*, number of FRET signals analyzed. **c** Distributions of the FRET-FISH scores for the three genes in cLADs and three genes in iLADs shown in (**b**), assessed by FRET-FISH in

female MEFs. Violins extend from minimum to maximum and horizontal lines represent (from top to bottom) the 75th percentile, the median, and the 25th percentile of each distribution. *n*, number of FRET signals analyzed. *P*, Wilcoxon test, two-tailed. **d** FRET-FISH score distributions in four concentric nuclear layers of equal volume, for each of the six genes shown in (**b**). The inflection point in each bimodal distribution was used to separate between the lower (green) and higher (orange) FRET-FISH score mode corresponding, respectively, to a less and more compact chromatin state. The percentages indicate the proportion of FRET signals in each group. *n* number of FRET signals analyzed. Source data for all the plots shown in the figure are provided as a separate Source Data file.

than G2 cells. We then further separated G1 cells into Hoechst^High (top quartile) and Hoechst^Low (all remaining G1 cells) and found that the FRET-FISH score calculated for the same three loci was significantly higher in Hoechst^High cells, which might represent cells that have just exited mitosis (Fig. 4a–c). To exclude that low DNA accessibility in highly compacted regions hinders the binding of the oligos in the FRET-FISH probes, we measured the intensity in the acceptor dye channel in G1 and M phase cells. The intensity was significantly higher for mitotic cells compared to cells in G1, strongly indicating that the oligo hybridization efficiency in FRET-FISH is not impaired even when targeting highly condensed chromatin (Supplementary Fig. 11f). These results demonstrate that the compaction state of a given locus is influenced by the cell cycle phase in addition to the radial position in the nucleus and highlight the ability of FRET-FISH in distinguishing different compaction states in different phases of the cell cycle.

**Local chromatin compaction changes upon serial cell passaging**
Lastly, we explored whether the FRET-FISH score distribution bimodality is also influenced by cellular senescence, which is associated with loss of constitutive heterochromatin as well as progressive silencing of euchromatin[30]. We hypothesized that, at increasing cell passages, the compaction of chromatin at active genes might gradually increase in primary cells undergoing senescence. To test this hypothesis, we monitored the FRET-FISH score for three of the genes targeted on chrX (*Magix*, *Kdm5c*, and *Atp2b3*) in female MEFs cultured up to the recommended number of passages (<5) or for longer periods (>10 passages). Cells cultured for more than 10 passages exhibited a significantly higher ($P = 1.8 \times 10^{-70}$, Wilcoxon test, two-tailed) global chromatin condensation, as assessed by the total nuclear intensity in the Hoechst 33342 channel (Supplementary Fig. 12a). Notably, the nuclear levels of two DNA damage markers, phosphorylated histone H2A.X (γH2A.X) and TP53 binding protein 1 (53BP1), also increased with the number of in vitro passages, suggesting that increased levels of DNA damage might be related with the observed increase in global chromatin condensation (Supplementary Fig. 12b, c and Methods: 'Immunofluorescence of DNA damage markers'). In two independent experiments, the distributions of the FRET-FISH scores for the three genes examined displayed a clear bimodality, independently of the passage number (Fig. 4g–i and Supplementary Fig. 12d–f). The right mode in the distributions—corresponding to a higher compaction state—progressively increased upon prolonged cell passaging, mirroring the global increase in chromatin condensation (Fig. 4g–i and Supplementary Fig. 12d–f). These results further highlight the reproducibility of our assay and demonstrate that cell ageing influences local chromatin compaction, contributing to the observed bimodality in the FRET-FISH score distributions.

## Discussion
We have developed a method—FRET-FISH—which combines the specificity of DNA FISH with the sensitivity of FRET to probe chromatin compaction at select genomic loci. FRET measurements are notoriously challenging in fixed cells and a combination of FRET and DNA FISH has not been described before. We demonstrate that the FRET

signal intensity can be modulated by rationally designing different types of FRET-FISH probes, with the highest FRET obtained with probes in which detection oligos binding to consecutive target-specific oligos carry extra 6 nt sequences which stabilize the proximity between the FRET donor and acceptor dyes coupled to them (see Design 3 in Fig. 1b). Although this probe design allowed us to monitor local chromatin compaction changes at various loci, we cannot exclude that further design optimization might result in even higher FRET signals and increased assay sensitivity. Of note, the current FRET-FISH probe design allows to measure chromatin compaction on the two homologue chromosomes for any locus of interest at the level of individual cells, allowing to detect inter-homologue differences.

DNA FISH is a relatively harsh procedure that requires cell fixation and denaturation of the DNA duplex for the FISH probes to hybridize to their targets. Despite the concern that the denaturation step in DNA FISH might disrupt the local chromatin structure, previous studies comparing classical DNA FISH on fixed cells with CRISPR/Cas9-assisted detection of DNA loci in living cells showed a very good concordance between the two approaches[31]. To minimize potential disruptions to the local chromatin structure and compaction, in FRET-FISH we use very mild denaturation conditions (typically, 2 min at 75 °C compared to 3–5 min at 80–90 °C used in published oligo-based DNA FISH protocols[32–37], see Supplementary Data 2) and always scout denaturation conditions aiming at the lowest denaturation temperature and time whenever working with a new cell line. Using these conditions, we have shown that the 3D distance between two loci on the same chromosome does not significantly change when a FRET-FISH probe is hybridized to the intervening region (Supplementary Fig. 2d, e). Furthermore, the high correlation between FRET-FISH and ATAC-seq further suggests that no major structural changes are introduced during the hybridization step in FRET-FISH.

Compared to the iFISH probes that we previously described[15], FRET-FISH probes contain more oligos (~200 for D and ~200 for A oligos vs. typically 96 oligos in iFISH) to maximize the likelihood that a sufficient number of FRET donor and acceptor dyes are in close proximity and thus can generate a detectable signal. Using probes containing <200 oligos per channel, FRET-FISH was still able to detect FRET at loci presumably in a more compact state, while it lost its sensitivity for more loosely packed loci. On the other hand, to probe chromatin compaction across larger regions, such as individual topologically associating domains (TADs)[38] or LADs[39], FRET-FISH probes containing more than 200 oligos should be used. FRET-FISH probes of any size can be easily designed using our freely available scripts (see Code availability) and produced from synthetic oligopools following the step-by-step protocol that we make available at Protocol Exchange[40].

Beyond studying local chromatin compaction, we envision that FRET-FISH could also be applied to study enhancer-promoter contacts or chromatin loop organization in single cells, without the need to rely on super-resolution microscopy techniques to bypass the inherent spatial resolution limitations of DNA FISH. However, this would require further optimization of the current FRET-FISH protocol to achieve high FRET detection sensitivity even with small numbers of oligos per probe, which is needed to tag short genomic regions such as

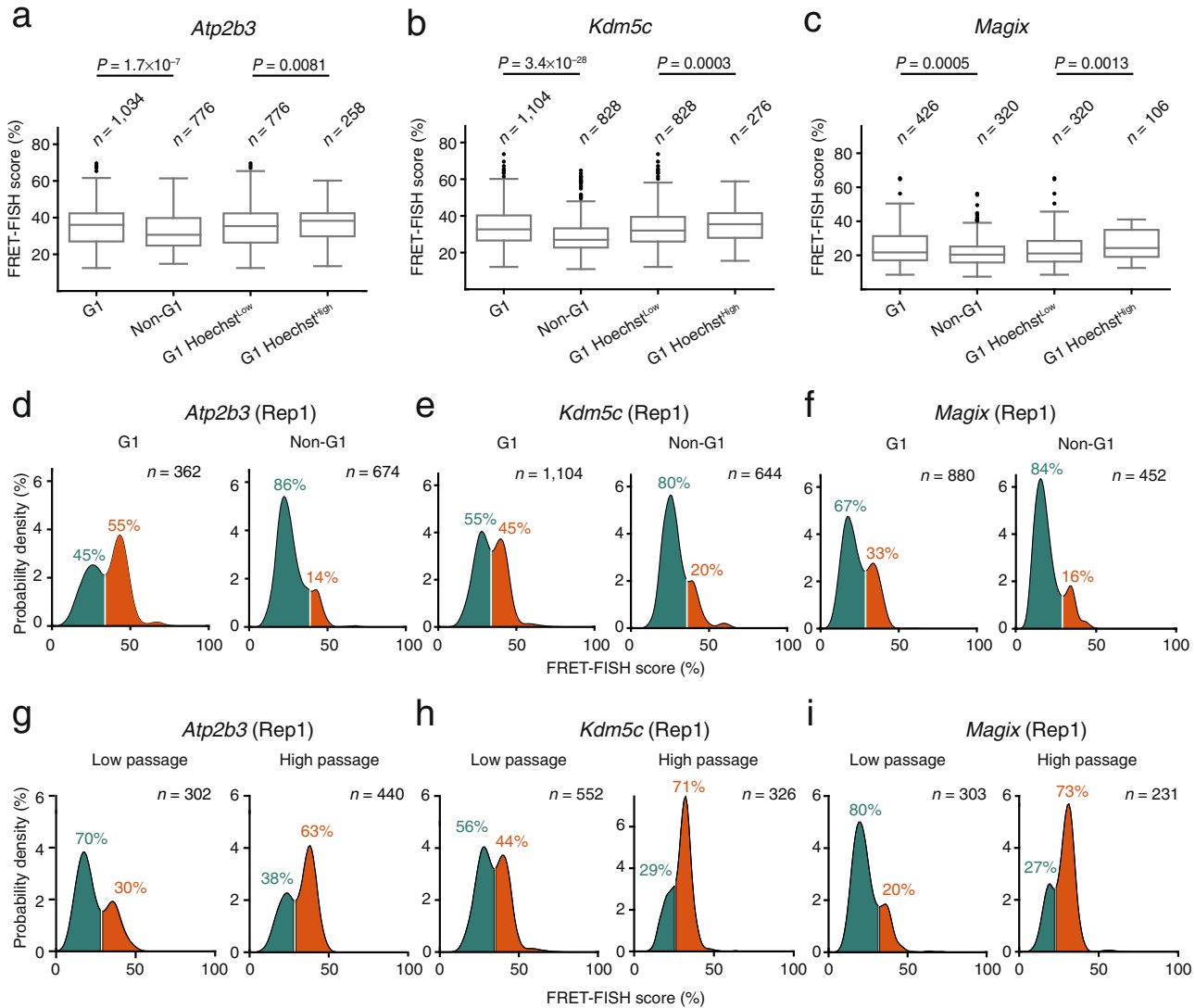

**Fig. 4 | Local chromatin compaction is influenced by the cell cycle phase and cell passaging. a–c** FRET-FISH score distributions in different cell cycle phases, for three of the six loci probed by FRET-FISH on chrX in female mouse embryonic fibroblasts (MEFs). Measurements from three replicate experiments pooled together are shown. G1 Hoechst^High and G1 Hoechst^Low indicate cells in G1 phase with values of Hoechst 33342 nuclear fluorescence intensity in the top and bottom quartile of the corresponding intensity distribution, respectively. Boxplots extend from the 25th to the 75th percentile, horizontal bars represent the median, and whiskers extend from −1.5×IQR to +1.5×IQR from the closest quartile, where IQR is the inter-quartile range. Black dots, outliers. In each boxplot, the minimum and maximum are defined, respectively, by the uppermost and lowermost outlier dot or extremity of the corresponding whisker. *n*, number of FRET signals analyzed. *P*, Wilcoxon test, two-tailed. **d–f** FRET-FISH score distributions in G1 and non-G1 cells for the same loci shown in (**a**), in one of three replicate (Rep) experiments. The inflection point in each bimodal distribution was used to separate between the lower (green) and higher (orange) FRET-FISH score mode corresponding, respectively, to a less and more compact chromatin state. The percentages indicate the proportion of FRET signals in each group. *n*, number of FRET signals analyzed. **g–i** Same as in (**d–f**) but comparing MEFs cultured for less (Low passage) or more (High passage) than 10 passages. Source data for all the plots shown in the figure are provided as a separate Source Data file.

enhancers and promoters. The FRET-FISH probe design described here could also be adapted to detect DNA-RNA and RNA-RNA interactions in single cells. Lastly, FRET-FISH probes carrying FRET donor dyes could be combined with antibodies labeled with FRET acceptor dyes to detect DNA-protein or RNA-protein interactions, instead of using proximity-ligation assays.

For all the loci that we assessed by FRET-FISH, we obtained clearly bimodal FRET-FISH score distributions, strongly suggesting that these loci can be found in two distinct compaction states. At the same time, each of the loci examined was associated with a characteristic FRET-FISH score distribution, which was highly reproducible between replicate experiments. Even though it was technically challenging to thoroughly compare FRET-FISH scores between active and inactive loci on chrX, our data using probes targeting multiple loci on chr18

clearly demonstrate that different compaction states characterize not only loci on chrX but also loci on autosomal chromosomes. Furthermore, we have shown that the FRET-FISH score distribution bimodality is strongly influenced by the radial location of a locus as well as by the cell cycle phase. Therefore, we hypothesize that the bimodality in the FRET-FISH score distributions represents different states of local compaction, which is influenced by both cell-intrinsic (nuclear position, cell cycle phase, passage number) and cell-extrinsic (external stimuli affecting global chromatin condensation) factors. As more factors influencing local compaction likely exist, we believe that FRET-FISH represents a powerful tool that can be harnessed to investigate this important feature of chromatin at the nanoscale.

In conclusion, we have developed a versatile assay that expands the existing toolkit for studying the spatial organization of the

genome in the cell nucleus. In the future, FRET-FISH might be harnessed not only to probe chromatin compaction at defined genomic regions—which was thus far not possible—but also to study important aspects of structural genome organization such as promoter-enhancer contacts or the formation of condensates. We therefore anticipate that FRET-FISH will be broadly adopted in the genome organization research field.

## Methods

### Experimental methods

A step-by-step FRET-FISH protocol is available at Protocol Exchange[40].

### Cell lines

We obtained mouse embryonic fibroblasts (MEFs) from ATCC (cat. no. SCRC-1040), NIH3T3 fibroblasts from ATCC (cat. no. CRL-1658) and HAP1 chronic myeloid leukemia cells from Horizon Discovery (cat. no. C859). We grew MEFs in Dulbecco's Modified Eagle's Medium (DMEM) (Sigma, cat. no. D6429) supplemented with 15% fetal bovine serum (FBS) (Sigma, cat. no. F9665), NIH3T3 cells in DMEM supplemented with 10% FBS, and HAP1 cells in Iscove's Modified Dulbecco's Medium (Sigma, cat. no. I6529) supplemented with 10% FBS. None of these cell lines is included in the ICLAC database of commonly misidentified cell lines. We regularly checked the cells for Mycoplasma contamination but did not authenticate them. All cell lines tested negative for Mycoplasma contamination. We incubated the cells at 37 °C in 5% $O_2$ and 5% $CO_2$. We grew the cells until they reached 80% confluency on coverslips (VWR, cat. no. 630-2185) in the case of MEFs and on 9-well chambered coverslips (custom-made by Grace Bio-Labs) in the case of HAP1 cells.

### FRET-FISH probe design

We designed FRET-FISH probes against several genes located along mouse chrX, reasoning that this would allow us to detect differences in chromatin compaction between the active (Xa) and inactive (Xi) homologue in female cells. To select FRET-FISH probe target gene candidates, we used publicly available allelic-specific RNA-seq from GEO accession: GSE75659 and ATAC-seq data from GEO accession: GSE71156. We used a custom script in MATLAB (R2020a), selection_gene_ATACseq_norm.m, to read the file with normalized ATAC-seq reads assigned to Xa or Xi based on their unique single nucleotide polymorphism (SNP) signature. The probes span regions close to 100 kb, therefore we searched for 110 kb windows that displayed the largest differences in ATAC-seq read counts between Xa and Xi. The most predominant gene or genes present in the selected windows were compared with the RNA-seq results where high expression levels from Xi means that the gene is an escapee. We aimed at having a balance in between both escapees and inactivated genes.

To design the probes, we created a custom pipeline in MATLAB, that is available at https://doi.org/10.5281/zenodo.7125033[41] and generates a list of oligonucleotides (oligos) that attain a strict score of uniqueness. The user specifies the oligo length (l), the number of total oligos (n), the spacing (s) in between consecutive oligomers, the genomic coordinates of the target region and the reference genome against which the oligos must be designed. The genomic sequence of the specified target region is then downloaded from the USCS Genome Browser. In the first MATLAB script, generate_oligos.m, all possible oligo sequences are generated from the target sequence that the user inputs, by moving a sliding window of length l in steps of 1 nt all along the target sequence. Oligos with more than 6 repetitive nucleotides or a GC-content off the 35–80% range are filtered out from the output list. Secondly, a file named Oligos_list.fa containing the list of oligos remaining after the previous filtering step as well as a variable file named Identities.mat containing all the penalties calculated in the filtering step are generated. The oligos in the Oligos_list.fa list are then run locally through BLASTn (version 2.6.0+), by setting a threshold of

80% homology and 80% mismatches using the following command in Unix:

```
blastn -query Oligos_list.fa -db whole_genome.fa -out
homologies.txt -evalue 10 -word_size 11 -gapopen 15
-gapextend 10 -penalty −3
```

The parameters of BLASTn are less stringent than in the iFISH pipeline because FRET-FISH probes require a larger number of oligos (typically 450 oligos per probe in FRET-FISH vs. 96 in iFISH) for FRET to be detected. The number of oligos and length of each probe can be found in Supplementary Data 1. The output file is considerably heavy for posterior analysis, so we suggest removing all the non-relevant information and only keeping the oligo name and score using the following command:

```
awk '$1 ~ /^Query=/ {print $2} /^ Score/ {print $3}'
<homologies.txt > homologies.txt.filt
```

This reduced file is then read line by line by the second MATLAB script Identities_penalty.m and all the homologies found for each oligo are summed up and saved in a MATLAB vector named Scores, which also contains the penalties before BLAST. Subsequently, the calculated homology values are added to the file Identities.mat to complete the penalty description for each oligo. The third script, select_best_window.m, searches the best window according to the following parameters specified by the user: (1) space (in nt) in between consecutive oligos carrying the same FRET dye; (2) space in between consecutive oligos carrying different FRET dyes; (3) number of oligos in a group of consecutive oligos carrying the same FRET dye; and (4) total number of oligos in the FRET-FISH probe to be designed. The recommended window is computed based on the penalty score of each oligo and the distance distribution in between the oligos. Other parameters can be fine-tuned such as the tolerance for some oligo groups to be incomplete, and slight changes can be made in the distance and homology threshold used to select oligos. The generated oligos are flagged as 'good' when the oligos are within all the thresholds set, whereas they are flagged as 'bad oligos' when they are marked by multiple penalties. An additional filter step is the identification of good oligos whose neighbors are classified as bad and then removed. Lastly, left (L) and right (R) adapter sequences are appended to the 5′ and 3′ end, respectively, of each target (T) sequence (see oligo schemes in Fig. 1b) and all the oligo sequences are printed in a text file named result.txt with design parameters information and various summary plots saved in the same folder. For FRET-FISH probes targeting mouse or human genomic regions, 20 nt orthogonal sequences suitable to be used as F and R adapters can be found in the list, which we previously described for iFISH (see Supplementary Data 8 in ref. 15). For FRET-FISH probe design 2 and 3 (Fig. 1b), we added complementary stabilization sequences (SS) to the 3′ (AATTA) and 5′ (TAATT) end of consecutive oligos. We chose sequences composed only of As and Ts since these bases are known to affect fluorescence decay at lower scale compared to Gs and Cs. We purchased all the fluorescently labeled detection oligos (L* and R* in Fig. 1b) from Integrated DNA Technologies (IDT), whereas we produced all the primary oligos composing each FRET-FISH probe starting from synthetic oligopools as described below.

### FRET-FISH sample preparation

We performed the experiments on mouse embryonic fibroblasts (MEFs) from ATCC (cat. no. SCRC-1040), NIH3T3 mouse fibroblasts from ATCC (cat. no. CRL-1658) and HAP1 chronic myeloid leukemia cells from Horizon Discovery (cat. no. C859). We grew the MEF cells in Dulbecco's Modified Eagle's Medium (DMEM, Sigma) supplemented with 15% FBS (Sigma), NIH3T3 cells in DMEM supplemented with 10% FBS and HAP1 cells in Iscove's Modified Dulbecco's Medium (IMDM,

Sigma) supplemented with 10% FBS. None of these cell lines is included in the ICLAC database of commonly misidentified cell lines. We regularly checked the cells for Mycoplasma contamination but did not authenticate them. We incubated the cells at 37 °C in 5% O₂ and 5% CO₂. We grew cells either on regular #1.5 coverslips (22 × 22 mm, VWR) or on custom-designed 9-well silicone-chambered #1.5 coverslips (custom-made by Grace Bio-Labs) and we processed them following an adapted version of the 3D-FISH protocol described before[24]. Unless otherwise specified, we performed all the incubations at room temperature using solutions either stored or brought to the same temperature. Briefly, we fixed the cells in 4% formaldehyde (EMS, cat.no. 15710)/1× PBS (Thermo Fisher Scientific, cat. no. 003002) for 10 min, followed by quenching of the unreacted formaldehyde in 125 mM glycine (Sigma-Aldrich, cat. no. 50046-250 G)/1x PBS for 5 min. Subsequently, we washed the coverslips three times, 5 min each with 0.05% Triton X-100 (Promega, cat.no. H5142)/1× PBS and permeabilized the cells in 0.5% Triton X-100/1× PBS for 20 min, then washed twice for 5 min with 0.05% Triton X-100/1× PBS. Afterwards, we incubated the coverslips in 0.1 N HCl for 5 min and quickly rinsed them twice with 0.05% Triton X-100/1× PBS. Lastly, we rinsed the coverslips in 2× SSC buffer (Thermo Fisher Scientific, cat. no. AM9763) and incubated them overnight in 50% formamide (Thermo Fisher Scientific, cat. no. AM9344)/50 mM sodium phosphate (Sigma Aldrich, cat.no. S5136-500G)/2× SSC. The following day, we transferred the coverslips to +4 °C and kept them for one week in 50% formamide/50 mM sodium phosphate/2× SSC. Lastly, we exchanged the buffer to 2× SSC and proceeded to hybridization. If we could not proceed to hybridization immediately, we stored the samples in 2× SSC at +4 °C for up to 2 weeks.

## FRET-FISH probe production and hybridization

We produced all the FRET-FISH probes starting from synthetic 12 K oligopools, using the iFISH pipeline[15]. For FRET-FISH, we prepared samples and hybridized them with FRET-FISH probes using a modified 3D DNA FISH protocol[42]. Since the FRET signal can be affected by multiple effects including bleed-through and crosstalk, to minimize inter-sample variability we performed all our FRET-FISH experiments using custom-made multi-chambered coverslips (see Supplementary Fig. 1a, c) in which control and test samples were processed and imaged together. Unless otherwise specified, we performed all the incubations at room temperature using solutions either stored or brought to the same temperature. We first immersed the coverslips in a pre-hybridization buffer (PHB) containing 5× Denhardt's solution (Thermo Fisher Scientific, cat. no. 750018)/50 mM sodium phosphate buffer (Sigma-Aldrich, cat. no. D8662-500ML)/1 mM EDTA (Thermo Fisher Scientific, cat. no. AM9261)/100 ng/μL ssDNA (Thermo Fisher Scientific, cat. no. 15632011)/50% formamide/2x SSC, pH 7.5–8.0, and incubated them for 1 h at 37 °C in a humidity chamber. During this time, we prepared the first hybridization mix (HM-1) by mixing each probe at 1:9 vol/vol ratio with 1.1× first hybridization buffer (HB-1) containing 5.5× Denhardt's solution/55 mM sodium phosphate buffer/1.1 mM EDTA/111 ng/μL ssDNA/55% formamide/11% dextran sulfate (Sigma-Aldrich, cat. no. D8906-10G)/2.2× SSC pH 7.5–8.0 (in this mix, the final concentration of each oligo is 0.05 nM). We then removed the coverslips from PHB and placed it on top of 30 μL of HM-1 deposited onto a microscope slide. We sealed the coverslips with fixogum (Leica, cat. no. LK071A) and waited until the fixogum solidifies. Next, we performed DNA denaturation by placing the coverslips for 2 min 30 s at 75 °C on a heating block and incubated the samples for 15–18 h at 37 °C. The next day, we washed the coverslips twice, 5 min each at 65 °C in 0.2% Tween (Promega, cat. no. H5152)/0.2× SSC pre-warmed to 65 °C inside a water bath, followed by a brief wash in 0.2% Tween/4× SSC at room temperature, a rinse in 2× SSC, and then exchanged the solution to 25% formamide/2× SSC. Next, we immersed the coverslips in 300 μL of the second hybridization mix (HM-2) containing the six secondary fluorescently labeled oligonucleotides (one per color), each

at a final concentration of 20 nM in HB-2 containing 25% formamide/10% dextran sulfate/1 mg/mL *E. coli* tRNA (Sigma-Aldrich, cat. no. R1753)/0.02% bovine serum albumin (Thermo Fisher Scientific, cat. no. AM2616)/2× SSC and incubated them for 24 h at 30 °C. Afterwards, we washed the coverslips for 1 h at 30 °C in 25% formamide/2× SSC, followed by 30 min at 30 °C in 1.23 ng/mL Hoechst 33342 (Thermo Fisher Scientific, cat no. H1399) in 25% formamide/2x SSC. Lastly, we briefly rinsed the coverslips twice in 2x SSC, before mounting them in GLOX buffer containing 2× SSC/10 mM Tris-HCl pH 7.5 (Thermo Fisher Scientific, cat. no. 15567-027)/0.4% Glucose (Thermo Fisher Scientific, cat. no. 15023021)/10 mM TROLOX (Sigma-Aldrich, cat. no. 238813)/37 ng/μL Glucose Oxidase (Sigma-Aldrich, cat. no. G2133-10KU)/32 mM Catalase (Sigma-Aldrich, cat. no. 3515-10MG).

## FRET-FISH combined with single-molecule FISH for *Xist*

We processed the samples as described above, with the exception that we added RNase inhibitors to all the buffers to preserve RNA. Briefly, we heated all the buffers except for HCl at 60 °C for 10 min after adding Ribonucleoside Vanadyl Complex (RVC, New England Biolabs, cat. no. S1402S) at a final concentration of 10 mM. We prepared HM-1 buffer by mixing each FRET-FISH probe at 1:9 vol./vol. ratio with 1.1× HB-1 with 1 U/μL Protector (Roche, cat. no. 3335399001) and then performed DNA denaturation for 2 min at 75 °C on a heating block. Afterwards, we incubated the coverslips for 15–18 h at 37 °C. The next day, we washed the coverslips twice 5 min each at 65 °C in 0.2× SSC/0.2% Tween/10 mM RVC pre-warmed at 65 °C inside a water bath, followed by a brief wash in 4× SSC/0.2% Tween/10 mM RVC at room temperature, a brief wash in 2× SSC/10 mM RVC at room temperature, and one final short wash in 2× SSC/25% formamide/10 mM RVC at room temperature. Next, we prepared the *Xist* RNA FISH probe (see Supplementary Data 1) by diluting it to a final concentration of 5 ng/μL in HB-2 buffer with 1 U/μL Protector and added the probe to each sample, followed by incubation for 24 h at 30 °C. The next day, we washed the coverslips for 1 h at 30 °C in 2× SSC/25% formamide and added the third hybridization mix containing the secondary fluorescently labeled oligonucleotides at a final concentration of 20 nM in HB-2 buffer and incubated the samples for 3 h at 30 °C. We washed the coverslips in 2× SSC/25% formamide for 1 h at 30 °C and then placed them in 2× SSC/25% formamide/1.23 ng/mL Hoechst 33342 for 30 min at 30 °C. We briefly rinsed the coverslips twice in 2× SSC before mounting them in GLOX buffer and imaging the samples as described above.

## FLIM-FRET

We performed all FLIM experiments at the Intravital Microscopy Facility at Stockholm University and National Microscopy Infrastructure in Stockholm. We used a Leica SP8 inverted confocal laser scanning microscope with an integrated FALCON module (Leica Microsystems) equipped with a pulsed White Light Laser (78 MHz), a UV laser diode (405 nm), an acousto-optical tunable beam-splitter and spectral detection via GAsP hybrid detectors. We performed donor excitation at 499 nm and collected the emitted fluorescence at 505–558 nm. We acquired all the images using an HC PL APO 63×/1.4 Oil CS2 objective lens and a camera with 256 × 256 pixels resolution and a scan speed of 400 Hz with a maximum accumulated photon count of 1000 photons per pixel. We performed all image acquisition steps and FLIM data analyses using the FLIM FCS module in the LAS X software (Leica microsystems).

## Testing whether FRET-FISH probes disrupt the target locus conformation

We co-hybridized each of the six different FRET-FISH probes designed to target the mouse *Ogt* gene (I1S50, I1S150, I12S50, I2S300, I4S50, I4S300) with two probes flanking the *Ogt* gene in female MEFs (see Supplementary Data 1 for the list of oligos in each probe and Supplementary Fig. 2a for a scheme of the oligo distribution and span of

different *Ogt* probe designs). For the *Ogt* probes, we hybridized only the primary oligos in each probe, i.e., we did not add their detection oligos to the second hybridization mix, in which we only included fluorescently labeled oligos detecting the flanking probes. As control, we hybridized the two probes flanking the *Ogt* locus without any FRET-FISH probe. We imaged all the samples together with fluorescent beads sample, which we then used to correct the shift on the images. We used our in-house image analysis suite DOTTER to detect FISH dots and assign them to individually segmented nuclei. We used a custom script to calculate the Euclidean distance (in 3D) between any two dots corresponding to the two *Ogt* flanking probes using the following equation:

$$\text{Distance} = \sqrt{(x_1 - x_2)^2 + (y_1 - y_2)^2 + (z_1 - z_2)^2} \tag{1}$$

### Induction of chromatin condensation by ATP depletion
To induce chromatin condensation by ATP depletion, we treated female mouse embryonic fibroblasts (MEFs) by adding sodium azide (Sigma, cat. no. S2002-5G) and 2-Deoxyglucose (Thermo Fisher Scientific, cat. no. 10560371) directly to the culture medium at a final concentration of 10 mM and 50 mM, respectively, and incubating for 1.5 h at 37 °C before processing the cells for FRET-FISH. To confirm the effect of the drug combination, we measured the total amount of nascent RNA in the nucleus with the Click-iT RNA Imaging Kit (Thermo Fisher Scientific, cat. no. C10330) following the manufacturer's instructions. Briefly, we plated MEFs at 40–50% confluency on 22 × 22 mm coverslips (VWR, cat. no. 630-2185) placed in a 6-well plate (1 coverslip per well). The next day, we added 5-ethynyl uridine (EU) to the medium at a final concentration of 1 mM and incubated for 1 hour in the cell incubator at 37 °C. Afterwards, we aspirated the medium in each well, replaced it with 1 mL per well of 1× PBS (Thermo Fisher Scientific, cat. no. AM9625)/3.7% formaldehyde (EMS, cat. no. 15710) and incubated for 15 min at room temperature. We then washed the cells once with 1× PBS at room temperature, followed by permeabilization of the cells with 1× PBS/0.5% Triton X-100 (Sigma, cat. no. T8787) for 15 min at room temperature and another wash in 1× PBS. We aspirated the wash solution and added 500 μL per well of freshly prepared Click-iT reaction cocktail (a mixture containing Click-iT RNA reaction buffer, CuSO4, Alexa Fluor azide, and Click-iT reaction buffer additive) and incubated for 30 min at room temperature in darkness. Lastly, we washed the cells once with 1 mL per well of Click-iT reaction rinse buffer at room temperature and stained DNA by incubating the cells in 1× PBS/1.23 ng/mL Hoechst 33342 (Thermo Fisher Scientific, cat. no. 62249) at 30 °C for 15 min. Finally, we rinsed the coverslips twice in 1x PBS at room temperature before mounting them with GLOX solution containing 2× SSC/10 mM TROLOX (Sigma, cat. no. 238813)/37 ng/μL Glucose Oxidase (Sigma, cat. no. G2133)/32 mM Catalase (Sigma, cat. no. C3515) for imaging. We imaged the samples using the same microscope system and settings as described for FRET-FISH above. To quantify the fluorescence intensity per nucleus, corresponding to the nascent transcripts visualized with the Click-iT RNA Imaging Kit, we passed the nd2 files generated by our microscope to the Fiji script *fromND2_toINTENSITIES.ijm*, which performs automatic nuclear segmentation and returns the average fluorescence intensity per nucleus in the selected channels.

### Immunofluorescence of DNA damage markers
We grew MEFs on coverslips and then fixed and permeabilized them following the same procedure for FRET-FISH described above. We incubated the coverslips in a blocking solution containing 1× PBS/5% Bovine Serum Albumin (BSA)/0.1% Tween for 1 h at room temperature. We then flipped each coverslip onto 150 μL of a primary antibody solution containing mouse anti-phospho-Histone H2A.X (Ser139, Merck Millipore, cat. no. 05-636) or rabbit anti-53BP1

(Novus, cat. no. NB100-304) diluted 1:1000 vol./vol. in 1× PBS/5% BSA placed on a piece of Parafilm, and incubated the samples overnight at 4 °C. The next day, we washed the cells three times 1× PBS/0.1% Tween for 10 min at room temperature shaking. We then flipped each coverslip onto 150 μL of a secondary antibody solution containing Alexa Fluor 555 Donkey Anti-Mouse IgG (H+L) (Thermo Fisher Scientific, cat. no. A31570) (for anti-phospho-Histone H2A.X) or Alexa Fluor Plus 488 Donkey anti-Rabbit (Thermo Fisher Scientific, cat. no. A32790) (for anti-53BP1) diluted 1:500 vol./vol. in 1x PBS/5% BSA placed on a piece of Parafilm and incubated the samples for 1 h at room temperature. We washed the cells three times in 1× PBS/0.1% Tween, each 10 min shaking. Lastly, we incubated the samples with 1× PBS/1 ng/μL Hoechst 33342 for 5 min at room temperature before mounting them in GLOX buffer and imaging them as described for FRET-FISH.

### Analytical methods
**Image processing and identification of FRET-FISH dots.** We imaged all the samples using a 100 × 1.45 NA objective mounted on a Nikon Eclipse Ti-E inverted microscope system controlled by the NIS Elements software (Nikon) and equipped with an iXON Ultra 888 EMCCD camera (Andor Technology). For each sample, we acquired multiple image stacks, each consisting of 49–70 focal planes spaced 0.3 μm apart. A list of filters and dichroic mirrors used in this study is available in Supplementary Table 2. To identify FISH dots, we used our in-house image analysis suite DOTTER (v0.0.1) written in MATLAB (MATLAB and Statistics Toolbox Release R2020a) and C99 with GSL (https://doi.org/10.5281/zenodo.7112586[43]). In DOTTER, cell segmentation is done by thresholding the max intensity projection (in the axial direction). Using watersheds for nuclei separation, the user can then adjust the threshold, in a semi-automatic manner, and tune low- and high-pass filters. To process the FRET-FISH images, we developed a completely automated pipeline in MATLAB, which detects the most intensive signals in each nucleus after segmentation. We acquired images in the donor and acceptor fluorescent channel separately, and only considered FISH dots with similar (*x, y, z*) coordinates (radial threshold: 7 pixels or 1.9 μm) between the donor and acceptor channels. To measure FRET, we excited the donor dye and measured fluorescence in the acceptor channel at the location of each acceptor dot.

### FRET-FISH score calculation
In this study we used an emission sensitized FRET assay, so we expected that the FRET intensity would increase when the donor and acceptor dyes are closer in space. Simultaneously, donor fluorescence emission is transferred to the acceptor dye which results in a decrease of donor intensity. Therefore, we calculated the FRET-FISH score as following:

$$\text{Score} = \frac{I_{\text{FRET}}}{I_{\text{FRET}} + I_D} \tag{2}$$

where $I_{\text{FRET}}$ is the fluorescence intensity measured in the acceptor channel upon donor excitation, whereas $I_D$ is the fluorescence intensity measured in the donor channel upon donor excitation.

### Comparison between FRET-FISH and ATAC-seq
To compare FRET-FISH measurements with ATAC-seq data, we downloaded available ATAC-seq data from MEF cells from the Gene Expression Omnibus (GEO) database (accession number "GSE127926"). We extracted the number of ATAC-seq reads within a genomic window with the same start and end coordinates as the ones of the corresponding FRET-FISH probe. We then compared the ATAC-seq read counts per gene with the mean of FRET efficiency calculated for the same gene.

## Analysis of chromatin compaction along the nuclear radius

For each FISH dot, we calculated its normalized 3D radial distance to the nuclear edge using the *Lamina_distance_boxplot.m* script available at https://doi.org/10.5281/zenodo.7125033[41], after segmenting nuclei in 3D as following: we first deconvolved the DNA staining channel using the Huygens Professional Software (Scientific Volume Imaging, v17.04) with the following parameters: CMLE algorithm, null background, signal-to-noise ratio equal to 7, and 50 iterations. After deconvolution, we performed 3D segmentation of the nuclei in each field of view, using the *tiff_auto3dseg* script in the pygpseq Python3 package, which we previously described[44] (https://github.com/ggirelli/pygpseq/). We then divided the distributions of normalized lamina distances into four quantiles (0–25%, 26–50%, 51–75%, and 76–100%) representing four imaginary concentric nuclear layers and assigned each FRET-FISH dot to one of the four quartiles based on the corresponding distance to the lamina (the first and fourth quartiles represent the outermost and innermost concentric nuclear layers, respectively).

## Assignment of cells to different cell cycle phases

We classified cells as G1 or non-G1 based on the distribution of the fluorescence intensity of the DNA staining dye, Hoechst 33342, in cell nuclei segmented in 2D by our in-house image analysis suite DOTTER (see paragraph 'Image processing and identification of FRET-FISH dots' above). The distribution of nuclear Hoechst intensity typically shows two clearly distinct peaks, one corresponding to cells in the G1 phase and the other corresponding to cells in G2/M. DOTTER automatically assigns the segmented nuclei to two groups (G1 and non-G1). To identify cells in mitosis, we visually inspected all the FRET-FISH images that we collected, searching for cells with highly condensed chromatin and the chromosome bouquet characteristic of cells in mitosis.

## FLIM-FRET data analysis

We acquired FLIM data after several repetitions of excitation and lifetime decay detection for each AF488 signal until we reached the intensity threshold (mean number of photons per pixel detected in the selected field) defined as 1000 photons. We fitted the lifetime values with a mono-exponential tail fit:

$$y(t) = \sum_{i=0}^{n-1} A[i] e^{\left(\frac{t-t_0}{\tau[i]}\right)} + Bkgr \tag{3}$$

$$I_{Sum} = \sum_{k=0}^{n-1} I[k] \tag{4}$$

$$A_{Sum} = \sum_{k=0}^{n-1} A[k] \tag{5}$$

$$\tau_{AvInt} = \frac{\sum_{k=0}^{n-1} I[k] \tau[k]}{I_{Sum}} \tag{6}$$

$$\tau_{AvAmp} = \frac{\sum_{k=0}^{n-1} A[k] \tau[k]}{A_{Sum}} \tag{7}$$

where $n$ is the number of exponential components; $A$ the amplitude (i.e., the exponential pre-factors); $t_0$ the lifetime offset (i.e., the extrapolated reference point for scaling the exponential pre-factors); $\tau$ the exponential decay times (i.e., lifetimes); $Bkgr$ the tail offset (i.e., the correction for background including after-pulsing, dark counts and environment light); $I$ the intensity associated with each exponential component, normalized to the time resolution of the measured decay curve in order to be displayed in photon counts; $I_{Sum}$ the sum of fluorescence intensity for all components; $A_{Sum}$ the sum of fluorescence intensity for all components at time zero; $\tau_{Av Int}$ the mean photon arrival time (i.e., the intensity-weighted average lifetime); and $\tau_{Av Amp}$ the mean decay time (i.e., the amplitude-weighted average lifetime). We measured a lifetime of $3.057 \pm 0.041$ ns (mean ± s.d., number of donor signals: 10) for AF488 dye without AF594, which is lower than the lifetime previously reported in vitro for Alexa dyes (4 ns)[45] most likely because of the presence of DNA and other quenching factors in situ[46]. We calculated the FRET efficiency, $E$ by model fitting of a mono-exponential donor as following:

$$E = \left(1 - \frac{\tau_{DA}}{\tau_D}\right) \tag{8}$$

$$y(t) = \left\{ IRF\left(t + Shift_{IRF}\right) + Bkgr_{IRF} \right\} \otimes \left\{ A_D\, e^{\left(\frac{t}{\tau_D}\right)} + A_{DA}\, e^{\left(\frac{t}{\tau_{DA}}\right)} + Bkgr \right\} \tag{9}$$

where $\tau_{DA}$ is the quenched donor lifetime; $\tau_D$ is the unquenched donor lifetime; $Shift_{IRF}$ is the correction for IRF displacement (i.e., the wavelength dependent zero time of the detector); $Bkgr_{IRF}$ is the IRF shift (i.e., the correction for IRF background); $A_D$ is the fraction of unbound (unquenched) donor molecules; $A_{DA}$ is the fraction of bound (quenched) donor molecules; and $Bkgr$ is the tail offset (i.e., the correction for background including after-pulsing, dark counts and environment light). We entered the parameters for unquenched donor lifetime and the Förster distance (6 nm for AF488 and AF594) in the FRET separator module of the LAS X software. Finally, we stored FRET efficiency values in a table for subsequent analysis.

## FRET-FISH score calculation for loci on the active and inactive chrX

We processed the images and picked FISH dots in the acceptor, donor, and FRET channel using DOTTER, as described above. We then visually inspected each field of view and manually annotated the dots falling into the *Xist*-positive chrX territory. Since the FRET-FISH score distributions obtained in this manner did not display the bimodality that we reproducibly observed when we probed the same genes probed with FRET-FISH alone (i.e., without combining it with *Xist* smFISH) and since the FRET intensities measures were unusually low, we used our Deconwolf software[25] (with default parameters) to deconvolve the images and enhance the contrast. We then repeated the dot picking procedure in DOTTER and analyzed the data as described above for non-deconvolved images.

## Statistics and reproducibility

We did not perform any a priori sample size calculation. In each experiment, we aimed at imaging at least 1000 cells, which we empirically found to be sufficient to obtain reproducible FRET-FISH score distributions. We excluded from downstream analyses FRET-FISH signal pairs (fluorescence dots in the donor and acceptor channel) detected in the same nucleus that were more than 7 pixels or 1.9 μm apart in 3D. As this study did not involve the treatment of human subjects or laboratory animals, we did not apply any randomization procedure. We did not apply blinding since all the image analyses were performed in an unsupervised manner (automatic FISH dot picking).

## Reporting summary

Further information on research design is available in the Nature Research Reporting Summary linked to this article.

## Data availability

The raw FRET-FISH images generated in this study have been deposited in Figshare under accession code 17080892. A description of all the datasets is available in Supplementary Data 3. The MEFs ATAC-seq data used to validate FRET-FISH in this study are available in GEO under accession code GSE127926. The source data for all the plots displayed in the main and supplementary figures are provided with this paper as a single.zip file. Source data are provided with this paper.

## Code availability

The custom MATLAB scripts used to design the FRET-FISH probes and analyze the FRET-FISH data described in this study are available at https://doi.org/10.5281/zenodo.7125033[41]. The DOTTER suite that we used to automatically pick up FRET-FISH dots is available at https://doi.org/10.5281/zenodo.7112586[43]. However, we regret not being able to assist with its use nor distribute updates to users outside of our laboratory since this package was developed for internal use only. The Deconwolf deconvolution software is available at https://github.com/elgw/deconwolf.

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

## Acknowledgements

We thank Sebastian Deindl and Anton Sabantsev (Deindl lab) for initial help with FRET-FISH implementation and Björn Reinius for discussions and suggestions on the design of FRET-FISH probes for chrX genes. We thank Britta Bouwman and Quentin Verron (Bienko-Crosetto Lab) for critically reading the manuscript and helping with editorial revisions. We acknowledge the Intravital Microscopy Facility at Stockholm University (IVMSU) and the National Microscopy Infrastructure, NMI (VR-RFI 2019-00217) for assisting with FLIM microscopy. This work was supported by grants from the Swedish Research Council (grant no. 2018-02950) and from the Swedish Cancer Society (grant no. CAN 2018/728) to N.C.; and by grants from the Science for Life Laboratory, the Karolinska Institutet KID Funding Program, the Swedish Research Council (grant. no. 621-2014-5503), the Human Frontier Science Program (grant. no. CDA-00033/2016-C), the Ragnar Söderberg Foundation (Fellows in Medicine 2016), the Swedish Cancer Society (grant no. 19 0130 Pj 03 H), and the European Research Council under the European Union's Horizon 2020 research and innovation programme (grant no. StG-2016_GENO-MIS_715727) to M.B.

## Author contributions

Conceptualization: A.M. Data curation: A.M., C.P., and S.B. Formal analysis: A.M., E.W., L.H., S.B., and N.C. Funding acquisition: N.C and M.B. Investigation: A.M., S.B., X.W., and K.G. Validation: A.M. and S.B. Methodology: A.M., C.P., and M.B. Project administration: A.M. and M.B. Software: A.M., E.W. and L.H. Supervision: M.B. and N.C. Visualization: A.M. and M.B. Writing: N.C., M.B., and A.M.

## Funding

## Competing interests

The authors declare no competing interests.
