## [Peer Review File · Nature Communications]

REVIEWER COMMENTS

Reviewer #1 (Remarks to the Author):

The authors describe a novel microscopy-based approach named “FRET-FISH” combining Förster Resonance Energy Transfer (FRET) measurements with Fluorescence in situ Hybridization (FISH) to analyse the chromatin compaction at selected loci in fixed single cells. FRET-FISH method is based on the hybridization of oligonucleotide probes targeting proximal DNA sequences that are coupled to Donor and Acceptor fluorophores allowing the measurement of FRET between them.

This is very important, because, to my knowledge, there is no method for addressing locally the level of chromatin compaction at specific loci. Although performed in living cell conditions, some of the recent techniques are monitoring bulk chromatin compaction with no information to the related underlying genomic sequence.

First, the authors implement the developed FRET-FISH method by testing three different probe designs in fixed human haploid HAP1 cells. Then, FRET-FISH was optimized to measure the local chromatin compaction by testing several spacing distances between D and A probes at the *Ogt* gene locus. The authors validate the FRET-FISH technique by targeting six X-linked genes and correlate their single cell microscopy measurements to cell-based population technique such as ATAC-seq and Hi-C. Furthermore, the authors provide evidence that FRET-FISH technique detects chromatin compaction changes at particular nuclear localisation, but also along the cell cycle, and upon stress (ATP depletion, prolonged cell culture). This evidence supports the authors’ interpretations, although it does not rule out other interpretations.

Overall, the manuscript describes a novel microscopy-based technique with high quality data. This is an interesting and timely study presented in a clear manner. The technical and informatics aspects of the study were impressive, and will be of high interest to readers. Although the technique requires fixation of the cells, it has a high potential to better understand chromatin organization at specific loci and/or at regulatory sequences. In the study, the results part on chromatin compaction changes along the cell cycle, and prolonged culturing of cells are insufficiently developed and remain largely descriptive. Below are several concerns that should be addressed and additional experiments required to strengthen the different parts of the study, as follows:

Major concerns:

- 1) The DNA-FISH protocol used in combination with FRET is briefly mentioned in the Supplementary Methods section by referring the reference 30. More details in the main METHODS section should be provided about the FISH hybridization procedure associated with the novel FRET-FISH approach. In particular, it is important to mention clearly which fixative reagent, percentage and incubation timing

have been used, as well as the denaturation conditions and buffers performed during the preparation of cells. It seems very possible that under these conditions employed, the chromatin organization and consequently chromatin compaction in vivo might have unfolded, or – conversely – that some sequences might have artifactually folded and compacted; what can the authors say about this important caveat ?

2) Figure 1 panels b,c illustrate the three different probe designs implemented for the FRET-FISH method. For all three designs, the Donor (D) alone samples show a significant mean FRET efficiency (above 10%) with some FRET values reaching 40-50% in some cases. How the authors explain such a high FRET % values in the absence of any acceptor (A) probes. Then, it is not clear to me whether the FRET efficiency % in D+A have been corrected from this “artefact” FRET coming from the Donor alone?

3) With regard to the emission sensitized FRET assay used by the authors, it might be possible that the differences between FRET % distributions in the three designs (for the D+A samples) observed in Figure 1c could be explained by some variations of the fluorescence intensities of the Donor, and Acceptor probes? Indeed, it is known that the ratio Donor/Acceptor molecules could affect the FRET % and that acceptor molecules should not be the limiting factor.

The same concern can be applied to Fig1d-I results: How homogeneous is the labelling of detection oligonucleotides D and A ? How efficient is the hybridization of primary and detection D and A probes between cells for the different conditions group of alternating probes: G: 1, 2 and 4?

4) Figure 1 panels d-i illustrate the six different probes design with spacing distances (50,150,300 nt) between consecutive oligos for the FRET-FISH measurements. When comparing the six designs and their respective mean FRET % distributions, to which extent the hybridization of the oligo-probes to the targeted sequences (different alternating group number 1,2,4, and also their spacing) perturb and/or constraint the local organization of the chromatin at the locus and finally the FRET-FISH chromatin compaction readout ? Therefore, it is very important that the authors demonstrate that FRET-FISH distribution observed are not caused by the FISH probes per se. One suggested experiment to address this point could be to repeat the 6 different designs by hybridizing primary oligo probes to Ogt locus (G: 1,2 and 4 + spacing 50,150, 300) but without the D and A detection oligos (or without Cy3 and Cy5 fluorophores), and then to perform 3D DNA-FISH using DNA probes (nick-translated fosmids or oligont) recognizing flanking regions of the whole Ogt locus (comprising non labelled hybridized primary oligos) and measure the inter-probes 3D distances (between the flanking hybridized probes). Differences in inter-probes 3D distances might reflect different states of compaction depending on the probe designs.

5) How the authors justify the choice of the G1-S150 probe design for all their subsequent experiments? This point is not clearly explained in the study. By using AF488 and AF594 probes (Figure 1j-k), a clear bimodal FRET distribution is observed but the FRET efficiencies % are low (one population centered at

5%). The authors should depict both the FRET % distribution (AF488+AF594) and the residual FRET % from the AF488 donor alone to highlight the specificity of FRET signal measured.

6) In the Figure 2 a-f, the authors designed FRET-FISH probes against six genes located on ChrX. For each targeted gene, a bimodal FRET distribution is found whether the genes are constitutively inactivated or escaping the inactivation. Why escape genes (Ddx3x, Kdm5c, Pbd1) display a bimodal FRET % distribution that reflect two chromatin compacted states? The FRET % distribution profiles of Atp2b3 (inactivated gene) and Kdm5c (escapees) genes look very similar, I wondering if targeting genes located to chrX by FRET-FISH was the most relevant to validate the potential of the FRET-FISH technique to measure chromatin compaction. Would not be more relevant to design FRET-FISH probes against imprinted gene loci where 3C, allelic 4C, and Hi-C data are available as well as inter-probe 3D distance measurements by FISH (see IGF2/H19, CDKN1C/KCNQ1OT1 or Dlk1-Dio3 imprinted domains). As an alternative, it would be elegant to perturb the X-inactivation process (Xist RNA expression) and see if the bimodal FRET efficiency distribution is altered.

7) In Figure 3k-s, the authors show that FRET-FISH is able to detect chromatin compaction changes during the cell cycle. In line with these results, could it be possible that the bimodal FRET % distributions observed at the six targeted loci are not reflecting a specific gene-locus chromatin compaction state associated to active vs inactive gene but instead some global chromatin compaction changes related to the different cell cycle phase in this asynchronous cell populations?

8) To validate their method, the authors use FRET-FISH to detect chromatin compaction changes along the nuclear radius for three loci (Apt2b3, Kdm5c, Magix). In addition, it will be relevant to design FRET-FISH probes against well-defined LADs and inter-LADs domains (see Kind et al Cell 153, 178–192, March 28, 2013).

9) The authors are comparing the compaction of chromatin by FRET-FISH at the different loci before and after ATP depletion (Figure 3a-d), they should control whether the ATP depletion treatment itself affect the nuclei size that could explain the increase of local chromatin compaction observed at the different loci. Is the chrX volume altered upon ATP depletion?

10) In their introduction and discussion, the authors frequently associate the term “compaction” of chromatin with “density” of chromatin. However, several studies have recently described that condensed chromatin behaves differently depending on its scale of organization. For example, condensed chromatin may be liquid-like at the nanoscale (nucleosomal level) but solid-like at the mesoscale level (described by density for example; see Strickfaden et al., 2020, Cell 183, 1–13.; Miron et al., Sci. Adv. 2020; 6 : eaba881; Shaban et al. Genome Biology (2020) 21:95). In addition, FRET is a readout of interactions /proximities occurring locally between 1-10 nm. So, in others words, dense chromatin state at the mesoscale might not necessary imply local compacted chromatin measured by

FRET resolution. The authors might consider rewriting their statements between density and compaction of chromatin.

Minor concerns:

1) In Figure 1a, why “No FRET” is referred when Donor and Acceptor are separated by $>R_0$?

R_0 is the distance between a D-A FRET pair when the FRET efficiency is 50%, so FRET can also occur if the distance $r > R_0$ but $< 90-100 \text{ \AA}$. For example, for CY3-CY5 FRET pair $R_0 = 52.7 \text{ \AA}$, leaving plenty of room for FRET above the R_0 . In this regard, l 116- l 117 should be rephrased.

The R_0 of AF488/AF594 should be added.

2) In Supplementary Fig1b, the condition D (alone) for design 3 is missing in the graph.

3) In Supplementary Fig2, how #Acceptor dots values for G4 conditions (groups of 4 D and 4A) are smaller or equal to #Acceptor dots values for G2 conditions (groups of 2D and 2A) (see S2i and S2l panels for example). Is the # Acceptor dots value not reflecting the amount of fluorescent Acceptor molecules ? Authors should clarify this point ?

4) In Supplementary Fig4 a-r, it appears that a certain variability in the FRET % distribution is observed between the three experiments at each targeted locus. The authors could compare the FRET distributions (two by two) using a quantile-quantile (Q-Q) plot statistical analysis to see how variable are the FRET measurements between experiments.

5) To illustrate some of the FRET results in the study, it would be helpful to show some microscopy FRET images next to them.

Reviewer #2 (Remarks to the Author):

Ana Mota et al. in the submitted manuscript “FRET-FISH probes chromatin compaction at individual genomic loci in single cells” report the use of FRET between FISH probes to measure chromatin compactization. Sequence specificity is a clear advantage of FRET-FISH method. They highlight this capability by examining chromatin compactization of the genes located in X chromosome. In addition, the authors demonstrate the application of FRET-FISH to measure compactization of chromatin in ATP

depleted cells. In general, FRET-FISH should be applicable to study chromatin compactization under various conditions and multiple cell types.

Overall, the manuscript is interesting and represents a novel application of FRET-FISH method, but lacks some important experiments and corrections. Thus, I am happy to recommend this manuscript to be accepted for publishing in Nature Communications after a major revision:

Major points:

1. The downside of all FISH-based methods is distortion of the local DNA structure by hybridization with probe oligonucleotide. The authors are not addressing this issue in the manuscript and assume that the arrangement of chromatin stays intact after hybridization. I strongly recommend including these statements in the manuscript text. The authors should discuss how this might influence FRET measurements. This discussion should be supported by appropriate control experiments.
2. FRET efficiency calculation is very primitive and sensitive to experimental conditions setup. The obtained value might be influenced by local environment of the fluorescent dyes which is reflected by relatively broad distributions of FRET efficiency. The authors also write in the supplementary information "...the FRET signal can be affected by multiple effects including cross-excitation, bleed-through and crosstalk, to minimize inter-sample variability we performed all our FRET-FISH experiments using custom-made multi-chambered coverslips." More accurate measurement could be performed using acceptor photobleaching and/or FLIM methods. This is essential for the described method and should be performed in order to convince readers about method reproducibility.
3. Why authors measured FRET efficiency only in non-G1 and G1 cells? Can other cell cycle stages be resolved? If not, please provided detailed explanation. It would be extremely interesting to measure FRET efficiency in M Phase.

Minor points:

1. The authors write, "...suggesting that Hi-C might be better suited than ATAC-seq to probe chromatin density genome wide." Please explain this statement in more details. Is it only based on the obtained correlations or supported by the unmentioned experiments?
2. The authors state, "To further validate FRET-FISH, we examined whether our method can also distinguish between different chromatin density states along the nuclear radius." Why you need further validation if the previous paragraph stated that it is a valid method?
3. The authors measured chromatin condensation in MEFs after 10 additional passages. It would be informative to describe how the first passage of MEFs is determined? Is it counted from purchase, are the passages done by provider company considered?
4. The authors bravely state, "...FRET-FISH could also be applied to study enhancer-promoter contacts or chromatin loop organization in single cells, without the need to rely on super-resolution microscopy

techniques to bypass the inherent spatial resolution limitations of DNA FISH.” However, it not only microscope resolution is the limiting factor of FISH-based method. Even more important is the size of FISH probes itself. Please provide the comparison of FISH probe size and achievable resolution of the microscopes.

5. The authors mention on line 489 “We acquired images in the donor and acceptor fluorescent channel separately, and only considered FISH dots with similar x, y, z coordinates (radial threshold: 7 pixels) between the donor and acceptor channels”. Distance in pixels is not informative, please write in micrometres and explain why such distance is selected.

6. The authors state in the supplementary information, “we prepared samples and hybridized them with FRET-FISH probes using a modified 3D DNA FISH protocol” What are these modifications?

7. ATP depletion experiment done at quite strange conditions by adding 2-Deoxyglucose and Sodium azide directly to growth medium which contains glucose. Only partial depletion could be achieved because of 2-Deoxyglucose and glucose competition. Why authors have not measured ATP level and instead tested amount of nascent RNA in the nucleus with the Click-iT RNA Imaging Kit?

Reviewer #3 (Remarks to the Author):

The authors of this work propose a novel approach, FRET-FISH, which combines the specificity of locus-specific FISH with the sensitivity of FRET to assay chromatin compaction and measure precise (local) differences in chromatin compaction in a locus-specific manner at the single-cell level. This approach addresses the lack of tools assessing chromatin compaction or differences in chromatin density in the field of genome organization.

The authors of this work demonstrate the following:

- 1- Chromatin density can be measured by comparing FRET-FISH efficiency, reproducibly.
- 2- The chromatin density measured can be compared to a) chromatin accessibility measured by ATAC-seq, and b) chromosome contact frequencies of the same loci measured by Hi-C.
- 3- FRET-FISH can detect changes in local chromatin compaction after chemical-induction of chromatin compaction.
- 4- FRET-FISH can detect differences in local chromatin compaction at different passage numbers as well as in G1 vs non-G1 cells. This final point emphasizes the importance of controlling for cell passages, tissue culturing techniques as well as cell cycle effects when making different chromatin biophysical measurements.

Overall, this is a proof-of-principle study that shows that FRET-FISH is an invaluable tool that can be complementary to other technologies that assess genome organization, cell-to-cell variability, and differences in chromatin density measurements. I think this manuscript is appropriate for publication in Nature Communications with more in-depth analysis of the imaging data, some clarification of the current data addressing inter-allelic differences, and after addressing the comments below.

Major comments:

1. There should be a figure or a clearer outline explaining the rationale for choosing the genomic size of the targets and number of oligos used in this study. The first two sections discussing implementation and optimization of FRET-FISH clearly describe the three FRET-FISH designs as well as the rationale for using 6 loci on the X chromosome. However, since this is the first study to introduce FRET-FISH as a tool to assess chromatin density, it would be beneficial to the field to show a clearer assessment of FRET-FISH efficiency for different ranges of number of oligos used and different genomic sizes targeted. This could be shown with preliminary data that was used for experimental optimization and could address the following in the text:

- ln122 and ln136: first test experiment targets 20 kb Myc gene locus and 134 primary D-A oligo pairs, what is the genomic size and # D-A oligo pairs for the second test at the Ogt locus, ln181? What were the criteria for picking genomic size and #oligos?

- ln338: in terms of experimental design for FRET-FISH: what was the lowest/highest #oligos tested?

- ln435: what are the range of sizes of genomic regions assessed?

- Are there any preliminary tests looking at loci on autosomes? How does that compare to X chromosome data?

2. There should be a clearer outline and description of the findings for the inter-allelic differences. One of the more significant findings of the study is the potential ability for FRET-FISH to differentiate between alleles. This point is brought up several times in the text (ln53, ln109, ln261, ln297, ln319, ln405) but this data and its interpretation is not always clearly set up or described in the text or the figures (except for Figure 3e-j, n-s, and some supplementary figures). It seems that the bimodal FRET efficiency graphs representing different populations of compaction are not meant to imply, for example, bimodal distributions within single cells. If so, the authors are encouraged to make this explicit in the text. Several other examples where the text and data can be more explicit:

- ln188, Fig1d-i, how can the authors be sure the bimodal curve is not due to differences in the cell-cycle stage?

- In213, Fig. 2a-f and Supplementary Fig. 4a-r, what accounts for the “two distinct chromatin states”? Can these datasets be split by cell cycle and by alleles?

- Fig2 j-l: are the different alleles combined for these datasets? The data should be clearly described to justify the statement emphasizing single-allele resolution in In261.

- For Fig3 e-j supplemental Fig5 h-m, are the datasets separated by cell cycle phase? If not, how can the authors definitively say the inflection point in the curve is due to differences in the level of compaction between the alleles? Can the authors clarify and comment on that?

3. Again, one of the powerful findings in this study is the ability to detect differences in FRET-FISH efficiencies within a population of cells for one locus. I found that there is no clear description of the cell-to-cell variability when interpreting the data. For example, in In186 and in Fig1d,e,f: do the different peaks represent different phases of the cell cycles? Different alleles (active vs inactive X)? Other differences within a cell population? These points should be clearly distinguished while interpreting the data and perhaps segueing to the following set of experiments addressing cell cycle phases.

4. After looking through the methods, I found that the measurements were done using max projection and analyzed in 2D: e.g., “...2D nuclear segmentation mask of the cell in which the dot was identified” and “...max projection in the axial direction”. This piece could be misleading especially when addressing the location of loci tested within the radial concentric layers within the nucleus (e.g. In255, Fig2f-j). How can the authors be sure the dot is not at the nuclear periphery in the axial position? Although I would recommend doing all the analysis in 3D, one way to acceptable way to address this could be to make the measurements in 3D for 2 loci or datasets and showing it still agrees with the max projection analysis shown in the rest of the figures.

5. There should be a clear assessment and/or discussion of how FRET-FISH efficiency can be affected in the following contexts, and how these contexts could affect the interpretation of the data:

- within a more compact or less compact chromatin environment

- what role quenching could play in more compact chromatin contexts

- possibility of reduced probe hybridization due to an increase in compaction. (e.g. the lower accessibility in mitotic cells)

- In344: has FRET-FISH efficiency been tested robustly at the same locus in different phases of the cell cycle?

6. For the results on In247, revealing the detection of chromatin compaction changes along the nuclear radius, addressing the following questions can help elucidate the biological importance of these measurements and observations:

- How were the nuclei selected for each different concentric nuclear layer? Did the authors select nuclei specific for each nuclear layer?

OPTIONAL:

- Are the two alleles within the same concentric layer? How often?

- How does the location within the layer correlate with transcriptional state? Is the transcriptional state accounted for in the measurements at all? Can the authors show or comment on some of that expression data (even if its nascent transcript quantification)?

- Were there other notable differences when assessing the loci within different nuclear layers?

Minor comments:

- In35: lack of chromatin accessibility not “Chromatin accessibility is often used as a proxy for chromatin compaction...”

- In47: compact not compacted: “...with peripheral loci being more compacted and central ones less compacted.”

- In56: “Three major types of chromatin have been described in the nucleus of mammalian cells...”. Two of the papers cited are Drosophila studies.

- In86 include a reference for FISH/bridge/multiplexing technologies e.g. Beliveau, Boettiger, Nir, Bintu et al 2017, or another comprehensive reference.

- With respect to iFISH, can the authors clarify or comment on how the probes used for this work are similar or different from Oligopaints (e.g. Beliveau et al 2015, Boettiger et al 2016)?

- In229, fig 2g,h, and Supplementary Table2:

1. Are the reported FRET-FISH efficiencies averaged for the two peaks? If so, can they be split by cell cycle stage and allele to see how they compare to Hi-C and ATAC-seq data?

2. Are the reads for Hi-C contacts, normalized reads?

The Hi-C contact frequency could be better assessed through the contact maps: contact frequency observed/expected, rather than mappable reads. Have the authors tried to get measurements for different loci directly from the normalized contact maps?

This could clarify the discrepancy between ATAC-seq and Hi-C data, In239.

- In230: The anti-correlation between FRET-FISH and Hi-C, not the correlation.
- In232: the conclusion at the Ddx3x locus is unclear, what are the authors trying to say about the differences in observations for Ddx3x?
- In90: “Although it is assumed that chromatin accessibility and density are inversely correlated, the exact relationship between these two biophysical properties remains unknown...”. The authors bring up a very important point about the potential implication of having less dense chromatin that is inaccessible due to protein occupancy, but do not address or consider this point with any of their data. It would be great to bring this up while considering some of their observations in the discussion if it applies. In addition, can the authors comment on the inverse relationship between chromatin accessibility and chromatin compaction and how that may affect their FRET-FISH results?
- Figure 1c: can the authors comment on why the donor violin plots in all three designs has such a wide distribution for FRET efficiency?
- Figure 1k: can the authors comment on why although there are two distinct FRET efficiency peaks, why is it that the overall efficiency of FRET-FISH <20% compared to the other fluor set?
- In307: Can the authors comment more clearly as to why Magix gene shows a higher FRET-FISH efficiency in mitotic cells but not in G2 cells? Would this be related to the possibility that replicating DNA is less compact, or could the proximity of an additional copy of DNA be quenching the FRET-FISH signal?

The following two comments are OPTIONAL requests for preliminary data that I would like to see and believe would make the paper and the approach presented more impactful, but I understand if it's not feasible or if they cannot be addressed in a timely manner:

- The authors only survey loci on the sex chromosomes, it would be interesting to see how comparable those observations would be on an autosome by checking 1-2 other chromosomes or loci.
- Can the authors show different applications for the FRET-FISH? For example, are there any other preliminary results showing how the technology applicable to assess other biological questions: enhancer-promoter contacts or chromatin loop organization?

Reviewer #4 (Remarks to the Author):

The paper from Mota et al describes a novel method for chromatin compaction assessment. Compaction of genomic loci is an important characteristic in respect to their transcriptional activity and, therefore, this paper is very relevant and timely for the chromosome/chromatin research field.

The authors developed an elegant method, so called FRET-FISH, combining the previously known oligo-FISH and FRET (fluorescence energy transfer), by targeting oligo-probes to a particular gene locus and detecting them with a certain combination of fluorophores serving as donors with a lower wavelength emission and acceptors with a higher wavelength emission. The rationale behind this strategy is that in compacted chromatin, donor and acceptor fluorophores occur in physical proximity, and thus the excitation of a donor results in an excitation of an acceptor via FRET. The beauty of the approach is that one can estimate compaction of specific genomic loci or even various parts of the same gene. I find this idea excellent and the method as very useful.

Therefore, the work is potentially valuable for the chromatin community and I am generally positive about it. However, I see several drawbacks, which in my view have to be attended by the authors before considering the manuscript for publication.

1. Introduction

Lines 67-85. When mentioning DNA staining and fluorophores fused to histones, the authors could do a better job discussing nucleic acid stains other than DAPI, as well as genetically-encoded methods such as, e.g., HI-NESS (10.1093/nar/gkab993)

2. Results

(1) The authors describe in great detail several probe designs that were ultimately recognized as suboptimal for the method and were not used for FRET-FISH validation. Lines 114-196 are dedicated to description of 9 versions of oligo-probe spacing and detecting probe designs – all based on conjugation with Cy3 as a donor and Cy5 as an acceptor, which, at the end of the day, were not used. The rest of the experiments were performed with only one version of oligo-probe spacing and using AF488 as a donor and AF594 as an acceptor. To focus the Results section, I suggest the authors to consider shortening the part about the optimizing experiments and moving the corresponding figures to Supplementary material.

In this respect, it is also important to show why the AF488/594 couple is superb to the Cy3/Cy5. For example, when comparing Fig 1j and SFig 3, it is not clear why there is a difference in the FRET channel in the upper two rows.

(2) Lines 208-218: To validate the method, the authors chose 3 genes that are constitutively silenced on Xi chromosome and 3 genes escaping inactivation. They measure FRET efficiency for all these genes in MEFs and find no obvious difference between silenced and active genes. Very surprisingly, the authors

do not distinguish between the Xa and Xi chromosomes, although this was the point of the whole experiment – to see the difference in gene compaction depending on their activity.

4 comments to this part: (1) Why didn't the authors distinguish between Xa and Xi, e.g. by Xist-detection or immunostaining, as it is routinely used (10.1016/j.cell.2021.10.022; 10.1007/s00412-021-00754-z)? Since the authors use automated acquisition and the coordinates of nuclei are stored, this step could be done after FRET-FISH to avoid interference with other fluorochromes. (2) Xi identification is especially important in the view that during cultivation, MEFs quickly become tetraploid or even might have an aneuploidy for one or several chromosomes, including X. Have the authors checked for this? (3) Since this paper is mostly about interphase chromatin compaction, it is really surprising that there are no images of nuclei with FRET signals for each of the studied genes so that the reader could see the number and location of the signals. (4) In connection to this: there is no statistics on how many AF488, AF594 and FRET signals per nucleus were usually detected – in other words, what the reproducibility of FRET-FISH from nucleus to nucleus is.

I do appreciate the colossal work of collecting thousands of signals in each experiment shown in the paper and all the efforts the authors undertook to develop a completely automated pipeline for signal identification. However, in this case, more careful observations of fewer nuclei but with a more precise chromosome identification can be of a higher benefit.

Unfortunately, all the rest of the experiments in the paper that I discuss below - in (4), (5) and (6) – are based on the same genes and thus require Xa and Xi chromosome identification.

I am also wondering why the authors have chosen the X-chromosome but not genes with clear LAD and interLAD signatures, which do not require chromosome identification but are unmistakably different and have well defined chromatin status (e.g., 10.1101/gr.141028.112).

(3) The authors show that FRET efficiency of the six selected probes for the X-chromosome strongly correlates with ATACseq and Hi-C read counts extracted from publically available databases on MEFs, which is a nice and convincing validation of the method. Furthermore, the authors attempt to show a similar correlation with nuclear radius (Lines 247-269). Indeed, it is well established that 3D radial distribution of chromatin in spherical nuclei is dictated by gene density and transcriptional activity, both being high towards the nuclear interior, which thus is supposedly filled with less condensed chromatin. However, surprisingly and illogically, the authors perform radial measurements of signal positioning (a) on 2D projections of segmented nuclei, (b) using MEFs with very flat nuclei (thus with a very small interior) and (c) using genes on the X-chromosome, which is known to be very flat and peripheral, especially in case of Xa (see e.g., 10.4161/nucl.2.5.17862; 10.1016/j.cell.2021.10.022). I suggest that the authors either do this evaluation properly in 3D and on another chromosome or completely remove this section from the manuscript.

(4) Lines 270-27: The authors refer to SFig 5a as demonstrating a drastic reduction of transcriptional activity. However, the micrograph 5a shows only labeled nucleoli. To prove reduction in expression of the 4 studied genes, qPCR is the most direct and reliable method for assessment.

(5) Lines 283-298: This is the vaguest part of the manuscript. The definition of the number of cell passages is very superficial and defined as <10 and >10. What happens with cells after the 10th passage? Why does chromatin become more condensed? Does it mean that fibroblasts become senescent? Or is the cell cycle changing? Or is there an increase in ploidy? In my view, it makes no sense to perform FRET measurements on a system so poorly defined.

(6) Lines 301-320: Finally, the authors show that FRET efficiency is higher in G1 cells in comparison to the rest of the cell cycle stages, using Hoechst 33342 staining as a criterion for G1 stage. Unless I have missed something, I have not found any evidence in the manuscript that cells with a high Hoechst 33342 staining are in G1 stage. I have no much experience with mouse fibroblasts, but for me it is not obvious that G1 cells have stronger Hoechst staining – e.g., what is with cells in prophase? The use of anti-Ki67 staining, the most common marker for G1 phase (e.g., 10.1023/a:1009210206855), would free the authors from unnecessary speculations about G1 and non-G1 cells, including Hoechst-low and Hoechst-high G1 subpopulations (Lines 317-318). Although this approach might reduce the number of analyzed loci, the work will gain more solid conclusions.

3. Figures

(1) Why are the replicates for each FRET-FISH experiment not averaged in a single graph? As far as I could figure it out, graphs in Fig 1c and SFig 1b, Fig 3a-d and SFig 5c-f, Fig 3e-j and SFig 5h-m, Fig 3n-s and SFig 6c-h show two replicates. If the authors find that, in addition to the averaged graph in the main text, it is important to show both replicates separately, it can be done in Supplementary data.

(2) Multiple letters in figures make them unnecessarily overloaded. I see no need in such an ample labelling in figs 2 and 3 (as well as corresponding supplemental figures), because the names of all genes and conditions are conveniently and clearly marked above the graphs.

Point-by-point response to the Reviewers' comments

Reviewer #1

The authors describe a novel microscopy-based approach named "FRET-FISH" combining Förster Resonance Energy Transfer (FRET) measurements with Fluorescence in situ Hybridization (FISH) to analyze the chromatin compaction at selected loci in fixed single cells. FRET-FISH method is based on the hybridization of oligonucleotide probes targeting proximal DNA sequences that are coupled to Donor and Acceptor fluorophores allowing the measurement of FRET between them.

This is very important, because, to my knowledge, there is no method for addressing locally the level of chromatin compaction at specific loci. Although performed in living cell conditions, some of the recent techniques are monitoring bulk chromatin compaction with no information to the related underlying genomic sequence.

First, the authors implement the developed FRET-FISH method by testing three different probe designs in fixed human haploid HAP1 cells. Then, FRET-FISH was optimized to measure the local chromatin compaction by testing several spacing distances between D and A probes at the Ogt gene locus. The authors validate the FRET-FISH technique by targeting six X-linked genes and correlate their single cell microscopy measurements to cell-based population technique such as ATAC-seq and Hi-C. Furthermore, the authors provide evidence that FRET-FISH technique detects chromatin compaction changes at particular nuclear localization, but also along the cell cycle, and upon stress (ATP depletion, prolonged cell culture). This evidence supports the authors' interpretations, although it does not rule out other interpretations.

Overall, the manuscript describes a novel microscopy-based technique with high quality data. This is an interesting and timely study presented in a clear manner. The technical and informatics aspects of the study were impressive and will be of high interest to readers. Although the technique requires fixation of the cells, it has a high potential to better understand chromatin organization at specific loci and/or at regulatory sequences. In the study, the results part on chromatin compaction changes along the cell cycle, and prolonged culturing of cells are insufficiently developed and remain largely descriptive. Below are several concerns that should be addressed and additional experiments required to strengthen the different parts of the study, as follows:

We thank the Reviewer for appreciating our work and valuing the potential impact that FRET-FISH can have in the field of chromatin biology and 3D genome organization. We are very grateful to the Reviewer for their insightful comments and suggestions that helped us further strengthen our manuscript and hopefully even more convincingly highlight the technical performance of FRET-FISH. We hope that the Reviewer will be satisfied with our revisions and be supportive of publication of our work in *Nature Communications*.

Major concerns:

1) The DNA-FISH protocol used in combination with FRET is briefly mentioned in the Supplementary Methods section by referring the reference 30. More details in the main METHODS section should be provided about the FISH hybridization procedure associated with the novel FRET-FISH approach. In particular, it is important to mention clearly which fixative reagent, percentage and incubation timing have been used, as well as the denaturation conditions and buffers performed during the preparation of cells. It seems very possible that under these conditions employed, the chromatin organization and consequently chromatin compaction in vivo might have unfolded, or – conversely – that some sequences might have artifactually folded and compacted; what can the authors say about this important caveat?

We thank the Reviewer for this important point. We now provide a step-by-step FRET-FISH protocol in the Supplementary Methods section in the revised **Supplementary Information**. In the same section, we have also added a step-by-step protocol for producing FRET-FISH probes.

Regarding the Reviewer's concern about possible drawbacks of the hybridization conditions used in our FRET-FISH protocol, we are aware of this possibility and over the years have strived to optimize our protocol to minimize the number of steps that can potentially disrupt native chromatin conformation. Indeed, we have managed to minimize the time of DNA denaturation as well as the temperature of denaturation. To our knowledge, our protocol features the mildest denaturation conditions among published oligo-based DNA FISH protocols that rely on denaturation (see for example PMID: 30361340; 33619390; 34591592; 33505024; 32822575; 32719531). While typical DNA FISH protocols recommend denaturing the sample for 3–5 min at 80-90 °C, we have found that 1-2 min at 75 °C is typically sufficient to yield a satisfactory signal. In all our DNA FISH experiments (including FRET-FISH), we also make sure to proceed to hybridization right after cell fixation to avoid sample deterioration caused by storage, as we observed this can have a negative effect on the quality of FISH signals and, more generally, on nuclear morphology.

Despite this concern, previous studies comparing results obtained by DNA FISH with those obtained by CRIPSR/Cas9-assisted detection of DNA loci, including through live-imaging, showed a very good concordance between the methods (see for example PMID: 28355536 and 27222091). Moreover, in our study, we detected a strong anti-correlation between FRET-FISH and ATAC-seq, as expected, further suggesting that no major structural changes are introduced during the denaturation procedure in our FRET-FISH protocol, at least at the length scale we tested.

2) Figure 1 panels b,c illustrate the three different probe designs implemented for the FRET-FISH method. For all three designs, the Donor (D) alone samples show a significant mean FRET efficiency (above 10%) with some FRET values reaching 40-50% in some cases. How the authors explain such a high FRET % values in the absence of any acceptor (A) probes. Then, it is not clear to me whether the FRET efficiency % in D+A have been corrected from this “artefact” FRET coming from the Donor alone?

We thank the Reviewer for this remark. The fact that a relatively high FRET efficiency can be observed even in the absence of acceptor (A) probes is most likely caused by the presence of very bright non-specific signals that bleed through into the FRET channel. This problem is normally bypassed when both donor (D) and A probes are hybridized because we then rely on the co-localization of both probes to call FRET signals confidently. However, when only D probes are used (as in the control experiments shown in **Figure 1** and **Supplementary Fig. 1**) our confidence in calling true signals unavoidably drops. In any case, we fully account for this effect when calculating the FRET efficiency (please see **Methods**). (Please note that, following the remark of Reviewer #2, throughout the revised manuscript and figures we have now substituted the expression 'FRET efficiency' with 'FRET-FISH score' which in our view better describes our measurements.)

3) With regard to the emission sensitized FRET assay used by the authors, it might be possible that the differences between FRET % distributions in the three designs (for the D+A samples) observed in Figure 1c could be explained by some variations of the fluorescence intensities of the Donor, and Acceptor probes? Indeed, it is known that the ratio Donor/Acceptor molecules could affect the FRET % and that acceptor molecules should not be the limiting factor.

The same concern can be applied to Fig1d-l results: How homogeneous is the labelling of detection oligonucleotides D and A? How efficient is the hybridization of primary and detection D and A probes between cells for the different conditions group of alternating probes: G: 1, 2 and 4?

In our FRET-FISH probes, the D oligos alternate with the A oligos (in the G1 design, each D oligo follows one A oligo; in the G2 design, two D oligos follow two A oligos, and so on). Therefore, both sets of oligos (D and A) target the same genomic region, hence their hybridization efficiency should be affected in the same way. We therefore do not foresee any significant difference in the hybridization efficiency between A and D oligos, which could affect the resulting FRET-FISH score.

We designed our FRET-FISH probes aiming at having an as homogeneous distribution of the oligos as possible within a given genomic region of interest. Our probe design pipeline aims at minimizing the number of 'isolated' oligos within a given target region, i.e., the number of oligos that do not have any unique oligo neighbor within a threshold genomic distance. This applies to both D and A oligos. In this way, all the probes contain an equal or near-equal number of evenly spread D and A oligos, without large gaps in between oligos or large islands (> 1kb) of A-only or D-only oligos, which could indeed bind with different efficiency to their target. Therefore, we do not have any obvious reason to suspect that, on average, D and A oligos behave differently in the FRET-FISH probe designs that we have tested. In the **new Supplementary Fig. 2a**, we now show the probe span and D and A oligo distribution for each of the *Ogt* probe designs described in **Fig. 1e-j**. As the Reviewer can see, the oligos are overall homogeneously spread, with only few small sized gaps where our pipeline could not find unique oligo sequences or sequences withing the specified GC-content range.

4) Figure 1 panels d-i illustrate the six different probes design with spacing distances (50,150,300 nt) between consecutive oligos for the FRET-FISH measurements. When comparing the six designs and their respective mean FRET % distributions, to which extent the hybridization of the oligo-probes to the targeted sequences (different alternating group number 1,2,4, and also their spacing) perturb and/or constraint the local organization of the chromatin at the locus and finally the FRET-FISH chromatin compaction readout? Therefore, it is very important that the authors demonstrate that FRET-FISH distribution observed are not caused by the FISH probes per se. One suggested experiment to address this point could be to repeat the 6 different designs by hybridizing primary oligo probes to *Ogt* locus (G: 1,2 and 4 + spacing 50,150, 300) but without the D and A detection oligos (or without Cy3 and Cy5 fluorophores), and then to perform 3D DNA-FISH using DNA probes (nick-translated fosmids or oligont) recognizing flanking regions of the whole *Ogt* locus (comprising non labelled hybridized primary oligos) and measure the inter-probes 3D distances (between the flanking hybridized probes). Differences in inter-probes 3D distances might reflect different states of compaction depending on the probe designs.

We thank the Reviewer for raising this important issue and for the suggested experiment. Following the Reviewer's suggestion, we have now tested whether, upon hybridization of the *Ogt* probes tested, the 3D distance between two regions flanking the targeted locus changes. As shown in the **new Supplementary Fig. 2d and e**, the distribution of 3D distances between the flanking probes was largely unaffected by co-hybridization of the FRET-FISH probe, for the majority of the probe designs tested. For the G1S150 design, which we chose as default design in all subsequent experiments (please see reply to the Reviewer's comment #5 below), the 3D distance distribution was not significantly different from the one obtained from a control sample in which no FRET-FISH probe was hybridized. Based on these new results, we are therefore confident that hybridization of FRET-FISH probes—using the gentle denaturation conditions that we have described above (see response to the Reviewer's comment #1)—does not result in a sizable perturbation of the local 3D genome structure at the scale relevant to our assay.

5) How the authors justify the choice of the G1-S150 probe design for all their subsequent experiments? This point is not clearly explained in the study. By using AF488 and AF594 probes (Figure 1j-k), a clear bimodal FRET distribution is observed but the FRET efficiencies % are low (one population centered at 5%). The authors should depict both the FRET % distribution (AF488+AF594) and the residual FRET % from the AF488 donor alone to highlight the specificity of FRET signal measured.

We thank the Reviewer for this remark and apologize for not having been clearer. Among all the design tested, G1S50, G1S150 and G2S50 yielded similar results indicative of their ability to detect two different compaction states given the bimodal distribution of the FRET-FISH score. We reasoned that having a larger spread between D and A oligos should in principle allow for a higher dynamic range, i.e., allow the detection of a larger spectrum of compaction states as compared to having D and A oligos adjacent, as in our initial test *Myc* probe. For this reason, we chose the G1S150 design over the G1S50 and the G2S50. We cannot exclude, however, that additional design optimizations might further increase the dynamic range of

FRET-FISH. We have now added these considerations in our revised manuscript, hoping to present a clearer motivation for our probe design choice.

Regarding the lower FRET-FISH scores obtained using AF488-AF594 as compared to Cy3-Cy5, these differences can be explained by the lower crosstalk and the bleed-through into the FRET channel in the case of Alexa Fluor dyes. As we now show in the **new Supplementary Fig. 3c**, the ratio of FRET intensity to D/A intensity is more favorable for Alexa Fluor dyes compared to cyanide dyes. Moreover, Alexa Fluor dyes suffer less from quenching effects and are more photostable, which is essential when imaging multiple fields of view.

6) In the Figure 2 a-f, the authors designed FRET-FISH probes against six genes located on ChrX. For each targeted gene, a bimodal FRET distribution is found whether the genes are constitutively inactivated or escaping the inactivation. Why escape genes (*Ddx3x*, *Kdm5c*, *Pbdc1*) display a bimodal FRET % distribution that reflect two chromatin compacted states? The FRET % distribution profiles of *Atp2b3* (inactivated gene) and *Kdm5c* (escapees) genes look very similar, I wondering if targeting genes located to chrX by FRET-FISH was the most relevant to validate the potential of the FRET-FISH technique to measure chromatin compaction. Would not be more relevant to design FRET-FISH probes against imprinted gene loci where 3C, allelic 4C, and Hi-C data are available as well as inter-probe 3D distance measurements by FISH (see *IGF2/H19*, *CDKN1C/KCNQ1OT1* or *DIK1-Dio3* imprinted domains). As an alternative, it would be elegant to perturb the X-inactivation process (*Xist* RNA expression) and see if the bimodal FRET efficiency distribution is altered.

This is a very valid comment, thank you for that. The reason why we initially designed probes targeting genes on chrX is because we reasoned that this approach would allow us to validate the ability of FRET-FISH to distinguish two supposedly different chromatin compaction states on the active and inactive chrX. Moreover, we reasoned that probing escapees vs. non-escapees would further expand the repertoire of possible compaction states that FRET-FISH could detect. Indeed, when we observed a bimodal distribution of the FRET-FISH scores for the probes targeting different loci on chrX that we tested, we suspected that the two modes might correspond to the active vs. inactive homologue. We were however surprised to see that both escapees and non-escapees showed this strong bimodality. Motivated by the Reviewer's comment we now tried to investigate this aspect further by performing a new set of experiments in which we combined FRET-FISH with smFISH against *Xist* in order to measure compaction separately for the active and the inactive homologues of *Magix* (non-escapee) and *Kdm5c* (escapee). These experiments proved to be rather challenging, since the combination of FRET-FISH and smFISH led to a substantial decrease in the intensities of the FRET signals detected and hence loss of sensitivity, possibly due to interference between the fluorescence dyes used, as already anticipated by Reviewer #4. Therefore, we could no longer observe bimodality in the FRET-FISH score distributions. However, we reasoned that, in this particular case where FRET signals are very weak, our analysis might benefit from image deconvolution to enhance the contrast and allow better detection of true FRET signals. To this end, we leveraged a powerful software for image deconvolution, which we recently developed in parallel to this work (see doi.org/10.21203/rs.3.rs-1303463/v1, currently under revision for

Nature Biotechnology). Indeed, after deconvolution, we could again observe a bimodal distribution of the FRET-FISH score for both genes tested, even though not as clearly as when we performed FRET-FISH alone. This prompted us to compare the FRET-FISH score distributions between the active and the inactive homologues. As shown in the **new Supplementary Fig. 8**, we could observe the expected difference in the distribution of the FRET-FISH score between active and inactive homologues, especially for the non-escapee gene *Magix*. However, despite the expected trend, the difference in the distributions was not statistically significant. Importantly though, both active and the inactive homologues showed a broad distribution of the FRET-FISH score as for both homologues pooled together, and in the case of *Magix* the active homologue displayed a clear bimodality, suggesting to us that there are other factors affecting the observed FRET-FISH score distribution bimodality. Indeed, when we profiled multiple loci on an autosomal chromosome (chr18) (as suggested by the Reviewer as well as Reviewer #3), we could observe a clear bimodality in the corresponding FRET-FISH score distributions (see **new Fig. 3b and c**), demonstrating that the distributions obtained for genes on chrX genes cannot be fully explained by the fact that this chromosome is present in two very different compaction states in female cells. In fact, as we show throughout the manuscript, the FRET-FISH score distribution bimodality represents different states of local compaction modulated by various parameters such as cell cycle phase, distance to the lamina, cell aging (perhaps due to accumulation of DNA damage) and possibly more. We believe that FRET-FISH is a powerful tool that opens up new lines of future investigation to fully understand and eventually predict the extent of chromatin compaction at any given genomic locus.

7) In Figure 3k-s, the authors show that FRET-FISH is able to detect chromatin compaction changes during the cell cycle. In line with these results, could it be possible that the bimodal FRET % distributions observed at the six targeted loci are not reflecting a specific gene-locus chromatin compaction state associated to active vs inactive gene but instead some global chromatin compaction changes related to the different cell cycle phase in this asynchronous cell populations?

Most of the analyses described in our manuscript were done on G1 cells (except for the part of our study in which we examined the influence of the cell cycle phase on local chromatin compaction). Hence, the observed bimodality in the FRET-FISH score distributions is not a simple consequence of working with an asynchronous cell population, even though the cell cycle phase influences the shape of the FRET-FISH score distributions as shown in the **revised Fig. 4a-f**. In fact, as we try to demonstrate throughout the manuscript, the bimodal shape of the FRET-FISH score distributions is most likely the result of multiple inter-playing variables, including cell cycle phase (and, more generally, global chromatin condensation), distance to the nuclear lamina, and cell passage number. As more factors influencing local compaction likely exist, we believe that FRET-FISH represents a powerful tool that can be harnessed to investigate this important feature of chromatin at the nanoscale.

8) To validate their method, the authors use FRET-FISH to detect chromatin compaction changes along the nuclear radius for three loci (Apt2b3, Kdm5c, *Magix*). In addition, it will be relevant to

design FRET-FISH probes against well-defined LADs and inter-LADs domains (see Kind et al Cell 153, 178–192, March 28, 2013).

We thank the Reviewer for this suggestion. As already mentioned above, we have now designed FRET-FISH probes for 6 loci on chr18, including 3 loci embedded in constitutive lamina associating domains (LADs) and 3 loci inside inter-LAD (iLAD) regions. The results of these new FRET-FISH experiments show that FRET-FISH probes targeting regions inside cLADs on chr18 yield higher FRET-FISH scores than probes targeting regions embedded in iLADs on chr18, as presented in the **new Fig. 3b and c**.

9) The authors are comparing the compaction of chromatin by FRET-FISH at the different loci before and after ATP depletion (Figure 3a-d), they should control whether the ATP depletion treatment itself affect the nuclei size that could explain the increase of local chromatin compaction observed at the different loci. Is the chrX volume altered upon ATP depletion?

This is a very good point. We have now re-analyzed our data and found that the DNA channel intensity per pixel is significantly higher in ATP-depleted samples compared to controls, while the nuclear area is decreased in the former, as expected (please see **new Supplementary Fig. 7a-d**). We believe these results further corroborate our claim that FRET-FISH can detect expected changes in chromatin compaction at a level of individual gene loci.

10) In their introduction and discussion, the authors frequently associate the term “compaction” of chromatin with “density” of chromatin. However, several studies have recently described that condensed chromatin behaves differently depending on its scale of organization. For example, condensed chromatin may be liquid-like at the nanoscale (nucleosomal level) but solid-like at the mesoscale level (described by density for example; see Strickfaden et al., 2020, Cell 183, 1–13.; Miron et al., Sci. Adv. 2020; 6 : eaba881; Shaban et al. Genome Biology (2020) 21:95). In addition, FRET is a readout of interactions /proximities occurring locally between 1-10 nm. So, in others words, dense chromatin state at the mesoscale might not necessary imply local compacted chromatin measured by FRET resolution. The authors might consider rewriting their statements between density and compaction of chromatin.

We are very grateful to the Reviewer for this insightful comment. We completely agree with the Reviewer that ‘compaction and ‘density’ cannot be used interchangeably to describe the condensation state of chromatin at different scales. Accordingly, we now use the term ‘compaction’ throughout the revised manuscript, since this more accurately portrays the actual physical property measured by FRET-FISH (proximity between DNA sequences).

Minor concerns:

1) In Figure 1a, why “No FRET” is referred when Donor and Acceptor are separated by $>R_0$? R_0 is the distance between a D-A FRET pair when the FRET efficiency is 50%, so FRET can also occur if the distance $r > R_0$ but $< 90-100 \text{ \AA}$. For example, for CY3-CY5 FRET pair $R_0 = 52.7 \text{ \AA}$, leaving plenty of room for FRET above the R_0 . In this regard, l 116- l 117 should be rephrased.

The R0 of AF488/AF594 should be added.

We apologize for this imprecision. The Reviewer is absolutely correct and, accordingly, we have now corrected the text and the figure.

2) In Supplementary Fig1b, the condition D (alone) for design 3 is missing in the graph.

Unfortunately, this sample got damaged during the experiment (the glass in the corresponding chamber in the multi-chambered coverslip accidentally broke during cell washes). However, since D-only controls consistently yielded low FRET-FISH scores in other experiments, we decided not to repeat the entire experiment and only show A-only and D+A for this design. We hope that the Reviewer will find this reasonable.

3) In Supplementary Fig2, how #Acceptor dots values for G4 conditions (groups of 4 D and 4A) are smaller or equal to #Acceptor dots values for G2 conditions (groups of 2D and 2A) (see S2i and S2l panels for example). Is the # Acceptor dots value not reflecting the amount of fluorescent Acceptor molecules? Authors should clarify this point?

We apologize for not being clearer. The y-axis in the plots now shown in **Supplementary Fig. 2b** represents the total number of events (i.e., fluorescence dots) detected for every intensity bin (x-axis) across many individual cells (n values). Therefore, this number is unrelated to how many A oligos are present in the probes (which instead affects the signal intensity).

4) In Supplementary Fig4 a-r, it appears that a certain variability in the FRET % distribution is observed between the three experiments at each targeted locus. The authors could compare the FRET distributions (two by two) using a quantile-quantile (Q-Q) plot statistical analysis to see how variable are the FRET measurements between experiments.

Following the Reviewer's suggestion, we have now compared the FRET-FISH score distributions using Q-Q plots. As it can be seen in the **new Supplementary Fig. 5c and 9b**, the inter-experiment variability was overall low.

5) To illustrate some of the FRET results in the study, it would be helpful to show some microscopy FRET images next to them.

We thank the Reviewer for this suggestion. Accordingly, we have now added representative FRET-FISH images in the **new Fig. 1d and 3a** and in the **new Supplementary Fig. 8a**.

Reviewer #2

Ana Mota et al. in the submitted manuscript “FRET-FISH probes chromatin compaction at individual genomic loci in single cells” report the use of FRET between FISH probes to measure chromatin compactization. Sequence specificity is a clear advantage of FRET-FISH method. They highlight this capability by examining chromatin compactization of the genes located in X chromosome. In addition, the authors demonstrate the application of FRET-FISH to measure compactization of chromatin in ATP depleted cells. In general, FRET-FISH should be applicable to study chromatin compactization under various conditions and multiple cell types.

Overall, the manuscript is interesting and represents a novel application of FRET-FISH method, but lacks some important experiments and corrections. Thus, I am happy to recommend this manuscript to be accepted for publishing in *Nature Communications* after a major revision:

We are very grateful to the Reviewer for appreciating our work and being supportive of its publication in *Nature Communications*. Following all the Reviewers' insightful comments and suggestions, we have now generated more data and performed more in-depth analyses aiming at further highlighting the performance of FRET-FISH and strengthening the validity of our claims. We hope the Reviewer will appreciate our efforts and find the revised manuscript stronger.

Major points:

1. The downside of all FISH-based methods is distortion of the local DNA structure by hybridization with probe oligonucleotide. The authors are not addressing this issue in the manuscript and assume that the arrangement of chromatin stays intact after hybridization. I strongly recommend including these statements in the manuscript text. The authors should discuss how this might influence FRET measurements. This discussion should be supported by appropriate control experiments.

We thank the Reviewer for raising this important issue. As we wrote above in response to a similar comment by Reviewer #1, we are well aware of the possible effects of the denaturation conditions used in DNA FISH on the global and local chromatin structure. Over the years we have strived to optimize our protocol to minimize the number of steps that can potentially disrupt native chromatin conformation. Indeed, we have managed to minimize the time of DNA denaturation as well as the temperature of denaturation. To our knowledge, our protocol features the mildest denaturation conditions among published oligo-based DNA FISH protocols that rely on denaturation (see for example PMID: 30361340; 33619390; 34591592; 33505024; 32822575; 32719531). While typical DNA FISH protocols recommend denaturing the sample for 3–5 min at 80-90 °C, we have found that 1-2 min at 75 °C is typically sufficient to yield a satisfactory signal. In all our DNA FISH experiments (including FRET-FISH), we also make sure to proceed to hybridization right after cell fixation to avoid sample deterioration caused by storage, as we observed this can have a negative effect on the quality of FISH signals and, more generally, on nuclear morphology.

Despite this concern, previous studies comparing results obtained by DNA FISH with those obtained by CRIPSR/Cas9-assisted detection of DNA loci, including through live-imaging, showed a very good concordance between the methods (see for example PMID: 28355536 and 27222091). Moreover, in our study, we detected a high correlation between FRET-FISH and ATAC-seq or Hi-C, which further suggests that no major structural changes are introduced during the denaturation procedure in our FRET-FISH protocol, at least at the length scale we tested.

We now provide a step-by-step FRET-FISH protocol in the Supplementary Methods section in the **revised Supplementary Information**. In the same section, we have also added a step-by-step protocol for producing FRET-FISH probes.

2. FRET efficiency calculation is very primitive and sensitive to experimental conditions setup. The obtained value might be influenced by local environment of the fluorescent dyes which is reflected by relatively broad distributions of FRET efficiency. The authors also write in the supplementary information "...the FRET signal can be affected by multiple effects including cross-excitation, bleed-through and crosstalk, to minimize inter-sample variability we performed all our FRET-FISH experiments using custom-made multi-chambered coverslips." More accurate measurement could be performed using acceptor photobleaching and/or FLIM methods. This is essential for the described method and should be performed in order to convince readers about method reproducibility.

We agree with the Reviewer that we opted for a rather simplistic way of calculating the FRET efficiency in our experiments. Since throughout our FRET-FISH experiments we observed very little variation in the FRET channel intensity when imaging control samples in which we hybridized only A or D oligos, we considered the ratio between the intensity in the FRET channel and the sum of the intensities in the FRET and D channels to represent a good proxy of FRET efficiency. However, as we agree with the Reviewer that the actual FRET efficiency depends on more parameters, which we cannot control for in our experiments, we now refer to the aforementioned ratio as 'FRET-FISH score' and we use this term throughout the revised manuscript and figures. We hope the Reviewer will be satisfied with this change but we remain open to further suggestions on alternative terms. Of note, by relying solely on the signal in the A channel we could not obtain the same results as with FRET-FISH, as we now show in the **new Supplementary Fig. 2b** and in the **new Supplementary Fig. 6a**. These are important new results, which further corroborate the need for combining DNA FISH with a FRET readout to assess chromatin compaction at the nanoscale.

Regarding the Reviewer's suggestion of using FLIM, we have now performed a pilot experiment using our *Magix* probe, which also yielded a bimodal distribution as when performing intensity-based FRET measurements. The results of this experiment are shown in the **new Supplementary Fig. 6b**. We note, however, that while using our standard FRET-FISH approach we could measure more than 2,400 FRET signals in a relatively short time, in the case of FLIM the throughput was two orders of magnitude lower ($n=30$), because not all focal planes get equally bleached and the same field of view requires several steps of

bleaching at different z-stacks. Moreover, FLIM requires a dedicated equipment, which is not available to most research labs (in fact, we needed to outsource our experiments to an external imaging facility). We therefore hope the Reviewer will agree that, despite not providing the best conditions for measuring FRET, our microscope setup is more user-friendly and more likely to be adopted by other laboratories.

3. Why authors measured FRET efficiency only in non-G1 and G1 cells? Can other cell cycle stages be resolved? If not, please provided detailed explanation. It would be extremely interesting to measure FRET efficiency in M Phase.

Indeed, we have previously measured FRET-FISH signals also in mitotic cells which, as expected, were dramatically higher compared to non-mitotic cells (see **Supplementary Fig. 6b** in the original manuscript). The reason why in the main text and figures we made a simple distinction between G1 and non-G1 cells is because we only relied on the intensity of DNA staining by Hoechst 33342 to classify cells in different cell cycle phases and could not reliably distinguish between G1, S and G2/M cells. However, we believe that distinguishing between G1 and non-G1 cells is sufficient for the analysis we aimed at presenting in this work. We also hope the Reviewer will find the analysis we performed on mitotic cells interesting.

Minor points:

1. The authors write, "...suggesting that Hi-C might be better suited than ATAC-seq to probe chromatin density genome wide." Please explain this statement in more details. Is it only based on the obtained correlations or supported by the unmentioned experiments?

We have now removed the comparison between FRET-FISH and Hi-C and therefore no longer include this sentence. We decided to remove the comparison with Hi-C from our revised manuscript because we obtained confusing results when including FRET-FISH data obtained with the newly designed chr18 probes. As it can be seen in the plots below, while we find a strong anti-correlation between FRET-FISH and Hi-C in the case of probes targeting chrX loci, we see the opposite for probes targeting chr18 loci. Since we have found these results rather confusing and difficult to interpret at the moment, we have decided not to include them in the manuscript. Moreover, by performing a more detailed analysis of public Hi-C data from MEFs (<https://www.ncbi.nlm.nih.gov/geo/query/acc.cgi?acc=GSE76479>), we realized that there is a large variability between the three available replicates and hence we would prefer not to rely on these data. We hope that the Reviewer will find our choice justified and the corresponding parts in the manuscript clearer.

Plot legend: FFS, FRET-FISH score (%). PCC, Pearson's correlation coefficient. SCC, Spearman's correlation coefficient. Dashed red lines, linear regression fit.

2. The authors state, "To further validate FRET-FISH, we examined whether our method can also distinguish between different chromatin density states along the nuclear radius." Why you need further validation if the previous paragraph stated that it is a valid method?

We thank the Reviewer for spotting this inconsistency. Our analysis of how local compaction (measured by FRET-FISH) varies in relation to the radial distance of a locus from the nuclear lamina was mainly done with an explorative intent. However, based on the available literature we expected that loci located more internally would display less compaction, hence potentially serving as further technical validation/confirmation. In the revised manuscript, we now present these results in the frame of an exploratory effort and strive to more clearly distinguish between experiments/analyses performed to technically validate our new method from experiments/analyses conducted with explorative intent. We hope that the Reviewer will find the revised manuscript structure clearer and easier to follow.

3. The authors measured chromatin condensation in MEFs after 10 additional passages. It would be informative to describe how the first passage of MEFs is determined? Is it counted from purchase, are the passages done by provider company considered?

We started counting the passage number from the frozen vial that we originally obtained from ATCC and put in culture in our lab. We have now included this information in the **revised Methods** section.

4. The authors bravely state, "...FRET-FISH could also be applied to study enhancer-promoter contacts or chromatin loop organization in single cells, without the need to rely on super-resolution microscopy techniques to bypass the inherent spatial resolution limitations of DNA FISH." However, it not only microscope resolution is the limiting factor of FISH-based method. Even more important is the size of FISH probes itself. Please provide the comparison of FISH probe size and achievable resolution of the microscopes.

We agree with the Reviewer that this is a bold statement given that we have no data supporting it. We reasoned that FRET-FISH might be able to detect contacts between enhancers (E) and promoters (P) by targeting them with short probes. However, the design of FRET-FISH probes will require further optimization before our method can be effectively used to study E-P contacts. Accordingly, we have now added the following sentence in the corresponding paragraph in the **revised Discussion**:

<<This would require further optimization to the FISH protocol in order to achieve high sensitivity of detection with small number of oligos per probe.>>

However, if the Reviewer thinks it would be better to avoid this part, we are open to follow their suggestion.

5. The authors mention on line 489 "We acquired images in the donor and acceptor fluorescent channel separately, and only considered FISH dots with similar x, y, z coordinates (radial threshold: 7 pixels) between the donor and acceptor channels". Distance in pixels is not informative, please write in micrometres and explain why such distance is selected.

We apologize for this imprecision. The way we proceeded to choose this threshold was first to analyze the data manually and visually inspect the distribution of pairwise 3D distances between D and A signals. As it can be seen in the plots below, most of A and D FISH dots were separated by less than 7 voxels corresponding to 1.9 μm . We therefore used this value as threshold for automatically identifying co-localized D and A signals in all our analyses. We have now substituted the values in voxels to micrometers throughout the figures.

6. The authors state in the supplementary information, “we prepared samples and hybridized them with FRET-FISH probes using a modified 3D DNA FISH protocol” What are these modifications?

We apologize with the Reviewer for not having described the FRET-FISH probe preparation and hybridization in detail. We have now added a step-by-step FRET-FISH protocol in the Supplementary Methods section in the **revised Supplementary Information**, including a step-by-step protocol for producing FRET-FISH probes.

7. ATP depletion experiment done at quite strange conditions by adding 2-Deoxyglucose and Sodium azide directly to growth medium which contains glucose. Only partial depletion could be achieved because of 2-Deoxyglucose and glucose competition. Why authors have not measured ATP level and instead tested amount of nascent RNA in the nucleus with the Click-iT RNA Imaging Kit?

We chose ATP depletion as this was used in a previous study that leveraged FLIM-FRET on histone fusion proteins to study global chromatin compaction in living cells (PMID: 19948497). In the same study, the Authors monitored ATP depletion indirectly using the Click-iT RNA Imaging kit. Therefore, for consistency, we tried to reproduce the same conditions and approach in our study. In fact, following a suggestion on the same topic by Reviewer #1, we have now applied an even simpler approach to monitor whether ATP depletion indeed leads to global chromatin condensation, by measuring nuclear size and nuclear intensity of the DNA stain, Hoechst 33342. As we now show in the **new Supplementary Fig. 7a-d**, addition of 2-Deoxyglucose and Sodium azide to the growth medium prior to cell fixation leads to a significant reduction in nuclear size and higher Hoechst intensity per pixel, strongly indicating global chromatin condensation.

Reviewer #3

The authors of this work propose a novel approach, FRET-FISH, which combines the specificity of locus-specific FISH with the sensitivity of FRET to assay chromatin compaction and measure precise (local) differences in chromatin compaction in a locus-specific manner at the single-cell level. This approach addresses the lack of tools assessing chromatin compaction or differences in chromatin density in the field of genome organization.

The authors of this work demonstrate the following:

- 1- Chromatin density can be measured by comparing FRET-FISH efficiency, reproducibly.
- 2- The chromatin density measured can be compared to a) chromatin accessibility measured by ATAC-seq, and b) chromosome contact frequencies of the same loci measured by Hi-C.
- 3- FRET-FISH can detect changes in local chromatin compaction after chemical-induction of chromatin compaction.
- 4- FRET-FISH can detect differences in local chromatin compaction at different passage numbers as well as in G1 vs non-G1 cells. This final point emphasizes the importance of controlling for cell passages, tissue culturing techniques as well as cell cycle effects when making different chromatin biophysical measurements.

Overall, this is a proof-of-principle study that shows that FRET-FISH is an invaluable tool that can be complementary to other technologies that assess genome organization, cell-to-cell variability, and differences in chromatin density measurements. I think this manuscript is appropriate for publication in *Nature Communications* with more in-depth analysis of the imaging data, some clarification of the current data addressing inter-allelic differences, and after addressing the comments below.

*We are very grateful to the Reviewer for valuing our work and being supportive of its publication in *Nature Communications*. Following all the Reviewers' insightful comments and suggestions, we have now performed more in-depth analyses and conducted more experiments aiming at further highlighting the performance of FRET-FISH and strengthening the validity of our claims. We hope that the Reviewer will appreciate our efforts and find the revised manuscript stronger.*

Major comments:

1. There should be a figure or a clearer outline explaining the rationale for choosing the genomic size of the targets and number of oligos used in this study. The first two sections discussing implementation and optimization of FRET-FISH clearly describe the three FRET-FISH designs as well as the rationale for using 6 loci on the X chromosome. However, since this is the first study to introduce FRET-FISH as a tool to assess chromatin density, it would be beneficial to the field to show a clearer assessment of FRET-FISH efficiency for different ranges of number of oligos used and different genomic sizes targeted. This could be shown with preliminary data that was used for experimental optimization and could address the following in the text:

- In122 and In136: first test experiment targets 20 kb *Myc* gene locus and 134 primary D-A oligo pairs, what is the genomic size and # D-A oligo pairs for the second test at the *Ogt* locus, In181? What were the criteria for picking genomic size and #oligos?
- In338: in terms of experimental design for FRET-FISH: what was the lowest/highest #oligos tested?
- In435: what are the range of sizes of genomic regions assessed?

We greatly appreciate these comments as we always strive to make the methods that we develop as accessible and easy to reproduce by others as possible. In our initial proof-of-principle experiments, we designed the *Myc* probe based on our extensive experience with iFISH probes (PMID: 30967549), which are typically composed of 96 oligos and target a locus of ~8 kb. We found that this number of oligos and probe size/span (i.e., the genomic distance from the first to the last oligo in a probe) is a good compromise between resolution (i.e., the minimum size of a locus that can be detected) and sensitivity (i.e., how many cells display the expected number of iFISH dots). We reasoned that, for FRET-FISH, we might need to increase the number of donor (D) and acceptor (A) oligos per probe, since a sufficient number of D and A oligos need to be simultaneously bound to their target region in order for FRET to occur. Therefore, we slightly increased the number of D and A oligos included in our first test *Myc* probe, having ~130 oligos for each fluorophore type. For the other FRET-FISH probes described in our study, we opted for a different design aiming at increasing the dynamic range of the technique by spreading D and A oligos more apart. Given that such probes will result in having less dyes per volume unit, we decided to further increase the number of oligos (to ~200 per channel) limited by how many unique oligos we could find along a given gene and by how homogeneously spread they were. Inspired by the Reviewer's comment, we have now tested smaller FRET-FISH probes, comparing the standard 200 D-A oligo pairs per probe with 150 and 100 D-A oligo pairs per probe. As shown in the **new Supplementary Fig. 4d and e**, already reducing the number of oligos to 150 leads to a loss of sensitivity, given that the corresponding FRET-FISH score distributions appear unimodal and overlap with the higher mode in the bimodal distribution of the standard FRET-FISH probes.

In our **revised Supplementary Information**, we have now added a **Supplementary Table 2** containing information about the probe span and the number of D and A oligos per probe for each of the 6 probes on chrX that we previously described, as well as for 6 new probes targeting different gene loci on chr18, which we designed and tested following the request of this Reviewer (see next comment) as well as of Reviewer #1. In the **new Supplementary Fig. 2a**, we also show a scheme of the oligo distribution for each of the 6 *Ogt* probe designs that we tested. We hope that the Reviewer will appreciate these amendments and that the information we provide will facilitate other researchers to successfully implement and apply FRET-FISH for their purposes.

- Are there any preliminary tests looking at loci on autosomes? How does that compare to X chromosome data?

We thank the Reviewer for this question. Indeed, following a similar remark by Reviewer #1, we have now designed 6 additional FRET-FISH probes targeting 6 loci on chr18, including 3 loci embedded in constitutive lamina associating domains (LADs) and 3 loci inside inter-LAD (iLAD) regions. Importantly, these probes are very similar in span and number of D-A oligo pairs as those previously described for chrX (see new **new Supplementary Table 2**), allowing a fair comparison between them. As shown in the **new Fig. 3b and c**, the FRET-FISH score distributions are again bimodal, for all the six genes.

2. There should be a clearer outline and description of the findings for the inter-allelic differences. One of the more significant findings of the study is the potential ability for FRET-FISH to differentiate between alleles. This point is brought up several times in the text (ln53, ln109, ln261, ln297, ln319, ln405) but this data and its interpretation is not always clearly set up or described in the text or the figures (except for Figure 3e-j, n-s, and some supplementary figures). It seems that the bimodal FRET efficiency graphs representing different populations of compaction are not meant to imply, for example, bimodal distributions within single cells. If so, the authors are encouraged to make this explicit in the text. Several other examples where the text and data can be more explicit:

We thank the Reviewer for this comment, which made us realize that we were not fully correct in stating that our FRET-FISH data have single-allele resolution. The current FRET-FISH probe design allows to measure chromatin compaction on the two homologue chromosomes for any locus of interest at the level of individual cells. Hence, to be precise, our assay can only distinguish between inter-homologue differences, without however being able to discriminate between the paternal or maternal chromosome. To be able to distinguish between maternal and paternal alleles, one would need to design the oligos in the FRET-FISH probes so that they can discriminate between allele-specific single-nucleotide polymorphisms (SNPs). SNP-specific probes have been previously used to achieve allele-specific single-molecule FISH (see PMID: 23934076 and 23913259). However, adapting the same strategy for FRET-FISH would be very challenging and certainly this was not our scope when we started developing FRET-FISH. Throughout the revised manuscript we now use the adjective 'inter-homologue' instead of 'inter-allelic'.

Motivated by the Reviewer's comment, we have nevertheless examined whether the level of compaction significantly differs between homologous loci, by calculating the delta FRET-FISH score between the two homologues in each nucleus. As shown in the **new Supplementary Fig. 5a**, the distributions of the delta FRET-FISH scores* for all the 6 genes on chrX are broadly distributed, suggesting the existence of large inter-homologue variability in local chromatin compaction. We also compared the FRET-FISH score between the two homologues in the same cell and observed a moderate positive correlation as shown in the **new Supplementary Fig. 5b**.

*Please note that, following a remark by Reviewer #1, in the revised manuscript and figures we have now substituted the expression 'FRET efficiency' with 'FRET-FISH score' in order to

avoid confusion over the term 'efficiency' which is strictly related to distance measurements between donor and acceptor dyes, which our assay does not provide.

Regarding the bimodality observed in the FRET-FISH score distributions, these distributions come from single-locus data points pooled across cells. However, for the reasons stated above, we are unable to distinguish between paternal and maternal alleles and therefore cannot rule out whether the two modes in the distributions would correspond to the paternal and maternal allele. However, we think this is unlikely because we have shown that the relative size of the two modes in the distributions is influenced by various parameters, such as the cell cycle phase, global chromatin condensation, distance to lamina, and cell culture passage number. Therefore, our current working hypothesis is that the bimodality in the FRET-FISH score distributions represents different states of local compaction that is influenced by both cell-intrinsic (nuclear position, cell cycle phase, passage number) and cell-extrinsic (external stimuli affecting global chromatin condensation) factors.

- In188, Fig1d-i, how can the authors be sure the bimodal curve is not due to differences in the cell-cycle stage?

As we also replied to a similar comment raised by Reviewer #1, we performed all the analyses described in our manuscript on G1 cells (except for the part in which we specifically focused on the influence of the cell cycle on the FRET-FISH score). Hence, the observed bimodality in the FRET-FISH score distributions is not a simple consequence of working with an asynchronous cell population, even though the cell cycle phase influences the shape of the FRET-FISH score distributions as shown in the **revised Fig. 4a-f**. In fact, as already discussed in reply to the previous comment of this Reviewer, the bimodal shape of the FRET-FISH score distributions is probably determined by multiple inter-playing variables, including cell cycle phase, distance to the nuclear lamina, and cell age. As more factors influencing local compaction likely exist, we believe that FRET-FISH represents a powerful tool that can be harnessed to investigate this important feature of chromatin at the nanoscale.

- In213, Fig. 2a-f and Supplementary Fig. 4a-r, what accounts for the “two distinct chromatin states”? Can these datasets be split by cell cycle and by alleles?

Please see our two replies above.

- Fig2 j-l: are the different alleles combined for these datasets? The data should be clearly described to justify the statement emphasizing single-allele resolution in In261.

As discussed above in response to the Reviewer's comment #2, although our assay is intrinsically built with single-locus resolution, we are unable, at this stage, to distinguish between maternal and paternal alleles. Therefore, all the FRET-FISH score distributions shown come from pooling single-locus measurements performed in hundreds of nuclei, without distinguishing between alleles. However, as mentioned above, we have now calculated the difference in FRET-FISH score between the two homologues in each nucleus and present the

results of this analysis in the **new Supplementary Fig. 5a, b**. Additionally, we have observed that the more peripherally located locus in a given cell is typically associated with a higher FRET-FISH score (see **new Supplementary Fig. 10d**). Finally, in response to comment #3 by the Reviewer, we have now performed a new set of experiments in which we combined FRET-FISH with single-molecule RNA FISH (smFISH) for *Xist*, allowing us to unambiguously distinguish between the active (Xa) and inactive (Xi) chrX copy. As shown in the **new Supplementary Fig. 8**, for one escapee (*Kdm5c*) and one non-escapee (*Magix*) gene that we examined, the FRET-FISH score distribution did not significantly differ between the active and inactive chrX copy, even though, especially for the non-escapee gene *Magix*, we could observe the expected trend of the inactive allele being associated with higher FRET-FISH score values.

- For Fig3 e-j supplemental Fig5 h-m, are the datasets separated by cell cycle phase? If not, how can the authors definitively say the inflection point in the curve is due to differences in the level of compaction between the alleles? Can the authors clarify and comment on that?

Please see our response above to this Reviewer's remark: - In188, Fig1d-i, how can the authors be sure the bimodal curve is not due to differences in the cell-cycle stage?

3. Again, one of the powerful findings in this study is the ability to detect differences in FRET-FISH efficiencies within a population of cells for one locus. I found that there is no clear description of the cell-to-cell variability when interpreting the data. For example, in In186 and in Fig1d,e,f: do the different peaks represent different phases of the cell cycles? Different alleles (active vs inactive X)? Other differences within a cell population? These points should be clearly distinguished while interpreting the data and perhaps segueing to the following set of experiments addressing cell cycle phases.

As we have already mentioned in our responses to the previous comments by the Reviewer, we have now performed a new set of experiments and analyses aiming at better understanding cell-to-cell and inter-homologue variability in local chromatin compaction. Specifically:

- We have assessed inter-homologue differences in compaction by calculating the delta FRET-FISH score per nucleus and found that there is considerable variability in compaction between homologues at the level of individual cells (see **new Supplementary Fig. 5a**).
- We have plotted the FRET-FISH score of the two homologues in the same cell and observed a moderate positive correlation (see **new Supplementary Fig. 5b**).
- We have found that the more peripherally located locus in a given cell typically shows a higher FRET-FISH score (see **new Supplementary Fig. 10d**).
- We have combined FRET-FISH for *Magix* and *Kdm5c* with smFISH for *Xist* and found the FRET-FISH score distribution of the inactive *Magix* homologue to be shifted towards higher FRET-FISH score values compared to the active homologue, even though the difference did not reach statistical significance (see **new Supplementary Fig. 8**).

We believe that this new set of experiments and analyses further highlights the ability of our new assay to capture so far unexplored aspects of chromatin compaction at the level of individual loci and cells, paving the way to future studies leveraging FRET-FISH to investigate compaction in different cell types and conditions.

4. After looking through the methods, I found that the measurements were done using max projection and analyzed in 2D: e.g., "...2D nuclear segmentation mask of the cell in which the dot was identified" and "...max projection in the axial direction". This piece could be misleading especially when addressing the location of loci tested within the radial concentric layers within the nucleus (e.g. In255, Fig2f-j). How can the authors be sure the dot is not at the nuclear periphery in the axial position? Although I would recommend doing all the analysis in 3D, one way to acceptable way to address this could be to make the measurements in 3D for 2 loci or datasets and showing it still agrees with the max projection analysis shown in the rest of the figures.

We thank the Reviewer for raising this important issue and fully agree that our analysis should have been performed in 3D. Following the Reviewer's suggestion and a similar remark by Reviewer #2, we have now repeated our analysis of the relationship between compaction and radial distance in 3D for the 6 loci on chrX as well as for the newly analysed 6 loci on chr18 (please see reply to the Reviewer's comment #1 above), showing even more clearly how the two modes in the FRET-FISH score distributions change depending on the radial distance of the locus from the nuclear lamina (please see **new Fig. 3d and new Supplementary Fig. 10c**). We also provide a description of the tools we used for 3D nuclei segmentation in the **revised Methods section**.

5. There should be a clear assessment and/or discussion of how FRET-FISH efficiency can be affected in the following contexts, and how these contexts could affect the interpretation of the data:

- within a more compact or less compact chromatin environment
- what role quenching could play in more compact chromatin contexts
- possibility of reduced probe hybridization due to an increase in compaction. (e.g. the lower accessibility in mitotic cells)

As also discussed in response to a remark of Reviewer #2 on FRET efficiency, we opted for a rather simplistic way of calculating the FRET efficiency in our experiments. Since throughout our FRET-FISH experiments we observed very little variation in the FRET channel intensity when imaging control samples in which we hybridized only A or D oligos, we considered the ratio between the intensity in the FRET channel and the sum of the intensities in the FRET and D channels to represent a good of FRET efficiency. However, we acknowledge that the actual FRET efficiency depends on more parameters, which we cannot control for in our experiments. Therefore, we have now removed the term 'FRET efficiency' throughout the revised manuscript and figures and instead use the term 'FRET-FISH score', which we believe is more appropriate.

Regarding the possible influence of different chromatin environments on the FRET signal measured, the following lines of evidence make us confident that FRET-FISH is a robust assay that can probe compaction in very different chromatin contexts:

- We have assessed the intensity (per pixel) of the A probe targeting the *Magix* locus in mitotic chromosomes, which have the most compact form of chromatin. Not only we could easily detect FISH signals in mitotic chromosomes but, as expected, the intensity of those signals was much higher than those coming from cells in the G1 phase of the cell cycle, indicating that our probes and hybridization protocol work efficiently even in the most inaccessible DNA regions (see the **new Supplementary Fig. 11f**). These results also indicate that quenching is not an issue even in highly compacted mitotic chromatin, especially when using Alexa Fluor dyes, which are notoriously less prone to quenching compared to cyanide fluorophores.
- We have analyzed the correlation of the intensity of FRET-FISH A signals with ATAC-seq. As shown in the **new Supplementary Fig. 6a**, the A intensity correlates poorly with ATAC-seq read counts in the corresponding genomic regions (Pearson's correlation coefficient: -0.18), suggesting that the oligo hybridization efficiency is independent of the compaction/accessibility of the targeted locus.
- Based on the results of the new experiment combining FRET-FISH with smFISH for *Xist* described above, we have assessed the intensity of the acceptor probes on the active (X_a) and inactive (X_i) homologue. As shown in the plots below, the distributions of intensities were highly similar between the two homologues, further indicating that our hybridization protocol works efficiently independently of the level of chromatin condensation.

- In344: has FRET-FISH efficiency been tested robustly at the same locus in different phases of the cell cycle?

As also discussed in response to a similar remark by Reviewer #1, we have indeed assessed the compaction of the six genes on chrX in different phases of the cell cycle (see the **new Fig. 4a-f** and **new Supplementary Fig. 11**). Based on these results, we conclude that the cell cycle is an important determinant of the observed FRET-FISH score distribution bimodality.

6. For the results on In247, revealing the detection of chromatin compaction changes along the nuclear radius, addressing the following questions can help elucidate the biological importance of these measurements and observations:

- How were the nuclei selected for each different concentric nuclear layer? Did the authors select nuclei specific for each nuclear layer?

Following the Reviewer's suggestion to repeat these analyses in 3D (please see response to comment #4 above), we have now performed automated 3D segmentation of cell nuclei stained with Hoechst 33342. To this end, we first deconvolved the Hoechst channel and then performed 3D segmentation of G1 nuclei in each field of view, using the `tiff_auto3dseg` script in the `pygpseq` Python3 package, which we previously described in our iFISH paper (PMID: 30967549). We then divided each segmented nucleus into four concentric layers of equal volume and assigned the detected loci to one of the four groups. We now describe this analysis in detail in the **revised Supplementary Methods** in the Supplementary Information and provide all the scripts for automated 3D nuclei segmentation in the following GitHub link: https://github.com/BiCroLab/FRET_FISH (see **Code Availability** in the revised manuscript).

- Are the two alleles within the same concentric layer? How often?

As discussed above (please see reply to the Reviewer's comment #3), we have now conducted a more in-depth analysis of inter-homologue differences and found that in 25–31% of the cells the two homologues were present in the same concentric layer as shown in the following plots:

We have not included these plots in the revised figures, as we thought this would make Supplementary Figures excessively long and since we found it difficult to incorporate this information in the current manuscript flow. However, if the Reviewer thinks that this should be included in our manuscript, we will be open to reconsider our decision.

- How does the location within the layer correlate with transcriptional state? Is the transcriptional state accounted for in the measurements at all? Can the authors show or comment on some of that expression data (even if its nascent transcript quantification)?

These are very good questions, which we would love to address. Unfortunately, we have not yet investigated the relationship between compaction, radial position, and expression since this would require performing FRET-FISH together with smFISH on the same cells. We are currently working on a protocol that would allow us to co-hybridize smFISH and FRET-FISH probes and detect their signals simultaneously and hope to be able to report the results of these efforts in a follow-up manuscript in the near future. We would like to point out that, even though we managed to combine FRET-FISH with smFISH for *Xist*, it is much more challenging to achieve the same with probes targeting the same small genomic regions.

- Were there other notable differences when assessing the loci within different nuclear layers?

As already discussed above, we have observed a clear influence of the radial location on the compaction state of the examined loci, with more centrally located loci displaying a clear tendency to be in a less compacted chromatin state (see new **Fig. 3d and Supplementary Fig. 10c**). Moreover, in our inter-homologue analysis, we could see that the more peripheral homologue in a given cell is typically associated with a higher FRET-FISH score (see **new Supplementary Fig. 10d**).

Minor comments:

- In35: lack of chromatin accessibility not “Chromatin accessibility is often used as a proxy for chromatin compaction...”

Given that both chromatin accessibility and compaction can be either high or low, we would prefer not to change this sentence in our revised manuscript. We hope that the Reviewer will agree that this sentence is more neutral.

- In47: compact not compacted: “...with peripheral loci being more compacted and central ones less compacted.”

Thank you, we have corrected all instances of ‘compacted’ to ‘compact’ in the revised manuscript.

- In56: “Three major types of chromatin have been described in the nucleus of mammalian cells...”. Two of the papers cited are *Drosophila* studies.

We apologize for this mistake and, accordingly, we have now corrected this part.

- In86 include a reference for FISH/bridge/multiplexing technologies e.g. Beliveau, Boettiger, Nir, Bintu et al 2017, or another comprehensive reference.

We thank the Reviewer for this suggestion. Accordingly, in the revised manuscript we have now added a reference to a comprehensive review on spatial genomics methods that we recently wrote (PMID: 35680466), which includes the reference mentioned by the Reviewer.

- With respect to iFISH, can the authors clarify or comment on how the probes used for this work are similar or different from Oligopaints (e.g. Beliveau et al 2015, Boettiger et al 2016)?

We developed iFISH aiming at providing genome-wide databases of pre-designed 40-mers and a set of free tools (www.ifish4u.org) that the community could use to easily design DNA FISH probes for visualizing any locus of interest in the human and mouse genome. We make a detailed comparison between the two approaches in our original manuscript describing iFISH (PMID: 30967549). Major differences between iFISH and Oligopaint include:

- 1) The parameters used for selecting oligos that constitute the final FISH probes (the resulting iFISH database of 40-mers covers a much larger portion of the human genome and more homogeneously compared to the database created with OligoMiner that is used for designing Oligopaint probes).
- 2) The orthogonal sequences used as flanking sequences to amplify the oligos and detect them with fluorescently labelled detection oligos.
- 3) Details of the FISH protocols using Oligopaints vs. iFISH probes.

- In229, fig 2g,h, and Supplementary Table2:

1. Are the reported FRET-FISH efficiencies averaged for the two peaks? If so, can they be split by cell cycle stage and allele to see how they compare to Hi-C and ATAC-seq data?

Indeed, the values that we used to compare FRET-FISH with ATAC-seq and Hi-C data were averaged over the two FRET modes, given that we have not yet identified conditions in which we can clearly separate them. As mentioned above, we have performed all our analyses on G1 cells (except for those in which we aimed at assessing the influence of the cell cycle phase on the bimodality of the FRET-FISH score distributions) and even when selecting for G1 cells we still observe bimodality. Therefore, we hope that the Reviewer will approve of our approach.

2. Are the reads for Hi-C contacts, normalized reads? The Hi-C contact frequency could be better assessed through the contact maps: contact frequency observed/expected, rather than mappable reads. Have the authors tried to get measurements for different loci directly from the normalized contact maps? This could clarify the discrepancy between ATAC-seq and Hi-C data, In239.

This is a very interesting point, however, as discussed in reply to a related comment by Reviewer #2, we have now removed the comparison between FRET-FISH and Hi-C from our revised manuscript since we obtained confusing results when including FRET-FISH data obtained with the newly designed chr18 probes (see above reply to the Reviewer's comment:

- Are there any preliminary tests looking at loci on autosomes? How does that compare to X chromosome data?

As it can be seen in the plots below, while we find a strong anti-correlation between FRET-FISH and Hi-C in the case of probes targeting chrX loci, we see the opposite for probes targeting chr18 loci. Since we have found these results rather confusing and difficult to interpret at the moment, we have decided not to include them in the manuscript. Moreover, by performing a more detailed analysis of public Hi-C data from MEFs (<https://www.ncbi.nlm.nih.gov/geo/query/acc.cgi?acc=GSE76479>), we realized that there is a large variability between the three available replicates and hence we would prefer not to rely on these data.

Plot legend: FFS, FRET-FISH score (%). PCC, Pearson's correlation coefficient. SCC, Spearman's correlation coefficient. Dashed red lines, linear regression fit.

Regarding the Reviewer's suggestion to work with normalized reads, we note that one would need to bin the data into same-sized bins, while to make a fair comparison between FRET-FISH and Hi-C we used genomic windows corresponding to the regions probed by FRET-FISH probes (therefore of varying sizes). While this is possible using raw read counts, it is not possible using normalized reads. However, as said, we have now removed the comparison with Hi-C from our revised manuscript not to provide a confusing message. We hope that the Reviewer will find this choice justified and the corresponding parts in the manuscript clearer.

- In230: The anti-correlation between FRET-FISH and Hi-C, not the correlation.

As mentioned above, we have now removed the comparison with Hi-C from the manuscript.

- In232: the conclusion at the *Ddx3x* locus is unclear, what are the authors trying to say about the differences in observations for *Ddx3x*?

We apologize for not having been clearer. What we meant was that, in the scatterplots comparing FRET-FISH and ATAC-seq scores, *Ddx3x* is rather distant from the regression line. However, we have removed this sentence from the new manuscript.

- In90: “Although it is assumed that chromatin accessibility and density are inversely correlated, the exact relationship between these two biophysical properties remains unknown...”. The authors bring up a very important point about the potential implication of having less dense chromatin that is inaccessible due to protein occupancy, but do not address or consider this point with any of their data. It would be great to bring this up while considering some of their observations in the discussion if it applies. In addition, can the authors comment on the inverse relationship between chromatin accessibility and chromatin compaction and how that may affect their FRET-FISH results?

Indeed, we believe that the relationship between chromatin accessibility and compaction has been rather overlooked and we hope that the Reviewer will appreciate our work in this direction. However, we are not sure how else we could investigate this further in the current work. The correlation between FRET-FISH and ATAC-seq, which we show in the **new Figure 2b** and in the **new Supplementary Fig. 9c**, is to our knowledge the first ever attempt to directly compare compaction and accessibility.

As already discussed in response to the Reviewer’s comment #5, we have now analyzed the correlation of the intensity of FRET-FISH A signals with ATAC-seq data. As shown in the **new Supplementary Fig. 6a**, the A intensity is weakly correlated with ATAC-seq read counts in the corresponding genomic regions (Pearson’s correlation coefficient: -0.18), suggesting that the oligo hybridization efficiency is independent of the compaction/accessibility of the targeted locus.

- Figure 1c: can the authors comment on why the donor violin plots in all three designs has such a wide distribution for FRET efficiency?

The reason why the distributions of the donor (D) intensity are broader than the FRET-FISH score distributions is because of how FISH dots are picked in our image analysis pipeline. In the case of samples hybridized with both D and A oligos, the pipeline requires that two fluorescent dots—one in the A channel and one in the D channel—are colocalized in the same nucleus in order for a FRET signal to be called. Thanks to this stringent criterion, the accuracy of FISH dot picking in FRET-FISH experiments with both D and A oligos is high. However, for samples in which only D or only A oligos are hybridized, the pipeline can pick more unspecific

signals (i.e., off-targets), which explains the broader distribution of the D intensity in the case of control samples.

- Figure 1k: can the authors comment on why although there are two distinct FRET efficiency peaks, why is it that the overall efficiency of FRET-FISH <20% compared to the other fluor set?

As we already discussed in response to a similar remark by Reviewer #1, this difference can be explained by the lower influence of crosstalk and bleed-through in the FRET channel, in the case of Alexa Fluor dyes. As we now show in the **new Supplementary Fig. 3c**, the ratio of FRET intensity to D or A intensity is more favorable for Alexa Fluor dyes compared to cyanide dyes. Together with the fact that Alexa Fluor dyes suffer less from quenching effects and are more photostable, this is the reason why in the end we opted for the Alexa Fluor dyes.

- In307: Can the authors comment more clearly as to why *Magix* gene shows a higher FRET-FISH efficiency in mitotic cells but not in G2 cells? Would this be related to the possibility that replicating DNA is less compact, or could the proximity of an additional copy of DNA be quenching the FRET-FISH signal?

Higher DNA accessibility in G2 compared to G1 has previously been reported in the first study describing the ATACsee method (PMID: 27749837). In that study, the Authors reported two different G1 subpopulations, with one showing a particularly low DNA accessibility in comparison to G2 cells. In the same study, it was shown that the global DNA accessibility of G1 cells remained lower compared to G2 cells even when the two G1 subpopulations were pooled together. We suspect that the low-accessibility G1 subpopulation represents cells that recently exited mitosis and still have a highly compacted chromatin.

Regarding the possibility that quenching could explain the difference in FRET-FISH score between G1 and G2 cells, we do not think this is the case as otherwise we would expect to detect a much weaker FRET signal in mitotic cells—whereas we detected the highest FRET in mitotic cells. Therefore, we are confident that the observed difference in local compaction between G1 and G2 cells at the *Magix* locus is a true biological phenomenon and not an artefact.

The following two comments are OPTIONAL requests for preliminary data that I would like to see and believe would make the paper and the approach presented more impactful, but I understand if it's not feasible or if they cannot be addressed in a timely manner:

- The authors only survey loci on the sex chromosomes, it would be interesting to see how comparable those observations would be on an autosome by checking 1-2 other chromosomes or loci.

We are grateful to the Reviewer for this suggestion, which was also made by Reviewers #1 and #2. We have now designed 6 additional FRET-FISH probes targeting 6 loci on chr18, including 3 loci embedded in constitutive lamina associating domains (LADs) and 3 loci inside inter-LAD (iLAD) regions. Importantly, these probes are very similar in span and number of D-

A oligo pairs as those previously described for chrX, allowing a fair comparison between them (see the **new Supplementary Table 2**). As shown in the **new Fig. 3b** as well as in the **new Supplementary Fig. 9a**, for all six genes the FRET-FISH score distributions are again bimodal. Therefore, we conclude that the observed bimodality must be a more general phenomenon not limited to loci on chrX. As already discussed in response to Major remark #2 by this Reviewer, our current working hypothesis is that the bimodality in the FRET-FISH score distributions represents different states of local compaction that is influenced by both cell-intrinsic (nuclear position, cell cycle phase, passage number) and cell-extrinsic (external stimuli affecting global chromatin condensation) factors.

- Can the authors show different applications for the FRET-FISH? For example, are there any other preliminary results showing how the technology applicable to assess other biological questions: enhancer-promoter contacts or chromatin loop organization?

We thank the Reviewer for this excellent suggestion. Unfortunately, we have not yet had a chance to apply FRET-FISH to tackle these exciting questions since we first needed to fully convince ourselves and others of the validity of our method. However, this revision process has been extremely helpful in helping us building further trust in our technique and expanding our hands-on experience. Therefore, as soon as this proof-of-principle work is accomplished, we will be eager to pursue these exciting lines of investigation.

Reviewer #4

The paper from Mota et al describes a novel method for chromatin compaction assessment. Compaction of genomic loci is an important characteristic in respect to their transcriptional activity and, therefore, this paper is very relevant and timely for the chromosome/chromatin research field.

The authors developed an elegant method, so called FRET-FISH, combining the previously known oligo-FISH and FRET (fluorescence energy transfer), by targeting oligo-probes to a particular gene locus and detecting them with a certain combination of fluorophores serving as donors with a lower wavelength emission and acceptors with a higher wavelength emission. The rationale behind this strategy is that in compacted chromatin, donor and acceptor fluorophores occur in physical proximity, and thus the excitation of a donor results in an excitation of an acceptor via FRET. The beauty of the approach is that one can estimate compaction of specific genomic loci or even various parts of the same gene. I find this idea excellent and the method as very useful.

Therefore, the work is potentially valuable for the chromatin community and I am generally positive about it. However, I see several drawbacks, which in my view have to be attended by the authors before considering the manuscript for publication.

We thank the Reviewer for appreciating our work and recognizing the potential value of FRET-FISH for the community. We are also grateful to the Reviewer for their insightful and constructive comments, which helped us extend further strengthen our manuscript. We hope that the Reviewer will appreciate our efforts and find our revised manuscript now suitable for publication in *Nature Communications*.

1. Introduction

Lines 67-85. When mentioning DNA staining and fluorophores fused to histones, the authors could do a better job discussing nucleic acid stains other than DAPI, as well as genetically-encoded methods such as, e.g., HI-NESS (10.1093/nar/gkab993)

We thank the Reviewer for this suggestion. Accordingly, in the **revised Introduction**, we have strived to provide a more complete summary of nucleic acid stains and genetically encoded methods, including HI-NESS.

2. Results

(1) The authors describe in great detail several probe designs that were ultimately recognized as suboptimal for the method and were not used for FRET-FISH validation. Lines 114-196 are dedicated to description of 9 versions of oligo-probe spacing and detecting probe designs – all based on conjugation with Cy3 as a donor and Cy5 as an acceptor, which, at the end of the day, were not used. The rest of the experiments were performed with only one version of oligo-probe

spacing and using AF488 as a donor and AF594 as an acceptor. To focus the Results section, I suggest the authors to consider shortening the part about the optimizing experiments and moving the corresponding figures to Supplementary material.

We thank the Reviewer for this suggestion. However, also in line with the remarks of the other three Reviewers, we think it is important to showcase the extensive optimization work we did by testing different *Ogt* probe designs. Therefore, we would prefer to leave these results in Figure 1 and not to move them to Supplementary Figures.

In this respect, it is also important to show why the AF488/594 couple is superb to the Cy3/Cy5. For example, when comparing Fig 1j and SFig 3, it is not clear why there is a difference in the FRET channel in the upper two rows.

As we already discussed in reply to a similar remark by the other Reviewers, the lower FRET-FISH scores that we obtained using AF488-AF594 as compared to using Cy3-Cy5 can be explained by the lower influence of crosstalk and the bleed-through in the FRET channel in the case of Alexa Fluor dyes. As we now show in the **new Supplementary Fig. 3c**, the ratio of FRET intensity to D or A intensity is more favorable for Alexa Fluor dyes compared to cyanide dyes. Moreover, Alexa Fluor dyes suffer less from quenching effects and are more photostable, which is essential when imaging multiple fields of view.

(2) Lines 208-218: To validate the method, the authors chose 3 genes that are constitutively silenced on Xi chromosome and 3 genes escaping inactivation. They measure FRET efficiency for all these genes in MEFs and find no obvious difference between silenced and active genes. Very surprisingly, the authors do not distinguish between the Xa and Xi chromosomes, although this was the point of the whole experiment – to see the difference in gene compaction depending on their activity.

4 comments to this part:

(i) Why didn't the authors distinguish between Xa and Xi, e.g. by *Xist*-detection or immunostaining, as it is routinely used (10.1016/j.cell.2021.10.022; 10.1007/s00412-021-00754-z)? Since the authors use automated acquisition and the coordinates of nuclei are stored, this step could be done after FRET-FISH to avoid interference with other fluorochromes.

We thank the Reviewer for this excellent comment and suggestion. Accordingly, we have now performed a new set of experiments in which we combined FRET-FISH with single-molecule RNA FISH (smFISH) for *Xist*, allowing us to unambiguously distinguish between the active (Xa) and inactive (Xi) chrX copy in order to measure compaction separately for the active and the inactive homologue in the same cell, for *Magix* (non-escapee) and *Kdm5c* (escapee). These experiments proved to be rather challenging, since the combination of FRET-FISH and smFISH led to a substantial decrease in the intensities of the FRET signals detected and hence loss of sensitivity, possibly due to interference between the fluorescence dyes used, as rightfully anticipated by the Reviewer. Therefore, we could no longer observe bimodality in the FRET-FISH score distributions. The reason why we opted for this way of detecting *Xist* was

because our lab has a long-standing experience in performing combined RNA FISH with DNA FISH assays and we already had smFISH probes for *Xist* available in the lab. Hence, we leveraged our previously established protocol for combined DNA-RNA FISH (PMID: 23263692)—in which we first perform DNA FISH and then proceed to smFISH and detect all the signals simultaneously—to combine FRET-FISH with smFISH. Since we obtained weaker FRET signals than when using our standard FRET-FISH procedure, we reasoned that our analysis might benefit from image deconvolution to enhance the contrast and allow better detection of true FRET signals. To this end, we leveraged a powerful software for image deconvolution, which we recently developed in parallel to this work (see doi.org/10.21203/rs.3.rs-1303463/v1, currently under revision for *Nature Biotechnology*). Indeed, after deconvolution, we could again observe a bimodal distribution of the FRET-FISH score for both genes tested, even though not as clearly as when we performed FRET-FISH alone. This prompted us to compare the FRET-FISH score distributions between the active and the inactive homologues. As shown in the **new Supplementary Fig. 8**, we could observe the expected difference in the distribution of the FRET-FISH score between active and inactive homologues, especially for the non-escapee gene *Magix*. However, despite the expected trend, the difference in the distributions was not statistically significant. Importantly though, both active and the inactive homologues showed a broad distribution of the FRET-FISH score as for both homologues pooled together, and in the case of *Magix* the active homologue displayed a clear bimodality, suggesting to us that there are other factors affecting the observed FRET-FISH score distribution bimodality. Indeed, when we profiled multiple loci on an autosomal chromosome (chr18), we could observe a clear bimodality in the corresponding FRET-FISH score distributions (see **new Fig. 3b and c**), demonstrating that the distributions obtained for genes on chrX genes cannot be fully explained by the fact that this chromosome is present in two very different compaction states in female cells. In fact, as we show throughout the manuscript, the FRET-FISH score distribution bimodality represents different states of local compaction modulated by various parameters such as cell cycle phase, distance to the lamina, cell aging (perhaps due to accumulation of DNA damage) and possibly more. We believe that FRET-FISH is a powerful tool that opens up new lines of future investigation to fully understand and eventually predict the extent of chromatin compaction at any given genomic locus.

(ii) Xi identification is especially important in the view that during cultivation, MEFs quickly become tetraploid or even might have an aneuploidy for one or several chromosomes, including X. Have the authors checked for this?

We have now tested the ploidy of chrX in our MEFs line and if/how it changes upon prolonged culturing. As the Reviewer can see in the plots below, the number of chrX copies in MEFs cultured for more than 10 passages decreased compared to cells cultured for less than 10 passages.

Moreover, to better characterize the changes occurring in our model system, we investigated whether with prolonged culturing DNA damage accumulates. Indeed, immunostaining for two DNA damage markers—phosphorylated histone H2A.X and 53BP1—revealed an accumulation of DNA breaks in cells at higher passage number. This phenomenon has been previously associated with global heterochromatinization and increase in chromatin compaction (PMID: 33494932), in line with our observation that the bimodal FRET-FISH score distributions shift towards the highest mode in MEFs upon prolonged passaging. While we show these results in the **new Supplementary Fig. 12b and c**, we decided not to include the above plots on ploidy in our revised figures, since it is unclear for us how to relate them to the main message that we wanted to convey (i.e., that prolonged cell passaging leads to progressive increase in compaction, at least for the loci we probed with FRET-FISH) and therefore we would like to avoid confusing the readers. However, if the Reviewer feels that it is important to show these results, we will be happy to add them to our revised Supplementary Fig. 12.

(iii) Since this paper is mostly about interphase chromatin compaction, it is really surprising that there are no images of nuclei with FRET signals for each of the studied genes so that the reader could see the number and location of the signals.

We fully agree with the Reviewer that since this manuscript focuses on a new microscopy technique, it is important to show at least some representative images. Accordingly, we have now added representative FRET-FISH images in the **new Fig. 1d and 3a** and in the **new Supplementary Fig. 8a**.

(iv) In connection to this: there is no statistics on how many AF488, AF594 and FRET signals per nucleus were usually detected – in other words, what the reproducibility of FRET-FISH from nucleus to nucleus is.

In the **new Suppl. Fig. 4a and b** we now show the distributions of FISH dot counts, which are fully in line with the expected copy number of the targeted loci (i.e., most nuclei should have 2 signals while nuclei with 3-4 signals correspond to S and G2 phase cells, respectively). However, we note that our pipeline only calculates the FRET-FISH score for co-localized AF488 and AF594 dots within the same nucleus. Therefore, even in the infrequent case of AF488 and AF594 counts not matching (because of unspecific signals), the resulting FRET-FISH score distribution will not be affected.

I do appreciate the colossal work of collecting thousands of signals in each experiment shown in the paper and all the efforts the authors undertook to develop a completely automated pipeline for signal identification. However, in this case, more careful observations of fewer nuclei but with a more precise chromosome identification can be of a higher benefit.

Unfortunately, all the rest of the experiments in the paper that I discuss below - in (4), (5) and (6) – are based on the same genes and thus require Xa and Xi chromosome identification.

I am also wondering why the authors have chosen the X-chromosome but not genes with clear LAD and interLAD signatures, which do not require chromosome identification but are unmistakably different and have well defined chromatin status (e.g., 10.1101/gr.141028.112).

We are grateful to the Reviewer for appreciating our big effort and for the excellent suggestion to examine genes in LADs vs. iLADs. The reason why we initially decided to focus only on genes on chrX is that we expected to detect a clear difference in compaction between homologue loci on the Xa and Xi, serving as a clear validation of our new method. However, we now know that this is not the case, since the bimodality in the FRET-FISH score distributions is still present when distinguishing between the Xa and Xi homologues (please see our previous reply to the Reviewer's comment (i) above). As said, following the Reviewer's suggestion, we have now designed FRET-FISH probes for 6 loci on chr18, including 3 loci embedded in constitutive LADs (cLADs) and 3 loci inside inter-LAD (iLAD) regions. As shown in the **new Fig. 3b** and in the **new Supplementary Fig. 9a**, the FRET-FISH score distributions are again clearly bimodal, for all six genes examined. Based on these new observations, our current working hypothesis is that the FRET-FISH score bimodality represents two different states of local compaction, which are influenced by various parameters such as cell cycle phase, distance to the lamina, cell aging (and hence accumulation of DNA damage) and possibly more. We believe that FRET-FISH is a powerful tool that opens up new lines of future investigation to fully understand and eventually predict the extent of chromatin compaction at any given genomic locus.

(3) The authors show that FRET efficiency of the six selected probes for the X-chromosome strongly correlates with ATACseq and Hi-C read counts extracted from publically available databases on MEFs, which is a nice and convincing validation of the method. Furthermore, the authors attempt to show a similar correlation with nuclear radius (Lines 247-269). Indeed, it is well established that 3D radial distribution of chromatin in spherical nuclei is dictated by gene density and transcriptional activity, both being high towards the nuclear interior, which thus is supposedly filled with less condensed chromatin. However, surprisingly and illogically, the authors perform radial measurements of signal positioning (a) on 2D projections of segmented nuclei, (b) using MEFs with very flat nuclei (thus with a very small interior) and (c) using genes on the X-chromosome, which is known to be very flat and peripheral, especially in case of Xa (see e.g., 10.4161/nucl.2.5.17862; 10.1016/j.cell.2021.10.022). I suggest that the authors either do this evaluation properly in 3D and on another chromosome or completely remove this section from the manuscript.

We thank the Reviewer for this valuable comment and suggestion. Following the Reviewer's suggestion, we have now repeated this analysis in 3D and provide a description of the tools we used for 3D nuclei segmentation in the **revised Methods section**. As shown in the **new Supplementary Fig. 10c**, the inverse relationship between the FRET-FISH score and the distance to the nuclear lamina still clearly persists for all six genes examined on chrX. In addition, we have performed the same type of 3D analysis for the six genes on chr18 that we targeted following the Reviewers' suggestion to compare genes in LADs vs. iLADs (see comment above), confirming the observation that the FRET-FISH score is influenced by the distance of the locus examined from the nuclear lamina (see **new Fig. 3d**).

In principle, we agree with the Reviewer that these analyses could be even stronger if performed on cells with more rounded nuclei. However, we hope that the Reviewer will agree that our data already quite convincingly show that chromatin compaction and distance to periphery are related and, therefore, repeating these experiments on cells with different nuclear shape would be beyond the scope of this proof-of-principle work.

(4) Lines 270-27: The authors refer to SFig 5a as demonstrating a drastic reduction of transcriptional activity. However, the micrograph 5a shows only labeled nucleoli. To prove reduction in expression of the 4 studied genes, qPCR is the most direct and reliable method for assessment.

The goal of this experiment was to investigate whether a global increase in chromatin condensation is mirrored by an increase in local chromatin compaction at selected loci, and not to study the relationship between compaction and transcription. We monitored global transcription in response to ATP depletion since this was used as an indirect readout in a previous study that leveraged FLIM-FRET on histone fusion proteins to study global chromatin compaction in living cells (PMID: 19948497). Following a remark on the same topic by Reviewer #2, we have now applied an even simpler approach to monitor whether ATP depletion indeed leads to global chromatin condensation, by measuring nuclear size or nuclear intensity of the DNA stain, Hoechst 33342. As we now show in the **new Supplementary Fig. 7a-d**, addition of 2-Deoxyglucose and Sodium azide to the growth medium prior to cell fixation leads to a significant reduction in nuclear size and higher Hoechst intensity per pixel, strongly indicating global chromatin condensation.

(5) Lines 283-298: This is the vaguest part of the manuscript. The definition of the number of cell passages is very superficial and defined as <10 and >10. What happens with cells after the 10th passage? Why does chromatin become more condensed? Does it mean that fibroblasts become senescent? Or is the cell cycle changing? Or is there an increase in ploidy? In my view, it makes no sense to perform FRET measurements on a system so poorly defined.

We apologize for not having been clearer here. We counted passages from the time we put in culture the frozen vial of cells that we purchased from ATCC. Therefore, the actual number of passages that these cells have undergone is higher. We agree with the Reviewer that the threshold of 10 passages is arbitrary. However, we chose this since we observed that already

after 10 passages the FRET-FISH score reproducibly increased. As already described above in response to a related comment by this Reviewer, we have now more thoroughly characterized this phenomenon and, in the **new Supplementary Fig. 12b and c** show that cells passaged more than ten times in our hands start accumulating DNA damage as revealed by phosphorylated histone H2A.X and 53BP1 immunofluorescence. We do not know at this point what exactly causes the observed local increase in chromatin compaction detected by FRET-FISH, but we suspect that the accumulation of DNA damage might play a role by causing some form of global 'heterochromatinization' to prevent further damage and stabilize the existing damage to facilitate its repair. We hope that the Reviewer finds these new results interesting and worth being reported in our manuscript. The reason why we would like to keep this part in the manuscript is because we would like to inform potential future users of FRET-FISH to closely monitor passage number when performing FRET-FISH experiments on primary cell cultures.

(6) Lines 301-320: Finally, the authors show that FRET efficiency is higher in G1 cells in comparison to the rest of the cell cycle stages, using Hoechst 33342 staining as a criterion for G1 stage. Unless I have missed something, I have not found any evidence in the manuscript that cells with a high Hoechst 33342 staining are in G1 stage. I have no much experience with mouse fibroblasts, but for me it is not obvious that G1 cells have stronger Hoechst staining – e.g., what is with cells in prophase? The use of anti-Ki67 staining, the most common marker for G1 phase (e.g., 10.1023/a:1009210206855), would free the authors from unnecessary speculations about G1 and non-G1 cells, including Hoechst-low and Hoechst-high G1 subpopulations (Lines 317-318). Although this approach might reduce the number of analyzed loci, the work will gain more solid conclusions.

In our long experience working with DNA FISH, we have observed that simple staining of nuclei with DAPI or Hoechst 33342 allows to clearly discriminate between G1 and G2/M cells based on the distribution of the integrated DAPI/Hoechst intensity in 2D or (even better) in 3D segmented nuclei (please see **Supplementary Fig. 11a** for an example of such distribution). Depending on the cell type, this microscopy-based approach can also allow distinguishing cells in S phase (between the two major peaks in the DAPI/Hoechst intensity distributions), although this is usually more challenging. In our experiments with mouse cells, we were unable to clearly distinguish S phase cells and therefore we performed a gross classification in two main subpopulations (G1 and non-G1) and identified rare mitotic cells by eye. We agree with the Reviewer that this classification could be improved by staining for specific cell cycle phase markers or using genetically encoded fluorescent reporters, such as the FUCCI system. However, combining those assays with FRET-FISH is far from trivial (due, for instance, to fluorescent dyes incompatibilities) and therefore we hope that the Reviewer will agree that attempting such experiments would go beyond the scope of the present work.

3. Figures

(1) Why are the replicates for each FRET-FISH experiment not averaged in a single graph? As far as I could figure it out, graphs in Fig 1c and SFig 1b, Fig 3a-d and SFig 5c-f, Fig 3e-j and SFig

5h-m, Fig 3n-s and SFig 6c-h show two replicates. If the authors find that, in addition to the averaged graph in the main text, it is important to show both replicates separately, it can be done in Supplementary data.

We agree with the Reviewer that showing multiple replicates in the same figure makes it unnecessarily heavy. However, we think it is important to keep replicates separately, especially in the case of histograms and boxplots, in order to faithfully show that independent experiments yielded similar distributions. In the revised main figures, we therefore now show only one replicate in case of histograms or boxplots/violin plots while the other replicate(s) is shown separately in the corresponding Supplementary Figure.

(2) Multiple letters in figures make them unnecessarily overloaded. I see no need in such an ample labelling in figs 2 and 3 (as well as corresponding supplemental figures), because the names of all genes and conditions are conveniently and clearly marked above the graphs.

Following the Reviewer's suggestion, we have now removed single-plot labels when two or more plots refer to different conditions or genes in the same experiment (see, for example, the **new Fig. 2a and c-f**). The reason why we initially labeled each individual plot is that we strictly followed the figure formatting style of *Nature Communications*.

REVIEWERS' COMMENTS

Reviewer #1 (Remarks to the Author):

Reviewer #1: The authors have presented a clarified revised version of the paper that I feel sufficiently addresses all of the major and minor concerns I had about the initial submission.

In particular I found the new data on combined FRET-FISH and single-molecule FISH (smFISH) with a probe targeting Xist RNA (new Fig S8) to be very exciting although experimentally challenging. I really appreciate the new amount of data put in the revised version using FRET-FISH probes targeting regions inside cLADs and iLADs on chr18 (Fig. 3b and c) as well as the additional 3D FISH distance measurements at the ogt locus (new Supplementary Fig. 2d and e).

Reviewer #2 (Remarks to the Author):

Ana Mota et al. in the submitted revised manuscript "FRET-FISH probes chromatin compaction at individual genomic loci in single cells" have considerably improved the quality of the presented work. Thus, I am happy to recommend this manuscript to be accepted for publishing in Nature Communications after a minor revision:

Minor points:

1. The authors claim that their protocol features the mildest denaturation conditions among published oligo-based DNA FISH. I find this information very useful, and the reader should be presented with a summary of the reported protocols. This also would strengthen the manuscript. Please include a table summarizing hybridization conditions reported by other studies.

2. The authors write „...we only relied on the intensity of DNA staining by Hoechst 33342 to classify cells in different cell cycle phases and could not reliably distinguish between G1, S and G2/M cells." This statement is strange since several FACS protocols, which distinguish G1, S and G2/M stages, rely on Hoechst/DAPI staining. Besides this, M-phase can be reliably detected using fluorescence microscopy of Hoechst 33342 stained cells. Please revise this statement and correct the manuscript accordingly.

Reviewer #3 (Remarks to the Author):

The authors have satisfactorily responded to my comments. Therefore, I think this manuscript is appropriate for publication in Nature Communications with some minor clarification of the data and/or rephrasing some of the text in the manuscript for better clarity, see my comments below:

- In 46: “during prolonged cell culturing”, consider replacing with “increasing passage number”

- In 59: pericentromeric regions are in fact transcribed, see following reference for overview:
<https://www.frontiersin.org/articles/10.3389/fgene.2018.00674/full>

- In 97: “select loci” not selected loci.

- In 99: “...cells with increasing passage number, as well as in cells in different phases of the cell division cycle”

Rephrase to:

“...cells with increasing passage number as well as in different cell-cycle stage”

- In 100-101: “We conclude that FRET-FISH is a sensitive assay for...assessing cell-to-cell variability in compaction”

Consider rephrasing to “assessing different compaction states within a cell population” since this is what the paper really shows rather than cell-to-cell variation.

- In 113: In my first comment in the first round of review, I asked to provide a rationale for selecting the MYC gene locus, genomic size, # oligos used... etc. I think this is the place in the manuscript where the authors could showcase how carefully this experiment was designed and thought through while referring to Sup. Fig4 d,e and Sup. Table2 . Including a polished and concise version of your response to my first comment here:

“In our initial proof-of-principle experiments, we designed the Myc probe based on our extensive experience with iFISH probes (PMID: 30967549), which are typically composed of 96 oligos and target a locus of ~8 kb. We found that this number of oligos and probe size/span (i.e., the genomic distance

from the first to the last oligo in a probe) is a good compromise between resolution (i.e., the minimum size of a locus that can be detected) and sensitivity...”

- In 184-185: “These three designs showed the broadest score distributions, suggesting their ability to detect a broader range of compaction”

The wording here does not clarify if this is a population of nuclei or different states of compaction, consider the following replacements:

“These three designs showed a variation in compaction states”

“These three designs showed different states/levels of compaction”

“These three designs showed a population of nuclei with a range of compaction states”

- In 190: it may be worth mentioning that although for the most part you obtained similar FRET-FISH scores with the different cell line, that there were some differences detected. e.g., G1-S5- is unimodal.

- In 212-213: “the resulting FRET-FISH score distribution displayed two separate modes indicative of different chromatin compaction states”

It is worth highlighting that the AF dyes were in fact more sensitive.

Suggested wording: “the resulting FRET-FISH score distribution displayed two separate modes indicative of a higher sensitivity for AF488 and AF594 in detecting different compaction states.”

- In 230: The authors have taken the time to carefully think about what may contribute to different compaction states in the bimodal FRET-FISH curves, including potentially homolog differences. Therefore, it will be much clearer to preface the section with how homolog differences may be one of the factors contributing to the bimodal FRET-FISH distribution with a more concise version of what the authors responded with to my second comment:

“The current FRET-FISH probe design allows to measure chromatin compaction on the two homologue chromosomes for any locus of interest at the level of individual cells. Hence, to be precise, our assay can only distinguish between inter-homologue differences”

-In 235: To emphasize the strength of this data and this section consider prefacing this section with “Since lack of chromatin accessibility is used as a proxy for chromatin compaction...:

-For figures with D/A and FRET-FISH composites: it will be easier to follow if the individual images are pseudo-colored with the same colors as composites: e.g., Fig 1D, 3A, S8A

-For Figure 3C: I think supplemental Fig S10A is more powerful in communicating the differences observed in FRET-FISH scores in cLADs vs iLADs and I would recommend including the supplemental panel 10A here instead.

- In 368-370: Consider rephrasing the sentence to highlight the ability of FRET-FISH in distinguishing the different compaction states in different cell-cycle stages.

“ These results demonstrate that the compaction state of a given locus is influenced by the cell cycle phase, and highlight the ability of FRET-FISH in distinguishing the different compaction states in different cell-cycle stages, in addition to the radial position in the nucleus.”

- The authors write out and explain the interpretation, including possible factors contributing to the bimodality of FRET-FISH curves to my comment #2 beautifully. I recommend including a concise version of one of those responses in the discussion. Here are a few examples of the specific responses I refer to:

“Therefore, our current working hypothesis is that the bimodality in the FRET-FISH score distributions represents different states of local compaction that is influenced by both cell-intrinsic (nuclear position, cell cycle phase, passage number) and cell-extrinsic (external stimuli affecting global chromatin condensation) factors.”

or

“the bimodal shape of the FRET-FISH score distributions is probably determined by multiple inter-playing variables, including cell cycle phase, distance to the nuclear lamina, and cell age. As more factors influencing local compaction likely exist, we believe that FRET-FISH represents a powerful tool that can be harnessed to investigate this important feature of chromatin at the nanoscale.”

Reviewer #4 (Remarks to the Author):

The authors did a very good job implementing new experiments, improving figures and modifying text.

I am fully satisfied with the added data and explanation to some of my points. In particular, I realized that my suggestions to use additional fluorophores for complementary immunostaining or hybridization

was not a good idea. Although it remains enigmatic why FRET-FISH score distributions are bimodal, the authors convincingly showed such distributions for multiple loci on various chromosomes and these data are solid.

I liked this work as an initial submission and after all improvements and alterations suggested by all 4 referees, I definitely support its publication in the present state.

Point-by-point response to the Reviewers' comments

We would like to thank all four Reviewers once again for their time and insightful suggestions that helped us tremendously in revising and strengthening our manuscript since the first submission. We have now addressed all the remaining remarks in our final revised manuscript. Below we respond to each Reviewer individually.

Reviewer #1

Reviewer #1: The authors have presented a clarified revised version of the paper that I feel sufficiently addresses all of the major and minor concerns I had about the initial submission. In particular I found the new data on combined FRET-FISH and single-molecule FISH (smFISH) with a probe targeting Xist RNA (new Fig S8) to be very exciting although experimentally challenging. I really appreciate the new amount of data put in the revised version using FRET-FISH probes targeting regions inside cLADs and iLADs on chr18 (Fig. 3b and c) as well as the additional 3D FISH distance measurements at the ogt locus (new Supplementary Fig. 2d and e).

We thank the Reviewer for their kind words and are glad to hear that our newly added data were highly appreciated.

Reviewer #2

Ana Mota et al. in the submitted revised manuscript "FRET-FISH probes chromatin compaction at individual genomic loci in single cells" have considerably improved the quality of the presented work. Thus, I am happy to recommend this manuscript to be accepted for publishing in Nature Communications after a minor revision:

We thank the Reviewer for appreciating our efforts and supporting publication of our revised manuscript in *Nature Communications*.

Minor points

1. The authors claim that their protocol features the mildest denaturation conditions among published oligo-based DNA FISH. I find this information very useful, and the reader should be presented with a summary of the reported protocols. This also would strengthen the manuscript. Please include a table summarizing hybridization conditions reported by other studies.

We thank the Reviewer for this nice suggestion. Accordingly, in the newly added **Supplementary Data 2** we now present a summary of hybridization conditions used in various oligonucleotide-based DNA FISH assays.

2. The authors write „...we only relied on the intensity of DNA staining by Hoechst 33342 to classify cells in different cell cycle phases and could not reliably distinguish between G1, S and G2/M

cells.” This statement is strange since several FACS protocols, which distinguish G1, S and G2/M stages, rely on Hoechst/DAPI staining. Besides this, M-phase can be reliably detected using fluorescence microscopy of Hoechst 33342 stained cells. Please revise this statement and correct the manuscript accordingly.

We thank the Reviewer for this comment. Indeed, Hoechst/DAPI staining is routinely used for distinguishing cells coming from different cell cycle phase by FACS. We would argue, however, that this distinction is aided by the fact that FACS relies on extra parameters such forward and side scattering. In our microscopy-based analysis, we relied purely on Hoechst staining and, even when we tried to add information about nuclear area, we did not manage to easily separate S from G2 phase. We want to point out, though, that we do use Hoechst staining to distinguish between G1, S and G2 phases whenever possible. However, in our experience, this is highly cell type dependent. For some cell types this simple staining allows us to observe different staining behavior for the different cell cycle phases. This however does not work well for all the cell types and unfortunately MEFs were among those. Having said that, we can surely separate mitotic cells from the rest and those were not present in the non-G1 cell population.

Reviewer #3

The authors have satisfactorily responded to my comments. Therefore, I think this manuscript is appropriate for publication in Nature Communications with some minor clarification of the data and/or rephrasing some of the text in the manuscript for better clarity, see my comments below:

We thank the Reviewer for appreciating our efforts to revise the manuscript following their advice and for supporting its publication in *Nature Communications*.

-In 46: “during prolonged cell culturing”, consider replacing with “increasing passage number”

We thank the Reviewer for this suggestion and have now replaced it in our revised manuscript.

-In 59: pericentromeric regions are in fact transcribed, see following reference for overview: <https://www.frontiersin.org/articles/10.3389/fgene.2018.00674/full>

Thank you for this remark. We have now rephrased this sentence as following:

<< On the other hand, inactive chromatin encompasses genomic regions that are typically less transcribed compared to active chromatin and is decorated by heterochromatic proteins like heterochromatin protein 1 (HP1) or histone H3 methylated on lysine 9 (H3K9me). >>

- In 97: “select loci” not selected loci.

We have corrected this.

- In 99: "...cells with increasing passage number, as well as in cells in different phases of the cell division cycle"

Rephrase to: "...cells with increasing passage number as well as in different cell-cycle stage"

We have corrected this.

- In 100-101:

"We conclude that FRET-FISH is a sensitive assay for...assessing cell-to-cell variability in compaction". Consider rephrasing to "assessing different compaction states within a cell population" since this is what the paper really shows rather than cell-to-cell variation.

We thank the Reviewer for this suggestion and we have now rephrased the sentence as suggested.

- In 113: In my first comment in the first round of review, I asked to provide a rationale for selecting the MYC gene locus, genomic size, # oligos used... etc. I think this is the place in the manuscript where the authors could showcase how carefully this experiment was designed and thought through while referring to Sup. Fig4 d,e and Sup. Table2 . Including a polished and concise version of your response to my first comment here:

"In our initial proof-of-principle experiments, we designed the Myc probe based on our extensive experience with iFISH probes (PMID: 30967549), which are typically composed of 96 oligos and target a locus of ~8 kb. We found that this number of oligos and probe size/span (i.e., the genomic distance from the first to the last oligo in a probe) is a good compromise between resolution (i.e., the minimum size of a locus that can be detected) and sensitivity..."

We agree with the Reviewer that this information should be included early on when we discuss the design of the first FRET-FISH probes. Accordingly, we have now added the following text to the paragraph in which we describe *MYC* probes:

<< In these initial proof-of-principle experiments, we designed the probes based on our extensive experience with iFISH probes (PMID: 30967549), which are typically composed of 96 oligos and target a locus of ~8 kb. We found that this number of oligos and probe size/span (i.e., the genomic distance from the first to the last oligo in a probe) is a good compromise between resolution (i.e., the minimum size of a locus that can be detected) and sensitivity. However, we reasoned that, for FRET-FISH, we might need to increase the number of D and A oligos per probe, since enough D and A oligos need to be simultaneously bound to their target region for FRET to occur. Therefore, we designed these FRET-FISH probes to contain ~130 oligos for each fluorophore type (**Supplementary Table 1**). >>

As the Reviewer can see, we cite here the table in which we list how many oligos we included in each of the probes used throughout the manuscript. However, we could not cite the **Supplementary Fig 4d, e** here given that those data were generated with probes targeting

the *Ogt* locus described in a later paragraph. We hope that the Reviewer will accept this compromise.

- In 184-185: “These three designs showed the broadest score distributions, suggesting their ability to detect a broader range of compaction”

The wording here does not clarify if this is a population of nuclei or different states of compaction, consider the following replacements:

“These three designs showed a variation in compaction states”

“These three designs showed different states/levels of compaction”

“These three designs showed a population of nuclei with a range of compaction states”

Thank you for the suggestions. We have now rephrased the sentence as following, trying to synthesize the three versions suggested by the Reviewer:

<< For design G1-S50, G1-S150 and G2-S50 the score distributions were relatively broad, indicating their ability to detect a variation in compaction states. >>

- In 190: it may be worth mentioning that although for the most part you obtained similar FRET-FISH scores with the different cell line, that there were some differences detected. e.g., G1-S5- is unimodal.

We thank the Reviewer for pointing that out. We have not rephrased the description of the distributions accordingly:

<<For design G1-S50, G1-S150 and G2-S50 the score distributions were relatively broad, indicative of their ability to detect a variation in compaction states. Moreover, the distributions of the G1-S150 and G2-S50 designs were clearly bimodal, featuring a higher mode on the right—presumably corresponding to *Ogt* loci with more compacted chromatin—and a lower mode on the left likely corresponding to a less compacted state (**Fig. 1f-g**). >>

- In 212-213: “the resulting FRET-FISH score distribution displayed two separate modes indicative of different chromatin compaction states”

It is worth highlighting that the AF dyes were in fact more sensitive.

Suggested wording: “the resulting FRET-FISH score distribution displayed two separate modes indicative of a higher sensitivity for AF488 and AF594 in detecting different compaction states.”

We thank the Reviewer for this suggestion and have included it in the revised manuscript:

<< Of note, when we labeled the *Ogt* S1-G150 probe with AF488 and AF594 dyes, the resulting FRET-FISH score distribution displayed two separate modes even more clearly than in the case of probes labelled with Cy3 and Cy5 dyes, indicative of a higher sensitivity for AF488 and AF594 in detecting different chromatin compaction states (**Supplementary Fig. 3d**). >>

- In 230: The authors have taken the time to carefully think about what may contribute to different compaction states in the bimodal FRET-FISH curves, including potentially homolog differences. Therefore, it will be much clearer to preface the section with how homolog differences may be one of the factors contributing to the bimodal FRET-FISH distribution with a more concise version of what the authors responded with to my second comment:

“The current FRET-FISH probe design allows to measure chromatin compaction on the two homologue chromosomes for any locus of interest at the level of individual cells. Hence, to be precise, our assay can only distinguish between inter-homologue differences”

We thank the Reviewer for this suggestion. However, we felt that the above sentence would be more appropriate in the Discussion section where the advantages and limits of FRET-FISH are discussed. Hence, we have added this sentence at the end of the first paragraph in the Discussion (starting at line 408 in the final revised manuscript).

-In 235: To emphasize the strength of this data and this section consider prefacing this section with “Since lack of chromatin accessibility is used as a proxy for chromatin compaction...:

We thank the Reviewer for the suggestion and have now added this sentence on line 236.

-For figures with D/A and FRET-FISH composites: it will be easier to follow if the individual images are pseudo-colored with the same colors as composites: e.g., Fig 1D, 3A, S8A

We thank the Reviewer for the suggestion. We would, however, prefer to use greyscale as this is our lab’s preferred choice for presenting single-color images, since they are the most information rich.

-For Figure 3C: I think supplemental Fig S10A is more powerful in communicating the differences observed in FRET-FISH scores in cLADs vs iLADs and I would recommend including the supplemental panel 10A here instead.

We have followed the Reviewer’s suggestion swapping Fig. 3c and Suppl. Fig 10a and their respective legends.

- In 368-370: Consider rephrasing the sentence to highlight the ability of FRET-FISH in distinguishing the different compaction states in different cell-cycle stages.

“ These results demonstrate that the compaction state of a given locus is influenced by the cell cycle phase, and highlight the ability of FRET-FISH in distinguishing the different compaction states in different cell-cycle stages, in addition to the radial position in the nucleus.”

Thank you for the suggestion. Accordingly, we have now re-written this sentence as following:

<< These results demonstrate that the compaction state of a given locus is influenced by the cell cycle phase in addition to the radial position in the nucleus and highlight the ability of FRET-FISH in distinguishing different compaction states in different phased of the cell cycle. >>

- The authors write out and explain the interpretation, including possible factors contributing to the bimodality of FRET-FISH curves to my comment #2 beautifully. I recommend including a concise version of one of those responses in the discussion. Here are a few examples of the specific responses I refer to:

“Therefore, our current working hypothesis is that the bimodality in the FRET-FISH score distributions represents different states of local compaction that is influenced by both cell-intrinsic (nuclear position, cell cycle phase, passage number) and cell-extrinsic (external stimuli affecting global chromatin condensation) factors.”

Or

“the bimodal shape of the FRET-FISH score distributions is probably determined by multiple inter-playing variables, including cell cycle phase, distance to the nuclear lamina, and cell age. As more factors influencing local compaction likely exist, we believe that FRET-FISH represents a powerful tool that can be harnessed to investigate this important feature of chromatin at the nanoscale.”

We are grateful to the Reviewer for appreciating how we wrote our proposed explanation of the bimodality of the FRET-FISH score distributions. Following the Reviewer’s suggestion, we have now re-phrased the Discussion section as following:

<< Therefore, we hypothesize that the bimodality in the FRET-FISH score distributions represents different states of local compaction, which is influenced by both cell-intrinsic (nuclear position, cell cycle phase, passage number) and cell-extrinsic (external stimuli affecting global chromatin condensation) factors. As more factors influencing local compaction likely exist, we believe that FRET-FISH represents a powerful tool that can be harnessed to investigate this important feature of chromatin at the nanoscale. >>

Reviewer #4

The authors did a very good job implementing new experiments, improving figures and modifying text.

I am fully satisfied with the added data and explanation to some of my points. In particular, I realized that my suggestions to use additional fluorophores for complementary immunostaining or hybridization was not a good idea. Although it remains enigmatic why FRET-FISH score distributions are bimodal, the authors convincingly showed such distributions for multiple loci on various chromosomes and these data are solid.

I liked this work as an initial submission and after all improvements and alterations suggested by all 4 referees, I definitely support its publication in the present state.

We are very grateful to this Reviewer for their kind words of appreciation of our work and for supporting publication of our revised manuscript in *Nature Communications*.